# FACET: A FRAGMENT-AWARE CONFORMER ENSEMBLE TRANSFORMER

**Duy M. H. Nguyen**[1,2,3]   **Trung Q. Nguyen**[3]   **Ha T. H. Le**[3]   **Mai Thanh Nhat Truong**[3]
**TrungTin Nguyen**[4,5]   **Nhat Ho**[6]   **Khoa D Doan**[7]   **Duy Duong-Tran**[8]   **Li Shen**[8]
**Daniel Sonntag**[3,9]   **James Zou**[10]   **Mathias Niepert**[1,2]   **Hyojin Kim**[11]   **Jonathan E Allen**[11]

[1]Max Planck Research School for Intelligent Systems (IMPRS-IS)   [2]University of Stuttgart
[3]German Research Center for Artificial Intelligence (DFKI)
[4]ARC Centre of Excellence for the Mathematical Analysis of Cellular Systems
[5]School of Mathematical Sciences, Queensland University of Technology
[6]University of Texas at Austin   [7]VinUniversity   [8]University of Pennsylvania
[9]Oldenburg University   [10]Stanford University   [11]Lawrence Livermore National Labs

## ABSTRACT

Accurately predicting molecular properties requires effective integration of structural information from both 2D molecular graphs and their corresponding equilibrium conformer ensembles. In this work, we propose FACET, a scalable Structure-Aware Graph Transformer that efficiently aggregates features from multiple 3D conformers while incorporating fragment-level information from 2D graphs. Unlike prior methods that rely on static geometric solvers or rigid fusion strategies, our approach utilizes a differentiable graph transformer to theoretically approximate the computationally expensive Fused Gromov-Wasserstein (FGW), enabling dynamic and scalable fusion of 2D and 3D structural information. We further enhance this mechanism by injecting fragment-specific structural priors into the attention layers, enabling the model to capture fine-grained molecular details. This unified design scales to large datasets, handling up to 75,000 molecules and hundreds of thousands of conformers, and provides over a 6x speedup compared to geometry-aware FGW-based baselines. Our method also achieves state-of-the-art results in molecular property prediction, Boltzmann-weighted ensemble modeling, and reaction-level tasks, and is particularly effective on chemically diverse compounds, including organocatalysts and transition-metal complexes. We provide implementations at this link: Code implementation

## 1 INTRODUCTION

Machine learning has become a powerful tool for predicting molecular properties, with wide-ranging applications in drug discovery and materials science (Choudhary et al., 2022; Fedik et al., 2022; Batatia et al., 2023). Most existing models rely either on 2D molecular graphs, which efficiently capture topological connectivity (Xu et al., 2018; Veličković et al., 2018), or on 3D representations derived from a single conformer (Schütt et al., 2017; Batzner et al., 2022; Batatia et al., 2022). While 2D graphs are computationally efficient, they lack geometric information that is often critical for accurate property prediction. Incorporating 3D conformers helps address this by introducing spatial features such as bond lengths, and torsion angles. However, relying on a single conformer still fails to capture the intrinsic flexibility of molecular structures.

In reality, molecules dynamically sample a range of thermodynamically accessible conformations due to bond rotations, vibrations, and environmental interactions (Ramsundar et al., 2019). As a result, many experimentally observable properties such as solubility and binding affinity depend on the full ensemble of conformers a molecule can adopt (Perola & Charifson, 2004). Yet, fully modeling this distribution is computationally prohibitive, as generating and evaluating large numbers of conformers using quantum methods is costly (Medrano Sandonas et al., 2024). This has motivated hybrid models that combine the structural efficiency of 2D graphs with the geometric richness of a small and representative subset of 3D conformers. By jointly capturing topological and spatial variation, hybrid models offer scalable and expressive frameworks for molecular representation, enabling more accurate prediction of conformation-sensitive properties across a range of chemical and biological tasks.

Building on this hybrid paradigm, recent methods have introduced hybrid models that integrate 2D molecular graphs with 3D conformer information to capture both topological and spatial features (Zhu et al., 2024b; Axelrod & Gomez-Bombarelli, 2023). Despite the successes, these methods often assume conformers contribute equally or can be reweighted without considering deeper geometric context. In practice, only a subset of conformers may be thermodynamically or functionally relevant, and naive aggregation overlooks their spatial relationships, such as alignment or structural similarity. Moreover, current strategies rarely leverage interactions between 2D structural priors and 3D conformational variability, hindering the formation of truly expressive representations.

To address this, structure-aware ensemble methods based on optimal transport, especially those using fused Gromov-Wasserstein (FGW) alignment, have shown promise Ma et al. (2023); Nguyen et al. (2024a). By aligning both feature and geometric spaces, these models better preserve spatial correspondences across conformers and enable expressive ensemble aggregation. However, such methods are computationally expensive and struggle to scale to large molecular datasets such as Drugs-75k Zhu et al. (2023); Axelrod & Gomez-Bombarelli (2022), limiting their utility for high-throughput applications in generative biology.

To address scalability challenges in geometry-aware molecular modeling, we introduce a novel approach that replaces expensive FGW alignment with efficient attention-based conformer aggregation. By supervising the model with FGW distances during training, we learn a latent embedding space where conformer similarities reflect both topological and geometric structure. This enables fast, permutation-invariant conformer integration suitable for large-scale generative pipelines. Beyond efficiency, we further enrich our model with fragment-level structural priors from 2D molecular graphs, injecting chemically meaningful hierarchies into both message passing and 3D attention layers. This unified 2D–3D framework captures fine-grained spatial and topological interactions essential for applications such as molecular property prediction, virtual screening, and functional optimization. In summary, our key contributions are:

- We propose a **scalable, geometry-aware conformer aggregation framework**, denoted as FACET, that replaces costly FGW alignment with a trainable Graph Transformer, enabling efficient, deterministic attention-based inference. We further provide theoretical bounds on the approximation error relative to FGW distances.

- We introduce a unified 2D–3D representation learning approach that embeds **fragment-level structural priors** into both 2D message passing and 3D spatial self-attention, capturing multi-scale interactions between molecular topology and geometry.

- Our method delivers over $6\times$ **faster aggregation** than prior geometry-aware baselines and achieves **state-of-the-art performance** across six benchmarks, including molecular property prediction and Boltzmann-weighted ensemble tasks, demonstrating robustness across diverse molecular scenarios and dataset scales.

## 2 RELATED WORK

**Conformer Ensemble Learning in Molecular Representations.** Traditional molecular representations span connectivity fingerprints (Morgan, 1965), 1D string encodings (Ahmad et al., 2022; Wang et al., 2019), 2D topological graphs (Yang et al., 2019a; Rong et al., 2020), and 3D geometric graphs (Fang et al., 2021; Zhou et al., 2023). 3D models typically rely on a single conformer, overlooking the fact that molecules often adopt multiple low-energy conformations, which can serve as informative features, particularly in capturing thermodynamic properties. Hybrid approaches now combine 2D graphs with ensembles of 3D conformers (Zhu et al., 2024b; Wang et al., 2024), aggregated via mean pooling, DeepSets (Zaheer et al., 2017), or self-attention (Vaswani et al., 2017). More advanced geometry-aware methods based on Fused Gromov-Wasserstein (FGW) alignment (Ma et al., 2023; Nguyen et al., 2024a) capture both feature and structural similarity across conformers, but remain computationally costly and scale poorly to large datasets (e.g., Drugs-75k) or foundation models (Zhou et al., 2023; Chithrananda et al., 2020). To address this, we propose a scalable framework that learns latent embeddings of 3D conformers with graph transformers, *integrating geometry-aware signals inspired by FGW and hierarchical fragment-level features*. This yields a permutation-invariant, expressive, and efficient method.

**Scalable Optimal Transport for Graph Learning.** Learning-based approximations of Optimal Transport (OT) have emerged as efficient alternatives to traditional solvers. Early works introduced

differentiable Sinkhorn distances with entropic regularization for stability and scalability (Cuturi, 2013; Feydy et al., 2019; Genevay et al., 2018). Later methods improved efficiency via structural assumptions - e.g., low-rank factorization (Scetbon et al., 2021; Cuturi et al., 2020) and spatial geometry (Bachmann et al., 2022; Solomon et al., 2015). Meta-learning approaches further accelerated convergence by learning initialization schemes (Amos et al., 2023). More recently, neural OT surrogates trained directly on data have bypassed iterative solvers entirely (Courty et al., 2017; Tong et al., 2021; Haviv et al., 2024).

However, prior works focus on standard OT and fail to extend to structure-aware variants like FGW, which jointly capture node attributes and graph topology. To address this, we introduce the first learned approximation of FGW with a graph transformer, enabling scalable, geometry-aware conformer aggregation. By embedding fragment-level priors into both 2D and 3D encoders, our approach supports multi-scale reasoning across topological and spatial hierarchies, effectively bridging molecular graphs with 3D conformational diversity.

**Fragment-biases in Molecular GNN.** Fragment-level substructures - such as rings, functional groups, and pharmacophores - are key to molecular property prediction and drug design (Merlot et al., 2003; Varnek et al., 2005). Recent works have leveraged these motifs for scaffold-aware drug discovery (Lee et al., 2024; Chan et al., 2024), self-supervised learning via fragment-based masking or contrastive tasks (Rong et al., 2020; Zhang et al., 2021; Wen et al., 2024), and GNN architectures that encode fragment-level inductive biases (Wang et al., 2025; Wollschläger et al., 2024). These methods show that fragments enhance generalization, interpretability, and data efficiency. Building on these insights, we explore a complementary direction: *integrating fragment-level priors into hybrid 2D–3D ensemble models*. In our approach, fragment hierarchies are embedded into both 2D message-passing and 3D spatial attention layers, enabling multi-scale processing across molecular topology and geometry. This design improves conformer aggregation and yields more expressive, geometry-aware representations suited for conformation-sensitive tasks.

## 3 FRAGMENT-AWARE CONFORMER ENSEMBLE TRANSFORMER

**Notation.** Let $\Delta_N := \{\boldsymbol{\omega} \in \mathbb{R}_+^N : \boldsymbol{\omega}^\top \mathbf{1}_N = 1\}$ denote the probability simplex, where $\mathbf{1}_N$ is the all-ones vector in $\mathbb{R}^N$. For $x \in \boldsymbol{\Omega}$, $\delta_x$ is the Dirac measure at $x$. We write $[K] := \{1, \ldots, K\}$ for $K \in \mathbb{N}$, and use $\langle \cdot, \cdot \rangle$ to denote the Frobenius inner product. For a tensor $\mathbf{L} = (L_{ijkl})$ and matrix $\boldsymbol{B} = (B_{kl})$, define the contraction $\mathbf{L} \otimes \boldsymbol{B} := \left( \sum_{kl} L_{ijkl} B_{kl} \right)_{ij}$. A graph $G = (V, E)$ has $N := |V|$ nodes and edges $E \subseteq \{\{u, v\} \subseteq V : u \neq v\}$. An attributed graph is given by $\mathcal{G} := (\boldsymbol{H}, \boldsymbol{A}, \boldsymbol{\omega})$, where $\boldsymbol{H} \in \mathbb{R}^{N \times d}$ is the node feature matrix (with row $\boldsymbol{H}_v$ for node $v$), $\boldsymbol{A} \in \mathbb{Z}_+^{N \times N}$ encodes structure (e.g., adjacency or shortest-path distance matrix), and $\boldsymbol{\omega} \in \Delta_N$ is a node weight distribution.

Given two graphs $\mathcal{G}_1$ and $\mathcal{G}_2$ with $N_1$ and $N_2$ nodes, the *Fused Gromov-Wasserstein (FGW)* distance (Peyré et al., 2016; Titouan et al., 2019; 2020) is: $\text{FGW}_{p,\alpha}(\mathcal{G}_1, \mathcal{G}_2) := \min_{\boldsymbol{\pi} \in \Pi(\boldsymbol{\omega}_1, \boldsymbol{\omega}_2)} \langle (1 - \alpha) \boldsymbol{M} + \alpha \mathbf{L}(\boldsymbol{A}_1, \boldsymbol{A}_2) \otimes \boldsymbol{\pi}, \boldsymbol{\pi} \rangle$, where $\Pi(\boldsymbol{\omega}_1, \boldsymbol{\omega}_2) := \{\boldsymbol{\pi} \in \mathbb{R}_+^{N_1 \times N_2} : \boldsymbol{\pi} \mathbf{1}_{N_2} = \boldsymbol{\omega}_1, \boldsymbol{\pi}^\top \mathbf{1}_{N_1} = \boldsymbol{\omega}_2\}$ is the set of valid couplings, $\boldsymbol{M}[i, j] = d_f(\boldsymbol{H}_1[i], \boldsymbol{H}_2[j])^p$ is the distance between feature of node $i$ in $\mathcal{G}_1$ and of node $j$ in $\mathcal{G}_2$, $\mathbf{L}(\boldsymbol{A}_1, \boldsymbol{A}_2)[i, j, l, m] = |\boldsymbol{A}_1[i, j] - \boldsymbol{A}_2[l, m]|^p$ captures structural mismatch, and $\alpha \in [0, 1]$ balances feature and structure alignment. Consider a set of $K$ graphs $\{\mathcal{G}_k\}_{k=1}^K$, the *FGW barycenter graph* $\overline{\mathcal{G}}$ is the graph that has the smallest distances to other graphs in the set: $\mathcal{G} := \arg \min_{\mathcal{G}} \sum_{k=1}^K \lambda_k \text{FGW}_{p,\alpha}(\mathcal{G}, \mathcal{G}_k)$

### 3.1 CONFORMER GENERATION

Following prior work, we generate molecular conformers using distance geometry methods that convert interatomic constraints - such as bond lengths, angles, stereochemistry, and steric limits - into 3D coordinates (Hawkins, 2017). A lightweight force field refines the structures toward low-energy conformations. Compared to quantum methods like DFT, this approach is highly scalable and efficient for large datasets. As in prior studies (Raza et al., 2022; Nguyen et al., 2024b), we use RDKit (Landrum, 2016) for fast and reliable conformer generation.

### 3.2 FRAMEWORK OVERVIEW

We propose a neural architecture with three main components (Fig. 1). First, a 2D message passing neural network (MPNN) captures molecular topology, while another 2D-MPNN operates on a fragment-level graph, which consists of pairwise edges between fragment nodes, to encode higher-

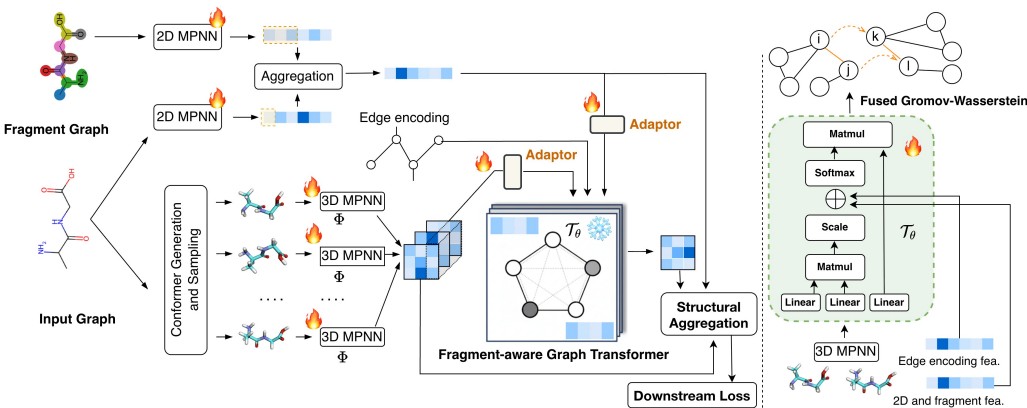

Figure 1: **FACET overview**. The model takes a 2D molecular graph and its fragment graph, encoded by separate 2D-MPNNs and aligned via fragment–atom correspondence (dashed boxes). In parallel, multiple 3D conformers are sampled and encoded by a shared 3D-MPNN ($\Phi$). The resulting 2D/3D features and edge information are adapted and fused through a **frozen fragment-aware graph attention** module ($\mathcal{T}_\theta$), pre-trained (right) using a graph transformer supervised by **Fused Gromov–Wasserstein** (FGW) distances to preserve FGW geometry (Sec. 5.1). The fused, geometry-aware representations are then used for downstream prediction. "*Fire*" and "*snowflake*" icons denote trainable and frozen components, respectively.

order structural priors (Sec. 3.3). Their outputs are fused and refined through a lightweight adaptor module before entering a pre-trained FGW-guided graph transformer (Sec. 3.4). For 3D information, a set of conformers is sampled from the input molecule, and a 3D-MPNN extracts conformer embeddings (Sec. 3.4.1), which are also calibrated by an adaptor layer to handle variability between 2D and 3D features. Then, conformer embeddings are fed into the graph transformer, where each node attends to all other nodes, taking into account the conformers graph structure and fragment-level information (Sec. 3.4.2). In essence, the graph transformer encodes conformer embeddings into another space where their pairwise Euclidean distance is equal FGW distance (Sec. 3.4.3). Finally, a permutation- and E(3)-invariant fusion module unifies the 2D and 3D features into a single embedding for downstream tasks (Sec. 3.4.4).

## 3.3 FRAGMENT-ENHANCED 2D MOLECULAR GRAPH

Each molecule is represented as a 2D graph $G = (V, E)$, where nodes $V$ correspond to atoms and edges $E$ to covalent bonds. Atom features $\mathbf{h}_v^{(0)} \in \mathbb{R}^d$ encode properties like atom type and valence, while bonds $(u, v)$ are annotated with features $e(u, v)$ (Scarselli et al., 2008; Gilmer et al., 2017). We adopt a 2D message-passing neural network (MPNN) that updates node embeddings layer-wise:

$$\mathbf{h}_v^\ell = \mathsf{UPD}^\ell(\mathbf{h}_v^{\ell-1}, \mathsf{AGG}^\ell(\mathbf{M}^\ell(\mathbf{h}_v^{\ell-1}, \mathbf{h}_u^{\ell-1}, \boldsymbol{e}_{v,u}) \mid u \in N(v))), \tag{1}$$

where $\mathbf{M}^\ell$ is a message function, $\mathsf{AGG}^\ell$ is sum aggregation, and $\mathsf{UPD}^\ell$ is identity or multilayer perception layers. We use Graph Attention Networks (GATs) (Veličković et al., 2017), where messages are computed as:

$$\mathbf{M}_{v,u}^\ell = \alpha_{v,u}^\ell \mathbf{W}^\ell \mathbf{h}_u^{\ell-1}, \quad \alpha_{v,u}^\ell = \mathrm{softmax}_u \left( \mathrm{LeakyReLU} \left( {}^\top[\mathbf{W}^\ell \mathbf{h}_v^{\ell-1}, |, \mathbf{W}^\ell \mathbf{h}_u^{\ell-1}]) \right) \right). \tag{2}$$

After $L$ layers, we obtain final atom-level features $\mathbf{h}_v^L$ for each atom $v$ used for downstream tasks.

**Fragment-Based Structural Augmentation.** To enhance atomic representations with higher-order structural context, we construct a fragment-level graph from the input molecular graph $G$ using ring-path decomposition (Kong et al., 2022; Geng et al., 2023; Wollschläger et al., 2024) to identify key substructures such as aromatic rings and functional groups (Fig. 5). Each fragment is treated as a node in a new graph $G^{\mathrm{frag}} = (V^{\mathrm{frag}}, E^{\mathrm{frag}})$, where nodes correspond to fragments and edges are induced from the connectivity in $G$, two fragments are connected if they share an atom or are directly bonded. In this work, we specifically follow the approach proposed in (Wollschläger et al., 2024), as it offers a good balance of simplicity and effectiveness for our use case.

We apply the same GAT formulation in Eq. (1) to the fragment graph to obtain fragment embeddings $\{\mathbf{h}_f^{\mathrm{frag}}\}_{f \in V^{\mathrm{frag}}}$. Then **for each atom $v$ that belongs to its fragment** $f(v)$, we fuse their atom-level representations $\mathbf{h}_v^{(L)}$ with $\{\mathbf{h}_f^{\mathrm{frag}}\}$ by:

$$\widetilde{\mathbf{h}}_v^{(L)} = \mathbf{h}_v^{(L)} + \mathrm{FFN}\left(\mathbf{h}_{f(v)}^{\mathrm{frag}}\right), \tag{3}$$

where $\mathrm{FFN}(\cdot)$ is a learnable feedforward network that projects fragment-level context into the same space as atom features. Finally, we define a fragment-enhanced graph level representation that is computed by applying a readout function $\mathbf{h}_{2\mathrm{D}} = \mathsf{READOUT}\left(\{\widetilde{\mathbf{h}}_v^{(L)} \mid v \in V\}\right) = \sum_{v \in V} \widetilde{\mathbf{h}}_v^{(L)}$. Intuitively, the **dual-level encoding** combining local atomic features and global fragment-level context as Eq.(3) allows the model to **reason over both fine-grained and coarse-grained structures**, enhancing the expressivity of the molecular representation.

## 3.4 LEARNING GRAPH TRANSFORMER FOR 3D MOLECULE AGGREGATIONS

A molecular conformer is represented as a set $S = \{\mathbf{r}_i, Z_i\}_{i=1}^N$, where $N$ denotes the number of atoms, $\mathbf{r}_i \in \mathbb{R}^3$ corresponds to the 3D Cartesian coordinates of atom $i$, and $Z_i \in \mathbb{N}$ indicates its atomic number.

**3.4.1 3D conformer feature representation.** For each conformer $S$, we can define its graph $\mathcal{G}_S$ and compute its 3D feature embedding by using the geometric MPNN SchNet (Schütt et al., 2017), though other $E(3)$-invariant neural architectures can be readily substituted without modification (Table 2). We represent the matrix of atom-level features from the final message-passing layer $L$ of SchNet as $\mathbf{H}$, where each column $\mathbf{H}[v]$ corresponds to the feature vector $\mathbf{h}_{3\mathrm{d},v}^{(L)}$ of atom $v$. We then compute the vector representation of a conformer $S$ as $\mathbf{h}_{3\mathrm{d},\mathrm{S}} = \sum_{v \in V} (\mathbf{W}_{3\mathrm{d}}) \mathbf{h}_{3\mathrm{d},v}^{(L)} + \mathbf{b}_{3\mathrm{d}} \in \mathbb{R}^d$ with $\mathbf{W}_{3\mathrm{d}}$ and $\mathbf{b}_{3\mathrm{d}}$ are learnable vectors. Given a set of $K$ conformers $\{\mathbb{S}_k\}_{k=1}^K$, we define $\mathbf{H}_{3\mathrm{d}}[k] = \mathbf{h}_{3\mathrm{D},\mathbb{S}_k}$ as the feature embedding of the $k$-th conformer. The matrix $\mathbf{H}_{3\mathrm{d}} \in \mathbb{R}^{K \times d}$ thus summarizes the feature representations of all conformers in the set.

**3.4.2 Fragment-aware Graph Transformer.** Given the atom-wise feature matrix $\mathbf{H}$ for each conformer $S$, we aim to learn structure-encoded latent representations using Graph Transformer architectures (Ying et al., 2021; Kreuzer et al., 2021; Luo et al., 2024). We adopt the architecture from Ying et al. (2021) due to its strong expressiveness on small molecular graphs, and further *extend its attention mechanism with fragment sub-structures* (Fig .5). It is important to note that our framework is flexible and can incorporate alternative transformer-based models.

In particular, we compose $N$ transformer layers (Vaswani et al., 2017), each consisting of a self-attention mechanism followed by a position-wise feed-forward network. Given $\mathbf{H} = [\mathbf{h}_1^\top, \ldots, \mathbf{h}_n^\top]^\top \in \mathbb{R}^{n \times d}$ computed in Section 3.4.1 by a 3D-MPNN, where $\mathbf{h}_i = \mathbf{h}_{3\mathrm{d},v_i}^{(L)} \in \mathbb{R}^{1 \times d}$ is the vector embedding of an atom $v_i$ with $d$ dimensions. We compute self-attention, by linearly projecting $\mathbf{H}$ into query ($\mathbf{Q}$), key ($\mathbf{K}$), and value ($\mathbf{V}$) matrices using learned weights $\mathbf{W}_Q, \mathbf{W}_K, \mathbf{W}_V \in \mathbb{R}^{d \times d}$:

$$\mathbf{Q} = \mathbf{H}W_Q, \ \mathbf{K} = \mathbf{H}W_K, \ \mathbf{V} = \mathbf{H}W_V, \ \widetilde{\mathbf{A}} = \mathbf{Q}\mathbf{K}^\top/\sqrt{d}, \quad \mathrm{Attention}(\mathbf{H}) = \mathrm{softmax}(\widetilde{\mathbf{A}})\mathbf{V}. \quad (4)$$

Here, $\widetilde{\mathbf{A}}$ denotes the attention score matrix representing pairwise similarities between tokens. For clarity, we present the single-head version; extending to multi-head attention is straightforward. Bias terms are omitted for brevity.

While the attention in Eq. (4) operates only on feature nodes, leveraging the structural information of the 3D conformer graph is essential. Following Ying et al. (2021), we incorporate (i) `centrality encoding`, which measures the importance of a node in the graph via its degree, and (ii) `spatial encoding`, which captures the spatial relation between two nodes $v_i$ and $v_j$ in $\mathcal{G}_S$ using the shortest path distance (SPD) (Cormen et al., 2022; Balaban, 1985), augmented with a learnable weight assigned to each edge along the SPD. Specifically, we incorporate (i) by:

$$\mathbf{h}_i = \mathbf{h}_i + z_{\deg^-(v_i)}^- + z_{\deg^+(v_i)}^+, \quad (5)$$

where $z^-, z^+ \in \mathbb{R}^d$ are learnable embedding vectors indexed by the indegree $\deg^-(v_i)$ and outdegree $\deg^+(v_i)$ of atom $v_i$ respectively. For (ii), the shortest-path distance (SPD) matrix is first computed, and these distances are used to retrieve the corresponding embeddings, which are then integrated into the attention mechanism to inject topology-aware structural bias. Assume $\widetilde{\mathbf{A}}_{ij}$ as the $(i,j)$-element of the Query-Key product matrix $\widetilde{\mathbf{A}}$, the condition (ii) extends $\widetilde{\mathbf{A}}_{ij}$ as:

$$\widetilde{\mathbf{A}}_{ij} = (\mathbf{h_i}\mathbf{W}_Q)(\mathbf{h}_j\mathbf{W}_K)^T/\sqrt{d} + s_{\phi(v_i,v_j)} + c_{ij}, \quad (6)$$

where $s_{\phi(v_i,v_j)}$ is a learnable scalar indexed by the SPD distance $\phi(v_i,v_j)$ and shared across layers; $c_{ij} = \mathbb{E}(x_{e_n}(w^E n)^T)$, with $x_{e_n}$ the feature of edge $e_n$ in $\mathrm{SPD}_{ij}$, $w_n^E \in \mathbb{R}^{d_E}$ its weight embed-

ding, and $d_E$ the dimensionality of edge features, computed as the difference between the feature embeddings of its incident nodes.

While the spatial encoding in Eq.(6) is implicated by the SPD, we argue that this might inadequately capture chemically meaningful substructures (ablation in Tab. 5). This motivates us to extend attention scores in Eq. (6) using values derived from (iii) `fragment-level node features` computed on 2D topology graph in Eq. (3), directly *guiding attention toward structurally and functionally relevant regions* such as rings, functional groups, or scaffolds. To this end, we compute an adjacency-like matrix $\mathbf{A}(G)$ using cosine distance over the final node embeddings $\widetilde{\mathbf{h}}_v^{(L)}$. Specifically, for each pair of atoms $(v_i, v_j)$ in the 2D molecular graph, we define

$$\mathbf{A}(G)_{ij} = 1 - \frac{\langle \widetilde{\mathbf{h}}_i^{(L)}, \widetilde{\mathbf{h}}_j^{(L)} \rangle}{|\widetilde{\mathbf{h}}_i^{(L)}|_2 \cdot |\widetilde{\mathbf{h}}_j^{(L)}|_2}, \tag{7}$$

which quantifies their directional dissimilarity in the embedding space. Finally, we compute the attention score as:

$$\widetilde{\mathbf{A}}_{ij} = (\mathbf{h_i}\mathbf{W}_Q)(\mathbf{h}_j\mathbf{W}_K)^T / \sqrt{d} + s_{\phi(v_i, v_j)} + c_{ij} + \mathbf{A}(G)_{ij}. \tag{8}$$

**3.4.3 Learning to Approximate FGW distance.** We denote $\mathcal{T}_\theta(.)$ as the graph transformer model whose attention operation is Eq.(8), our goal is to train $\mathcal{T}_\theta(\cdot)$ to map the feature representation of each conformer $S$ into a latent space where the $L_2$ distance between any pair $S_i, S_j$ approximates their FGW distance - an *effective*, yet *computationally expensive*, geometry-aware metric (Ma et al., 2023; Nguyen et al., 2024a). To this end, given a set of $\Omega = \{S_i\}_{i=1}^K$ of $K$ generated conformers, we sample $B$ conformers from $\Omega$, then compute their encoding features by $\mathcal{T}_\theta(\mathbf{H}_i)$ for each $S_i \in B$. These outputs are compared with their pair-wise FGW distance to optimize the loss:

$$\mathcal{L}_{\text{enc}} = \sum_{ij} \left| ||\mathcal{T}_\theta(\mathbf{H}_i) - \mathcal{T}_\theta(\mathbf{H}_j)||_2^2 - \text{FGW}_{p,\alpha}(\mathcal{G}(S_i), \mathcal{G}(S_j)) \right|. \tag{9}$$

By minimizing the loss $\mathcal{L}_{\text{enc}}$, we update the parameters of the transformation module $\mathcal{T}_\theta(\cdot)$ using gradient descent: $\theta \leftarrow \theta - \epsilon \nabla \mathcal{L}_{\text{enc}}$. Once trained, we freeze $\mathcal{T}_\theta$ and incorporate it back into the framework to compute a *geometry-aware representation* across $K$ conformers $\{\mathbb{S}_k\}_{k=1}^K$ as follows: $\overline{\mathbf{H}} = \mathbb{E}\left(\{\mathcal{T}_\theta(\mathbf{H}_i)\}_{i=1}^K\right)$, where $\overline{\mathbf{H}}$ denotes the aggregated structural embedding. Intuitively, $\overline{\mathbf{H}}$ acts as the feature embedding of the FGW barycenter in the latent space (see **Notation** at begin of Sec 3). It represents the geometric mean of the input conformers, taking into account both their structural characteristics and features. However, the 3D conformer feature distribution, extracted by 3D-MPNN, used to train $\mathcal{L}_{\text{enc}}$ (Eq. 9) may experience a *domain shift* when co-trained with other components in the full framework (Sec. 3.4) due to the continuous updating of 3D-MPNN. To address this, we design *adapter layers* as simple FFN layers to transform the input features in Eq. (9), aligning them to the seen distribution during training $\mathcal{T}_\theta$.

**3.4.4 Invariant Aggregation of 2D and 3D Representation.** We integrate representations from the 2D molecular graph and multiple 3D conformers using both average pooling and a GraphTransformer-based aggregation. The transformer captures rich spatial interactions while ensuring permutation invariance across conformers and E(3) equivariance, preserving robustness to 3D transformations. Given $K$ conformers, using $\overline{\mathbf{H}}$ as the GraphTransformer (GT)-aggregated atom features. We compute the global GT representation as: $\mathbf{h}_{\text{GT}} = \sum_{v \in V} \left(\mathbf{W}_{\text{GT}} \cdot \overline{\mathbf{h}}_v + \mathbf{b}_{\text{GT}}\right)$, where $\overline{\mathbf{h}}_v = \overline{\mathbf{H}}[v]$ and $\mathbf{W}_{\text{GT}}, \mathbf{b}_{\text{GT}}$ are learnable parameters. We then define $\mathbf{H}_{\text{2D}}$ and $\mathbf{H}_{\text{GT}}$ be the matrices whose columns are, respectively, $K$ copies of the 2D feature $\mathbf{h}_{\text{2D}}$ (Sec.3.3) and $\mathbf{h}_{\text{GT}}$ representations. We fuse those representations with the 3D conformer features $\mathbf{H}_{\text{3D}}$ to produce the final atom-wise embedding: $\mathbf{H}_{\text{comb}} = \widetilde{\mathbf{W}}_{\text{2D}} \mathbf{H}_{\text{2D}} + \widetilde{\mathbf{W}}_{\text{3D}} \mathbf{H}_{\text{3D}} + \widetilde{\mathbf{W}}_{\text{GT}} \mathbf{H}_{\text{GT}}$, where each $\widetilde{\mathbf{W}}_i$, $i \in \{2D, 3D, GT\}$ are trainable projection matrix. The combined embedding $\mathbf{H}_{\text{comb}}$ is fed into a final FFN layer to predict the target property (Sec.K Appendix).

## 4 THEORETICAL BOUNDS FOR EMBEDDING NON-EUCLIDEAN FGW

Learning a Transformer $\mathcal{T}_\theta(.)$ to predict the FGW problem is closely related to multidimensional scaling (MDS) (Torgerson, 1952). Building on recent advances (Haviv et al., 2024; Sonthalia et al., 2021), we extend MDS theory to derive bounds on the error of embedding non-Euclidean distances, specifically Wasserstein and FGW, into a Euclidean space suitable for graph transformer integration. While computing FGW barycenters is costly, our embedding enables efficient approximation via

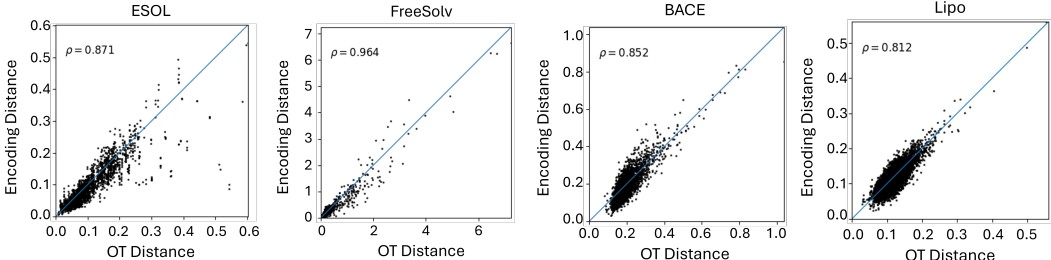

Figure 2: Correlations between FGW distance and trained GraphTransformer on four datasets in **MoleculeNet** benchmark. For each test molecule, we compute pairwise FGW distances between conformers and compare them with Euclidean distances between their Graph Transformer embeddings. The correlation $\rho$ is reported, with the reference line $y = x$ shown in blue.

averaging and decoding in latent space. Prior work (Haviv et al., 2024) validated this approach for Wasserstein distances; we generalize it to FGW and provide theoretical justification, offering a scalable path for structure-aware graph alignment.

**Cumulative Stress Optimization Problem via Pairwise FGW Distance Matrix.** We define the **pairwise FGW distance matrix** $D$ for a set of $K$ distributions as $D_{ij} := \mathrm{FGW}_{p,\alpha}(\mathcal{G}(S_i), \mathcal{G}(S_j))$ for all $i, j \in [K]$, following Section 3.4. The **empirical FGW barycenter** is given by $\overline{\mathcal{G}}_K \in \arg\min_{\mathcal{G} \in \mathcal{P}_p(\Omega)} \frac{1}{K} \sum_{i=1}^{K} \mathrm{FGW}_{p,\alpha}^p(\mathcal{G}, \mathcal{G}(S_i))$, where $\mathcal{P}_p(\Omega)$ denotes the space of attributed graphs with finite $p$-th order FGW distance.

To approximate this barycenter in embedding space, we require $\|\overline{e}_K - e_j\|_2^2 \approx \mathrm{FGW}_{p,\alpha}(\overline{\mathcal{G}}_K, \mathcal{G}(S_j)) := \overline{D}_{K,j}$ for all $j \in [K]$, where $\overline{e}_K = \frac{1}{K} \sum_{i=1}^{K} e_i$ is the mean embedding and $e_i := \mathcal{T}_\theta(\mathbf{H}_i)$ is the learned representation. To assess how well the embeddings $\{e_i\}_{i=1}^{K} \subset \mathbb{R}^d$ preserve both pairwise FGW distances and barycenter structure, we define the **cumulative stress**: $\mathcal{S} = \min_{e_i \in \mathbb{R}^d} \sum_{i,j \in [K]} \left( \|e_i - e_j\|_2^2 - D_{ij} \right)^2 + \sum_{j \in [K]} \left( \|\overline{e}_K - e_j\|_2^2 - \overline{D}_{K,j} \right)^2$. This objective encourages faithful reconstruction of both the distance structure and the barycenter alignment in the learned embedding space, as formalized in Theorem 1, which is proved in Appendix J.

**Theorem 1.** *Let $D$ denote the pairwise $FGW_{p,\alpha}$ distance matrix, and let $\{\lambda_i, v_i\}_{i=1}^{K}$ represent the eigendecomposition of the associated criterion matrix $F = -CDC$, where $C = I_K - \frac{1}{K}\mathbf{1}_K\mathbf{1}_K^\top$ is the centering matrix. The optimal stress value, denoted by $\mathcal{S}^*$, is bounded as follows: $\mathcal{L} \leq \mathcal{S}^* \leq \mathcal{U}$, where $\mathcal{L} := \sum_{i:\lambda_i < 0} \lambda_i^2$, $\mathcal{U} := \sum_{ij}(\Delta g_i + \Delta g_j)^2 + \mathcal{L} + \mathcal{C}$, $\Delta g_i = \frac{1}{2}\sum_{j:\lambda_j < 0} \lambda_j \cdot v_{ij}^2$. Here, $v_{ij}$ denotes the $j$-th component of the $i$-th eigenvector $v_n$ of $F$, and $\mathcal{C}$ quantifies the approximation error between the empirical barycenter in the Euclidean embedding space and the one in the original space of undirected attributed graphs.*

# 5 EXPERIMENTS

## 5.1 IMPLEMENTATION DETAILS

**General pipeline.** Our training consists of three stages. **Stage 1**: We train the 2D and 3D MPNNs independently for 150 epochs and the learning rate of $1e^{-3}$ to extract features from 2D molecular graphs and 3D conformers, used to predict molecular properties by regression loss. These extracted features also serve as a dataset to supervise the training of Graph Transformer for approximating the FGW distance. **Stage 2**: The Graph Transformer is trained separately to approximate the computationally expensive FGW distance between pairs of conformers, using the learned representations from Stage 1. We use the architecture of Graphormer (Ying et al., 2021), with 12 attention layers, 8 heads, and a hidden size of 64 (372k parameters). It is trained for 1000 epochs with a learning rate of $1e^{-5}$. **Stage 3**: We train the full model end-to-end with 2D/3D MPNNs and the Graph Transformer (300 epochs, learning rate $5e^{-4}$). Details of the training scheme are provided in Table 6 and Section H of the Appendix. To mitigate feature shift during finetuning, MLP adaptors project conformers into 64-dim refined embeddings for both 2D and 3D features before the Graph Transformer.

## 5.2 APPROXIMATION OF FGW DISTANCE VIA GRAPH TRANSFORMER

Beyond theoretical estimation, we empirically evaluate how well the Graph Transformer approximates FGW distances between conformers in Euclidean space. As shown in Figure 2, results on the MoleculeNet benchmarks reveal a strong correlation between learned embeddings and true FGW distances, validating the transformer's effectiveness in simulating costly FGW computations. While

Table 1: Number of samples for each split on molecular property prediction, classification tasks, and reaction prediction for **MoleculeNet** and the **MARCEL** benchmark.

| Dataset | Lipo | ESOL | FreeSolv | BACE | Drugs-75k | Kraken |
|---------|------|------|----------|------|-----------|--------|
| Train   | 2940 | 789  | 449      | 1059 | 52569     | 1086   |
| Valid.  | 420  | 112  | 64       | 151  | 7509      | 155    |
| Test    | 840  | 227  | 129      | 303  | 15021     | 311    |
| Total   | 4200 | 1128 | 642      | 1513 | 75099     | 1552   |

Table 2: FACET results on SchNet and VisNet.

| Model | Lipo | ESOL | FreeSolv | BACE |
|-------|------|------|----------|------|
| CONAN (VisNet) | $0.55 \pm 0.45$ | $1.03 \pm 0.12$ | $0.69 \pm 0.03$ | $0.61 \pm 0.15$ |
| CONAN-FGW | $0.50 \pm 0.01$ | $0.55 \pm 0.05$ | $0.64 \pm 0.02$ | $\mathbf{0.47 \pm 0.01}$ |
| FACET | $\mathbf{0.48 \pm 0.01}$ | $\mathbf{0.53 \pm 0.05}$ | $\mathbf{0.61 \pm 0.02}$ | $\mathbf{0.47 \pm 0.01}$ |
| CONAN (SchNet) | $0.56 \pm 0.013$ | $0.57 \pm 0.019$ | $1.50 \pm 0.16$ | $0.64 \pm 0.051$ |
| CONAN-FGW | $\mathbf{0.42 \pm 0.02}$ | $0.53 \pm 0.02$ | $1.07 \pm 0.08$ | $0.55 \pm 0.02$ |
| FACET | $\mathbf{0.42 \pm 0.01}$ | $\mathbf{0.52 \pm 0.04}$ | $\mathbf{0.97 \pm 0.08}$ | $\mathbf{0.50 \pm 0.03}$ |

correlation varies slightly across datasets, the results consistently highlight the model's reliability as a fast FGW surrogate, especially as the number of conformers in the aggregation increases.

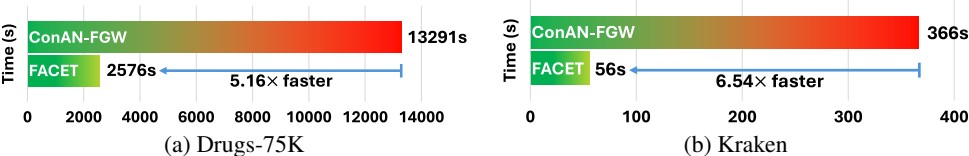

(a) Drugs-75K                    (b) Kraken

Figure 3: Comparison of the **one-epoch training time** of CONAN-FGW (Nguyen et al., 2024b) and the proposed FACET on the Drugs-75K and Kraken datasets from the **MARCEL** benchmark.

## 5.3 SCALING FRAGMENT GEOMETRY-AWARE AGGREGATION

To validate the scalability of FACET model, based on a Graph Transformer for structure-aware aggregation, we compare it against ConAN-FGW (Nguyen et al., 2024a), a method computing FGW distances on-the-fly during training and inference. We evaluate two key aspects: (i) *inference-time efficiency with varying numbers of conformers*, and (ii) *average **training time** per epoch at different dataset scales*. For inference-time, we measure the time required to generate output embeddings from $K$ conformers ($K \in 5, 10, 15, 20$) using a single GPU. Experiments are conducted on FreeSolv and BACE, which differ in node/edge distributions, to assess performance across molecular graph complexities. In addition to ConAN-FGW, we further compare FACET against strong 3D-GNN baselines (e.g., SchNet, VisNet, GemNet) to assess efficiency and accuracy relative to established geometry-aware models. In the second setting, we compare the average per-epoch training time of FACET and ConAN-FGW on two datasets of different scales: Kraken (1,086 molecules) and Drugs-75k (52,569 molecules).

As shown in Figures 4 and 10, FACET scales linearly with the number of conformers and achieves a 5–6× reduction in training time compared to ConAN-FGW (Figure 3). This improvement is especially important for large-scale training; for example, ConAN-FGW requires 1,107.58 GPU hours to train on Drugs-75K for 300 epochs, whereas FACET completes the same schedule in 214 hours, and in only 26.75 hours with 8 GPUs (vs. 138 hours for ConAN-FGW). Relative to other 3D GNN baselines, FACET provides an accuracy–efficiency balance: it avoids the heavy cost of FGW alignment while remaining competitive with - and in some cases more efficient than - existing geometric GNNs. Further analysis of these scaling behaviors is provided in Sections D and E.

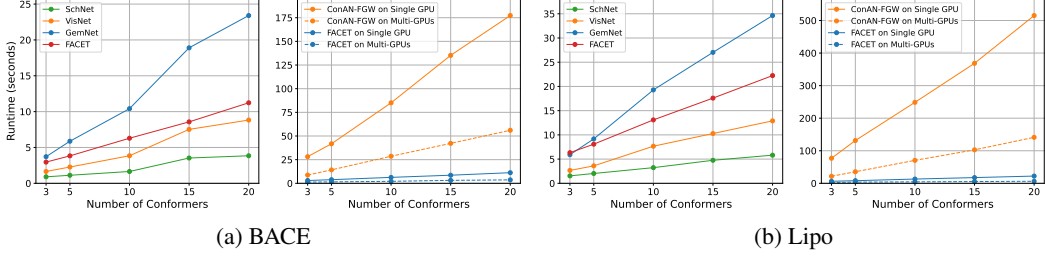

(a) BACE                    (b) Lipo

Figure 4: **Inference running time comparison** between **FACET** and other **GNN-based methods** on two datasets, BACE (a) and Lipo (b). Results are shown for both single-GPU and 4-GPU (multi-GPU) configurations. Reported runtimes represent the total time required to extract structural embeddings for all molecules in the test set of each dataset.

Table 3: Comparison of molecular property regression performance on the **MoleculeNet** benchmark (MSE ↓). The results of competing methods are adapted from Nguyen et al. (2024b). FACET uses a SchNet backbone.

| Model | Lipo | ESOL | FreeSolv | BACE |
|---|---|---|---|---|
| 2D-GAT | $1.387 \pm 0.206$ | $2.288 \pm 0.017$ | $8.564 \pm 1.345$ | $1.844 \pm 0.33$ |
| D-MPNN | $0.534 \pm 0.022$ | $0.923 \pm 0.045$ | $4.213 \pm 0.068$ | $0.723 \pm 0.021$ |
| Attentive FP | $0.520 \pm 0.001$ | $0.771 \pm 0.026$ | $4.197 \pm 0.193$ | - |
| PretrainGNN | $0.545 \pm 0.003$ | $1.210 \pm 0.005$ | $6.392 \pm 0.003$ | - |
| GROVER_large | $0.676 \pm 0.012$ | $0.798 \pm 0.018$ | $5.162 \pm 0.047$ | - |
| ChemBERTa-2* | $0.639 \pm 0.006$ | $0.795 \pm 0.033$ | - | $1.858 \pm 0.029$ |
| ChemRL-GEM | $0.486 \pm 0.008$ | $0.706 \pm 0.061$ | $3.924 \pm 0.436$ | - |
| MolFormer | $0.492 \pm 0.012$ | $0.766 \pm 0.026$ | $5.485 \pm 0.045$ | $1.091 \pm 0.021$ |
| ConfNet | $1.360 \pm 0.038$ | $2.115 \pm 0.484$ | - | $1.329 \pm 0.042$ |
| UniMol | $\mathbf{0.374 \pm 0.012}$ | $0.741 \pm 0.014$ | $2.867 \pm 0.186$ | - |
| SchNet-scalar | $0.704 \pm 0.032$ | $0.672 \pm 0.027$ | $1.608 \pm 0.158$ | $0.723 \pm 0.100$ |
| SchNet-emb | $0.589 \pm 0.022$ | $0.635 \pm 0.057$ | $1.587 \pm 0.136$ | $0.692 \pm 0.028$ |
| ChemProp3D | $0.602 \pm 0.035$ | $0.681 \pm 0.023$ | $2.014 \pm 0.182$ | $0.815 \pm 0.170$ |
| CONAN | $0.556 \pm 0.013$ | $0.571 \pm 0.019$ | $1.496 \pm 0.158$ | $0.635 \pm 0.051$ |
| CONAN-FGW | $0.422 \pm 0.016$ | $0.529 \pm 0.022$ | $1.068 \pm 0.083$ | $0.549 \pm 0.016$ |
| FACET | $0.424 \pm 0.009$ | $\mathbf{0.516 \pm 0.044}$ | $\mathbf{0.967 \pm 0.082}$ | $\mathbf{0.495 \pm 0.034}$ |

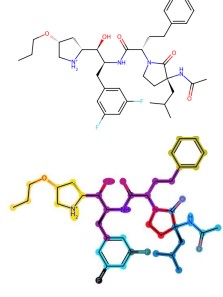

Figure 5: RingsPaths decomposition on BACE, splitting molecules into rings, paths, and linkers. This reflects molecular topology and improves interpretability and generalization.

## 5.4 State-of-the-Art Performance Comparison on Molecular Tasks

**Datasets.** We evaluate molecular property regression on the **MoleculeNet** (Wu et al., 2018) and **MARCEL** (Zhu et al., 2024a) benchmarks. **MoleculeNet** includes four datasets, **ESOL**, **BACE**, **Lipo**, and **FreeSolv**, with targets covering solubility, inhibitory concentration ($pIC_{50}$), lipophilicity, and hydration free energy. **MARCEL** consists of **Drugs-75K** and **Kraken**, where the goal is to predict the Boltzmann-averaged property $\langle y \rangle_{k_{\mathcal{B}}}$ from sampled conformers. **Drugs-75K** uses quantum descriptors (IP, EA, $\chi$), while **Kraken** focuses on Sterimol features ($B_5$, L, and their buried forms). The Boltzmann average is computed as a weighted sum over conformer-specific values $y_i$ with probabilities $p_i$. All datasets follow the original random split settings, using the provided sampled conformers.

**Baselines.** For the **MoleculeNet** benchmark (Wu et al., 2018), we compare FACET with a wide range of baselines, including (i) `2D supervised` methods (e.g., GAT (Veličković et al., 2018), D-MPNN (Yang et al., 2019a), AttentiveFP (Xiong et al., 2019)), (ii) `pre-training approaches` (e.g., PretrainGNN (Hu et al., 2020b), GROVER (Rong et al., 2020), ChemBERTa-2* (Ahmad et al., 2022), ChemRL-GEM (Fang et al., 2022), MolFormer (Ross et al., 2022)), (iii) `3D-conformers` based models (ConfNet (Liu et al., 2021), UniMol (Zhou et al., 2023), SchNet (Schütt et al., 2017), ChemProp3D (Axelrod & Gómez-Bombarelli, 2023),CONAN-FGW (Nguyen et al., 2024b)). Training follows the setup in CONAN-FGW (Nguyen et al., 2024b).

For the **MARCEL** benchmark (Zhu et al., 2024a), we compare FACET against `2D models` (e.g., GIN (Xu et al., 2019), GIN+VN (Hu et al., 2020a), ChemProp (Yang et al., 2019b), GraphGPS (Rampášek et al., 2022)), `3D models` (e.g., SchNet (Schütt et al., 2017), DimeNet++ (Klicpera et al., 2020), GemNet (Gasteiger et al., 2021), PaiNN (Schütt et al., 2021), ClofNet (Du et al., 2022), LEFTNet (Du et al., 2023)), and `ensemble strategies` such as DeepSets-based ensemble (Zaheer et al., 2017), self-attention (Vaswani et al., 2017), etc. All methods are evaluated under the same settings as described in the MARCEL benchmark.

### 5.4.1 Results

**MoleculeNet**. As shown in Table 3, FACET achieves state-of-the-art performance on three molecular property regression tasks (ESOL, FreeSolv, BACE), with the lowest MSEs: $0.516 \pm 0.044$, $0.967 \pm 0.082$, and $0.495 \pm 0.115$, respectively. Its consistent gains over ConAN-FGW indicate that, beyond geometry-aware aggregation, FACET's use of fragment substructures (Figure 5) enhances attention to localized chemical contexts. This demonstrates the advantage of combining 3D spatial information with chemically meaningful substructures for molecular property prediction.

**MARCEL.** In Table 4, we evaluate FACET on two backbones, SchNet and GemNet. FACET consistently boosts both, confirming the benefits of structure-aware aggregation and fragment-level hierarchy. Unlike ConAN-FGW, which struggles to scale on the large MARCEL benchmark, FACET remains efficient and achieves near-SOTA performance across all targets, demonstrating robust effectiveness in diverse molecular property prediction tasks.

**Additional Analysis.** In this section, we analyze the key components of FACET through ablation studies. Specifically, we evaluate the impact of: (i) removing fragment structures from

Table 5: FACET ablation study. *"w/o Frag."* means without using the fragment graph, *"w/o Frag. in Trans."* indicates without using the fragment graph in the graph transformer, and *"w/o Adapt."* depicts not using adaptors to adjust features from 3D-GNN and 2D-GNN.

| Dataset | FACET | w/o Frag. | w/o Frag. in Trans. | w/o Adap. |
|---|---|---|---|---|
| ESOL | **0.516** | 0.531 | 0.525 | 0.546 |
| FreeSolv | **0.967** | 1.072 | 0.973 | 1.085 |
| Kraken | **0.238** | 0.247 | 0.242 | 0.262 |

Table 6: Performance (MSE) of different training strategies. *"FACET (default)"* refers to training three steps with ablation studies for merging, and *"FACET (w/o FGW)"* refers to a version without using FGW to supervise the graph transformer.

| Settings | ESOL($\downarrow$) | FreeSolv($\downarrow$) | BACE($\downarrow$) | Lipo($\downarrow$) |
|---|---|---|---|---|
| ConAN-FGW | $0.53 \pm 0.022$ | $1.07 \pm 0.083$ | $0.55 \pm 0.016$ | **$0.42 \pm 0.016$** |
| FACET (default) | $0.52 \pm 0.044$ | $0.97 \pm 0.082$ | **$0.50 \pm 0.115$** | $0.42 \pm 0.009$ |
| - Merge all steps: | $0.57 \pm 0.023$ | $1.26 \pm 0.094$ | $0.59 \pm 0.062$ | $0.53 \pm 0.013$ |
| - Merge steps 2-3: | **$0.51 \pm 0.014$** | **$0.87 \pm 0.102$** | $0.50 \pm 0.035$ | $0.44 \pm 0.014$ |
| FACET (w/o FGW) | $0.54 \pm 0.053$ | $0.98 \pm 0.007$ | $0.53 \pm 0.024$ | $0.45 \pm 0.080$ |

both the 2D MPNN and the self-attention mechanism in the graph transformer (**w/o Frag**); (ii) using fragments only in the 2D MPNN but not in the graph transformer (**w/o Frag in Trans.**); and (iii) omitting the trainable adaptor (**w/o Adap.**) that aligns 3D conformer features with the graph transformer, which can lead to performance degradation due to domain shift during training. Furthermore, we also evaluate training strategies that (iv) **merge all stages** into a unique step, (v) retain stage 1, but **merge steps 2-3**, and finally (vi) **FACET (w/o FGW)** means without supervised Graph Transformer with FGW.

As shown in Table 5, the absence of (i) significantly reduces performance, making FACET comparable to ConAN-FGW but with better scalability. Incorporating fragments into both components (ii) provides further gains, while (iii) proves essential for mitigating the domain shift introduced by changes in the 3D MPNN during training. The Table 6 presents results for settings (iv)-(vi), showing that learning a geometry-aware module explicitly regularized by FGW is important, which cannot be replaced by downstream loss alone. In Table 7, we further present FACET's model parameters compared with other GNN

Table 4: Comparison of molecular property regression performance on the **MARCEL** benchmark (MAE $\downarrow$). The results of competing methods are adapted from Zhu et al. (2024a).

| Category | Model | Drugs-75K | | | Kraken | | | |
|---|---|---|---|---|---|---|---|---|
| | | IP | EA | $\chi$ | $B_5$ | L | BurB$_5$ | BurL |
| 2D models | GIN | 0.4354 | 0.4169 | 0.2260 | 0.3128 | 0.4003 | 0.1719 | 0.1200 |
| | GIN+VN | 0.4361 | 0.4169 | 0.2267 | 0.3567 | 0.4344 | 0.2422 | 0.1741 |
| | ChemProp | 0.4595 | 0.4417 | 0.2441 | 0.4850 | 0.5452 | 0.3002 | 0.1948 |
| | GraphGPS | 0.4351 | 0.4085 | 0.2212 | 0.3450 | 0.4363 | 0.2066 | 0.1500 |
| 3D models | SchNet | 0.4394 | 0.4207 | 0.2243 | 0.3293 | 0.5458 | 0.2295 | 0.1861 |
| | DimeNet++ | 0.4441 | 0.4233 | 0.2436 | 0.3510 | 0.4174 | 0.2097 | 0.1526 |
| | GemNet | 0.4069 | 0.3922 | **0.1970** | 0.2789 | 0.3754 | 0.1782 | 0.1635 |
| | PaiNN | 0.4505 | 0.4495 | 0.2324 | 0.3443 | 0.4471 | 0.2395 | 0.1673 |
| | ClofNet | 0.4393 | 0.4251 | 0.2378 | 0.4873 | 0.6417 | 0.2884 | 0.2529 |
| | LEFTNet | 0.4174 | 0.3964 | 0.2083 | 0.3072 | 0.4493 | 0.2176 | 0.1486 |
| Ensemble Strategy with DeepSets | SchNet | 0.4452 | 0.4232 | 0.2243 | 0.2704 | 0.4322 | 0.2024 | 0.1443 |
| | DimeNet++ | 0.4126 | 0.3944 | 0.2267 | 0.2630 | 0.3468 | 0.1783 | 0.1185 |
| | GemNet | 0.4066 | 0.3910 | 0.2027 | 0.2313 | **0.3386** | 0.1589 | **0.0947** |
| | PaiNN | 0.4466 | 0.4269 | 0.2294 | **0.2225** | 0.3619 | 0.1693 | 0.1324 |
| | ClofNet | 0.4280 | 0.4033 | 0.2199 | 0.3228 | 0.4485 | 0.2178 | 0.1548 |
| | LEFTNet | 0.4149 | 0.3953 | 0.2069 | 0.2644 | 0.3643 | 0.2017 | 0.1386 |
| FACET | SchNet | 0.4235 | 0.3971 | 0.2155 | 0.2508 | 0.3982 | 0.1803 | 0.1245 |
| | GemNet | **0.3891** | **0.3852** | **0.1970** | **0.2225** | 0.3402 | **0.1503** | 0.0952 |

baselines, indicating a balance between model size and performance that matches or outperforms much larger 3D-GNNs. More details on training time are discussed in Table 10 Appendix.

## 6 CONCLUSION

We introduce FACET, a scalable framework that replaces costly FGW alignment with a Graph Transformer trained to approximate FGW fusion between 2D fragments and 3D conformers. This approximation enables efficient, end-to-end fusion of 2D and 3D structure, yielding strong gains across MoleculeNet and state-of-the-art performance on the large-scale MARCEL benchmark.

Table 7: FACET vs 3D-GNN ensemble baselines on model size, inference time, Mean Absolute Error (MAE) $\downarrow$.

| Model | Kraken (BurL) | | | Drugs (ip) | | |
|---|---|---|---|---|---|---|
| | Param | Run. time (s) | MAE | Param | Run. time (s) | MAE |
| SchNet | 215215 | 1.33 | 0.1443 | 210607 | 64.45 | 0.4452 |
| PaiNN | 1310209 | 2.36 | 0.1324 | 1305601 | 80.44 | 0.4466 |
| ClofNet | 605122 | 2.02 | 0.1548 | 600514 | 88.48 | 0.4280 |
| LEFTNet | 2722724 | 6.49 | 0.1386 | 2718116 | 138.28 | 0.4149 |
| FACET | 584065 | 3.17 | 0.1245 | 584065 | 130.68 | 0.4235 |

While FACET performs well on small, drug-like molecules, its evaluation is limited to standard benchmarks, leaving open questions about generalization to more complex regimes, including **biomacromolecules** with long-range dependencies, **polymers and materials** without stable conformers, and multi-molecular systems such as protein–ligand interactions. Future directions include (i) attention mechanisms capturing both local fragment-level and long-range structure, (ii) extensions to flexible input formats (e.g., voxel grids or material-specific graphs), and (iii) cross-graph or co-embedding strategies for intermolecular modeling. Broader evaluation on datasets such as PDBbind (Liu et al., 2015) and PolyInfo (Otsuka et al., 2011) in future work would further assess FACET's applicability.

## ACKNOWLEDGMENT

This work was partially performed under the auspices of the U.S. Department of Energy by Lawrence Livermore National Laboratory under contract DE-AC52-07NA27344. Lawrence Livermore National Security, LLC. This work was funded by the Defense Threat Reduction Agency (DTRA), HDTRA1242044 (HK) and HDTRA1036045 (JA). The project was also supported by Deutsche Forschungsgemeinschaft (DFG, German Research Foundation) under Germany's Excellence Strategy - EXC 2075 – 390740016, the DARPA ANSR program under award FA8750-23-2-0004, the DARPA CODORD program under award HR00112590089. The authors thank the International Max Planck Research School for Intelligent Systems (IMPRS-IS) for supporting Duy M. H. Nguyen. Duy M. H. Nguyen, Trung Q. Nguyen, Mai Thanh Nhat Truong, and Daniel Sonntag are also supported by the No-IDLE project (BMBF, 01IW23002), the MASTER project (EU, 101093079), and the Endowed Chair of Applied Artificial Intelligence, Oldenburg University.

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

SUPPLEMENTARY MATERIAL FOR
"FACET: A FRAGMENT-AWARE CONFORMER ENSEMBLE
TRANSFORMER"

## CONTENTS

## A  IMPLEMENTATION DETAILS

Our training pipeline includes three stages: In the first stage, we train only the 2D and 3D MPNNs to learn corresponding features from 2D molecular graph and 3D conformers. The features in this stage also serve as a dataset for approximating Graph Transformer to the FGW distance. In the next stage, the Graph Transformer is trained separately to simulate the costly computation of FGW distance between two conformers using learned features from stage 1. In the last stage, Graph Transformer is integrated in a single end-to-end training with 2D and 3D MPNNs. At this stage, only 2D and 3D MPNNs are trained. As a result of changing MPNNs during the last stage, a shift in the distribution of the Graph Transformer input might occur. We solve this problem by

adding an adaptor layer using an MLP on both 3D and 2D features before feeding them to the GraphTransformer. For all experiments on the **MoleculeNet** and **MARCEL** benchmarks, we use the same number of conformers as specified in their original settings.

In all stages, we use Adam as our optimizer. We train our model on an 8 V100-GPUs cluster.

**Stage 1. Learning 2D and 3D features.** For each molecule, we define by $\mathbf{H}_{2d-3d} = \widetilde{\mathbf{W}}_{2D}\mathbf{H}_{2D} + \widetilde{\mathbf{W}}_{3D}\mathbf{H}_{3D}$, we then train for 150 epochs and set the learning rate to $1e^{-3}$. to optimize target property tasks $\mathcal{L}_{\text{pred}} = ||\widehat{\boldsymbol{y}}_{2d-3d} - \widetilde{\mathbf{y}}||_2^2$ where $\widetilde{\mathbf{y}}$ be the ground-truth value and $\widehat{\boldsymbol{y}}$ be our predicted value defined by:

$$\widehat{\boldsymbol{y}}_{2d-3d} = \mathbf{W}^{\mathcal{G}}\left(\frac{1}{K}\sum_{k=1}^{K}\mathbf{H}_{2d-3d}[k]\right) + \mathbf{b}^{\mathcal{G}}, \tag{10}$$

with $\mathbf{W}^{\mathcal{G}}$ and $\mathbf{b}^{\mathcal{G}}$ are learnable parameters and $K$ is number of conformers.

**Stage 2. Training Graph Transformer to approximate FGW distance.** The Graph Transformer is trained separately in the second stage to approximate the FGW distance by Euclidean embedding space. For the Graph Transformer architecture, we employ the same setting as Graphormer from Ying et al. (2021). Specifically, a number of attention layers, a number of attention heads, and the hidden dimension of the transformer are set to 12, 8, and 64, respectively, which makes the total number of parameters of the Graph Transformer 372k. In our attention, we use the shortest-path distance (SPD) between a pair of nodes. Following practical implementation in Ying et al. (2021), we pre-compute SPD distance for each 3D molecule graph and load these values during training and inference. We set a learning rate of $1e^{-5}$ and train for 1000 epochs with the following loss function:

$$\mathcal{L}_{\text{enc}} = \sum_{ij}\left[||\mathcal{T}_{\theta}(\mathbf{H}_i) - \mathcal{T}_{\theta}(\mathbf{H}_j)||_2^2 - \text{FGW}_{p,\alpha}(\mathcal{G}(S_i), \mathcal{G}(S_j))\right]. \tag{11}$$

**Stage 3. Training Fragment-aware Graph Transformer.** In the final stage, we freeze the trained GraphTransformer $\mathcal{T}\theta(\cdot)$ and use it to compute aggregated features from 3D conformer embeddings generated by the 3D-MPNN. To accommodate potential distribution shifts, we add lightweight FFN adaptor layers on top of both the 2D- and 3D-MPNNs used in $\mathcal{T}\theta(\cdot)$, while continuing to update the MPNNs during training. The full model is trained for 300 epochs with a reduced learning rate to optimize the training loss $\mathcal{L}_{\text{pred}} = ||\widehat{\boldsymbol{y}} - \widetilde{\mathbf{y}}||_2^2$ where

$$\widehat{\boldsymbol{y}} = \mathbf{W}^{\mathcal{G}}\left(\frac{1}{K}\sum_{k=1}^{K}\mathbf{H}_{\text{comb}}[k]\right) + \mathbf{b}^{\mathcal{G}}. \tag{12}$$

$\mathbf{H}_{\text{comb}}$ is final atom-wise embedding.

# B  FURTHER VISUALIZATION FRAGMENT OUTPUTS

**Fragment Generation Algorithms.** We use a structural fragmentation method based on Ring-Path algorithms (Kong et al., 2022; Geng et al., 2023; Wollschläger et al., 2024) that decompose a molecular graph $G = (V, E)$, where $V$ denotes atoms and $E$ denotes covalent bonds, into a set of chemically interpretable fragments. The fragmentation process identifies a set of *ring fragments* $\mathcal{F}_{\text{ring}} \subseteq \mathcal{F}$ using RDKit's cycle basis algorithm (SSSR), where each ring $f_r \in \mathcal{F}_{\text{ring}}$ is encoded by its atom indices and size class.

Next, all bonds not part of any ring are grouped into *acyclic path fragments* $\mathcal{F}_{\text{path}} \subseteq \mathcal{F}$, where each $f_p \in \mathcal{F}_{\text{path}}$ is a linear chain of nodes, extracted via depth-first search under a degree constraint. Each fragment $f \in \mathcal{F} = \mathcal{F}_{\text{ring}} \cup \mathcal{F}_{\text{path}} \cup \mathcal{F}_{\text{junction}}$ is assigned a type $t(f) \in \{0, 1, 2\}$ (representing ring, path, or junction) and a *type index* $\phi(f) \in \{0, 1, \ldots, K-1\}$ within a fixed vocabulary of size $K$. Fragments whose sizes exceed a predefined threshold $k_{\max}$ are mapped to the final index of their category to preserve bounded dimensionality.

We define a *fragment-atom incidence matrix* $M \in \{0, 1\}^{|V| \times |\mathcal{F}|}$, where $M_{v,f} = 1$ if atom $v \in V$ belongs to fragment $f$. From this, we derive a fragment-level graph $G^{\text{frag}} = (V_{\text{frag}}, E_{\text{frag}})$, where each node $f \in \mathcal{F}$ represents a molecular fragment and an edge $(f_i, f_j) \in E_{\text{frag}}$ is added if two fragments share at least one atom or are directly bonded.

Compared to traditional fragmentation algorithms like BRICS (Degen et al., 2008), BBB (Sommer et al., 2023), or MagNet (Hetzel et al., 2023), the RingPath algorithm offers a more topology-aware decomposition by explicitly capturing key structural motifs such as rings, paths, and linkers. While BRICS and BBB often generate chemically meaningful fragments based on retrosynthetic rules, they may overlook contextual connectivity critical for graph-based learning. In contrast, RingPath preserves the relational structure between fragments, aligning closely with how molecules are built and understood in topological space—making it particularly beneficial for tasks requiring structural interpretability and generalization in graph neural networks. The advantages of RingPath have also been empirically validated in recent studies, demonstrating improved performance across various molecular property prediction benchmarks.

**Visualization of Typical Extracted Fragment Graphs.** Figures 6 and 7 illustrate representative examples of fragment extraction using the RingPath algorithm on the Kraken and Drug-75k datasets. The top row displays the original 2D molecular structures, while the bottom row shows the corresponding RingPath decompositions. Each colored region highlights a distinct structural fragment, such as a ring or path, demonstrating the algorithm's ability to segment complex molecules into chemically meaningful and interpretable components.

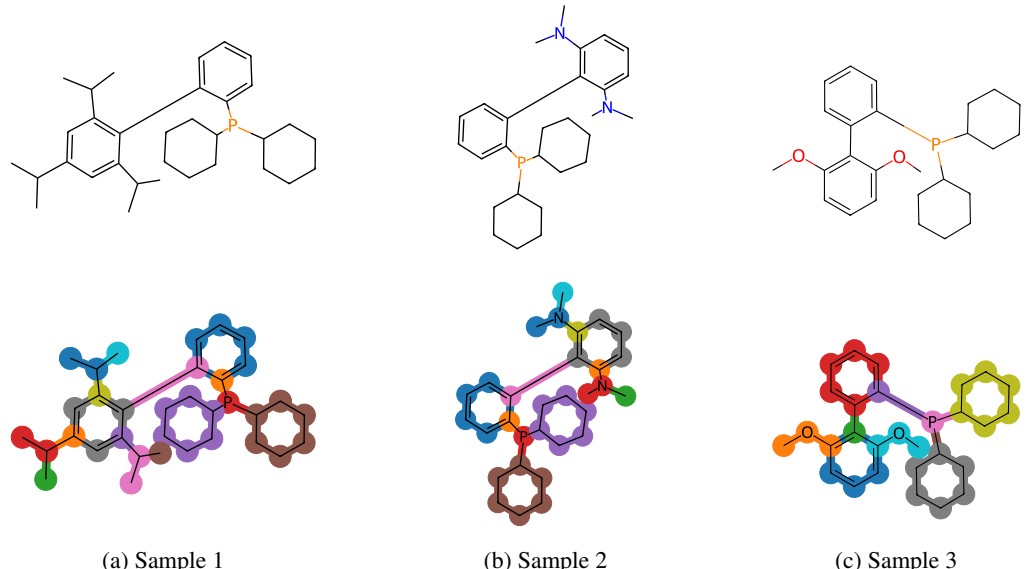

(a) Sample 1       (b) Sample 2       (c) Sample 3

Figure 6: RingsPaths decomposition on three samples of the **Kraken** dataset. Top: 2D molecules; bottom: corresponding RingsPaths decomposition results.

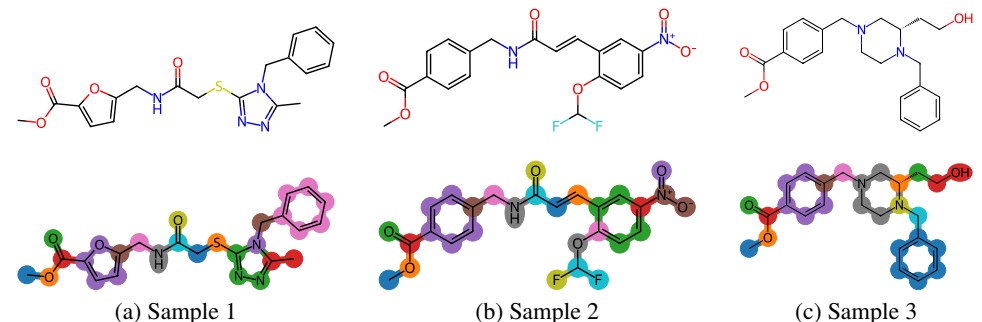

(a) Sample 1       (b) Sample 2       (c) Sample 3

Figure 7: RingsPaths decomposition on three samples of the **Drugs-75K** dataset. Top: 2D molecules; bottom: corresponding RingsPaths decomposition results.

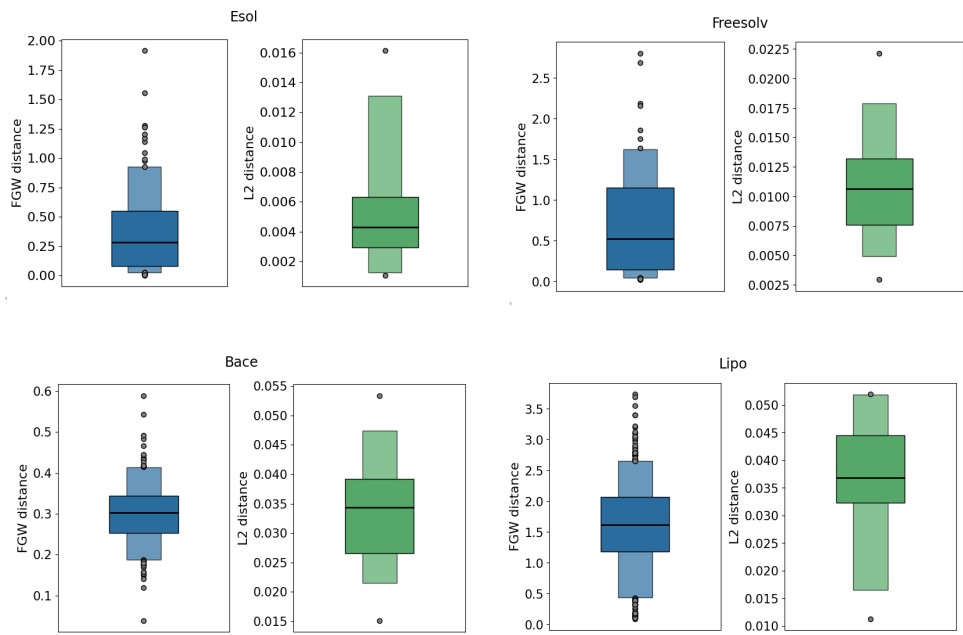

Figure 8: **Boxplots illustrating conformer diversity across the four datasets**. For each dataset, two boxplots are shown. **Left** (blue): distribution of molecules based on the average FGW distance between their conformers, reflecting conformer diversity. **Right** (green): distribution of the 20 molecules with the lowest conformer diversity, measured by the $L_2$ distance between the mean embedding produced by our Graph Transformer and the nearest conformer embedding.

## C  ANALYSIS OF CONFORMER DIVERSITY

The diversity of conformers plays a crucial role in learning effective molecular representations. When molecules have very similar conformers, the embeddings produced by the Graph Transformer may collapse into a single, overly similar conformer representation. To assess whether this collapse occurs in our model, we conducted both quantitative and qualitative analyses.

Quantitatively, Figure 8 shows two boxplots summarizing conformer diversity for each dataset. For the first boxplot, we computed the average FGW distance between all pairs of conformers for each molecule in the test set. This captures how structurally diverse the conformers are. The results show that most molecules exhibit non-zero average FGW distances, indicating meaningful conformer variation; the Lipo dataset in particular contains molecules with highly diverse conformer sets.

For the second boxplot, we selected the 20 molecules with the lowest conformer diversity based on the FGW distance. For each molecule, we obtained the latent embeddings from our trained Graph Transformer, calculated their mean embedding, and subsequently measured the $L_2$ distance between this mean embedding and the closest conformer-level embedding. It can be seen that even among these least-diverse molecules, the embeddings remain distinct: the average embedding does not collapse into a single conformer representation.

To complement our quantitative analysis and provide a more intuitive view of conformer behavior, we applied t-SNE van der Maaten & Hinton (2008) directly to the embeddings produced by our Graph Transformer. For each molecule, we used both the mean embedding (obtained by averaging its conformer embeddings) and the individual conformer embeddings from each molecule in the test set to map into a 2D domain. Rather than summarizing distances as boxplots, this visualization allows us to inspect how embeddings are arranged in a lower-dimensional space. We randomly selected two molecules with large FGW distances and two random others with small FGW distances and visualized their embeddings in Figure 9. In both cases, high and low conformer diversity, the mean embedding remains well separated from the individual conformer embeddings, and the conformers themselves occupy distinct regions in 2D space, confirming that the model preserves conformer variability without collapsing representations.

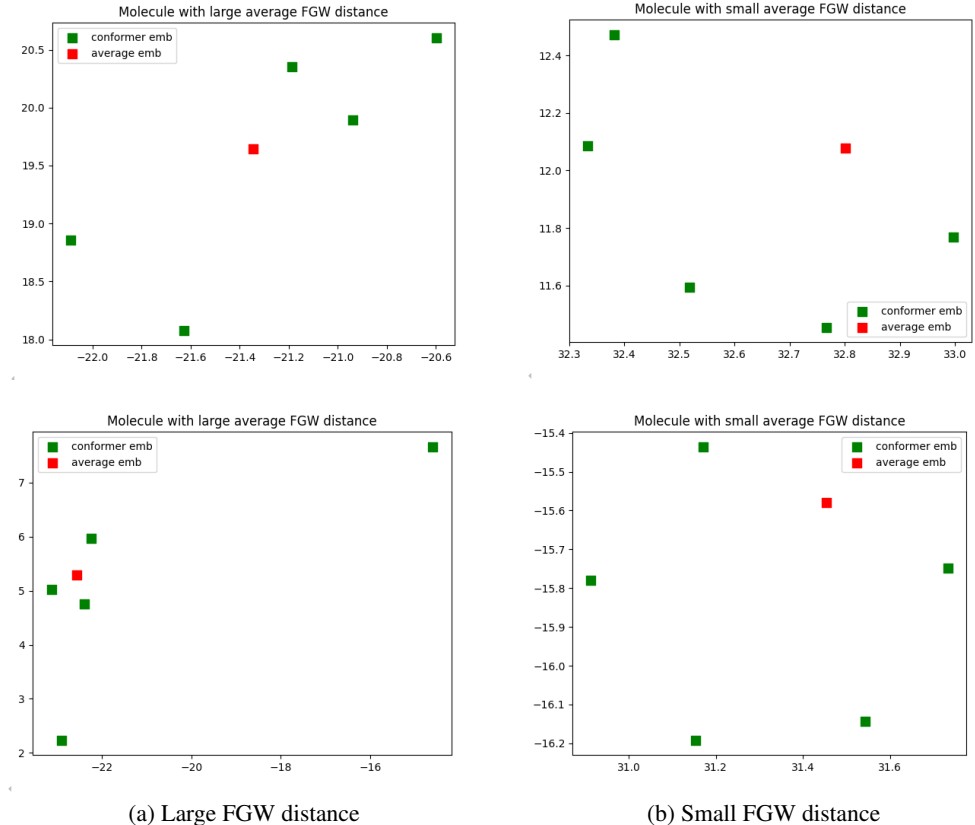

(a) Large FGW distance         (b) Small FGW distance

Figure 9: t-SNE visualization of graph transformer embeddings of four molecules in **FreeSolv**, in which two molecules have a large average FGW distance between their conformers (**left**) and the other two molecules have a small average FGW distance (**right**).

# D ADDITIONAL ANALYSIS OF FACET'S SCALABILITY AND PERFORMANCE WITH MORE 3D CONFORMERS

In this section, we further analyze FACET's scalability on the following two factors:

## D.1 INFERENCE TIME WHEN INCREASING THE NUMBER OF 3D CONFORMERS FOR EACH MOLECULE.

We compare FACET against two versions of CONAN-FGW in running time to extract structure-aware embedding aggregation with different input of 3D conformers. We use two variations of CONAN-FGW, including a single GPU version and another relaxed solver that permits running Sinkhorn iterations on GPUs by matrix multiplication, thus supporting distributed multi-GPUs acceleration. The experiments are conducted on a **single GPU** using a batch size of 32 molecules, each with different conformers ranging from 3, 5, 10, 15, and 20, and another experiment with **four GPUs** on the same batch size, i.e., 8 molecules per GPU.

Figure 10 indicates our observations across four datasets of **MoleculeNet** benchmark, where we report the required time to extract embedding aggregations for all molecules in the test set. We see that (i) **FACET** demonstrates excellent scalability where its runtime remains nearly constant regardless of the number of conformers, both in single-GPU and multi-GPU settings. In contrast, ConAN-FGW shows poor scalability where runtime increases steeply with the number of conformers. While the multi-GPU usage improves runtime over single-GPU, the growth trend remains significant, with runtimes still exceeding 30 seconds at 20 conformers (e.g., with ESOL dataset).

Secondly, the nearly identical runtime of FACET across single- and multi-GPU settings, as shown in the plot, can be attributed to its computational efficiency and the relatively small workload in this experiment. In such cases, the overhead introduced by multi-GPU parallelization - such as inter-GPU communication and data synchronization - can outweigh its potential speedup benefits. Therefore, we argue that multi-GPU acceleration for FACET becomes advantageous only under substantially larger workloads, such as batch processing of thousands to millions of molecules or handling complex input representations that exceed the memory capacity of a single GPU.

## D.2 AVERAGE TRAINING TIME PER EPOCH AS A FUNCTION OF DATASET SIZE.

We analyze the scalability of FACET with respect to the number of training molecules. To this end, we report the average training time per epoch across four datasets from the MoleculeNet benchmark. Figure 11 compares the training time of FACET and ConAN-FGW on a single GPU, using a batch size of 256 and 5 conformers per molecule. As shown in the figure, FACET achieves a 2.28× to 3.17× speedup over ConAN-FGW. Notably, this speedup is roughly proportional to the number of training molecules in each dataset, as reported in Table 1.

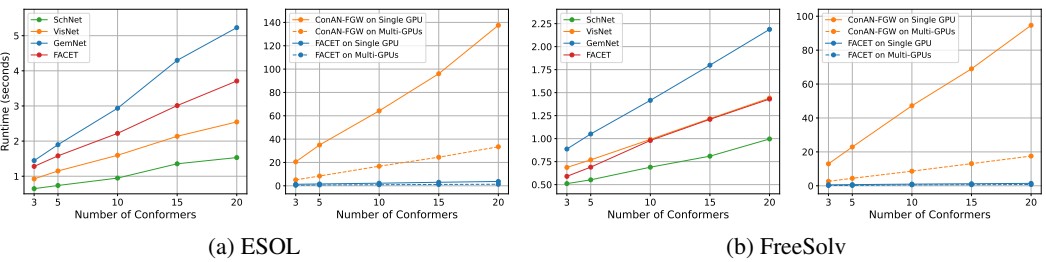

(a) ESOL        (b) FreeSolv

Figure 10: **Inference running** time comparison between **FACET** and other **GNN-based methods** on two datasets, ESOL (left) and FreeSolv (right). Results are shown for both single-GPU and 4-GPU (multi-GPU) configurations. Reported runtimes represent the total time required to extract structural embeddings for all molecules in the test set of each dataset.

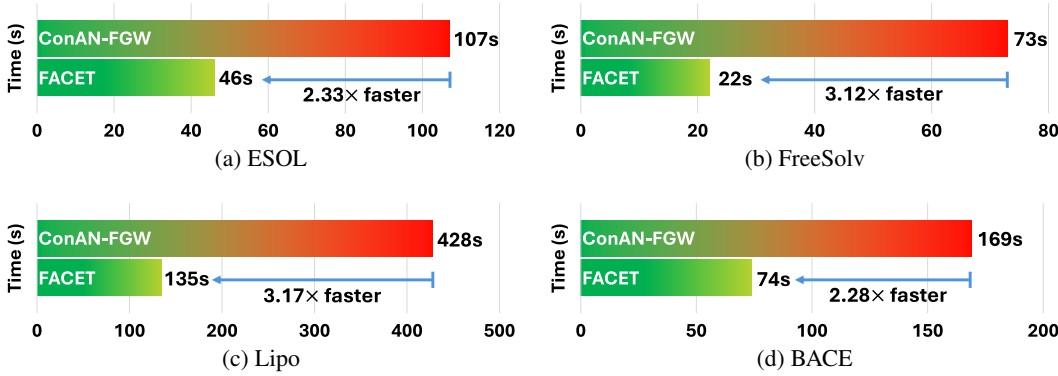

(a) ESOL        (b) FreeSolv

(c) Lipo        (d) BACE

Figure 11: Comparison of the one-epoch training time of CONAN-FGW (Nguyen et al., 2024b) and the proposed FACET on four datasets from the **MoleculeNet** benchmark.

## D.3 ABLATION STUDY ON THE IMPACT OF INCREASING THE NUMBER OF 3D CONFORMERS IN FACET

We provide below a comprehensive ablation study on the impact of using an increasing number of RDKit-generated conformers in a set of 3, 5, 10, 15, 20, 50, 100 across four datasets (ESOL, FreeSolv, BACE, and Lipo). For completeness, we note that using 100 conformers for the BACE and LIPO datasets exceeded GPU memory (OOM) capacity in our setup and therefore could not be evaluated.

Table 8: Comparisons on performance with different numbers of conformers generated by RDKit, "OOM" indicates out-of-memory.

| Settings | 3 conf. | 5 conf. (default) | 10 conf. | 15 conf. | 20 conf. | 50 conf. | 100 conf. |
|---|---|---|---|---|---|---|---|
| ESOL | $0.539 \pm 0.06$ | $0.516 \pm 0.04$ | $0.501 \pm 0.02$ | $0.511 \pm 0.03$ | $0.546 \pm 0.02$ | $0.529 \pm 0.040$ | $0.530 \pm 0.037$ |
| FreeSolv | $0.977 \pm 0.25$ | $0.967 \pm 0.08$ | $0.933 \pm 0.23$ | $0.946 \pm 0.24$ | $0.949 \pm 0.21$ | $0.940 \pm 0.036$ | $0.945 \pm 0.039$ |
| BACE | $0.542 \pm 0.05$ | $0.495 \pm 0.03$ | $0.513 \pm 0.02$ | $0.519 \pm 0.01$ | $0.517 \pm 0.03$ | $0.463 \pm 0.004$ | OOM |
| Lipo | $0.445 \pm 0.02$ | $0.424 \pm 0.01$ | $0.444 \pm 0.02$ | $0.447 \pm 0.08$ | $0.445 \pm 0.01$ | $0.440 \pm 0.010$ | OOM |

Table 9: Comparisons on performance without conformers generated by RDKit.

| Method | ESOL($\downarrow$) | FreeSolv($\downarrow$) | BACE($\downarrow$) | Lipo($\downarrow$) |
|---|---|---|---|---|
| FACET | $0.516 \pm 0.04$ | $0.967 \pm 0.08$ | $0.495 \pm 0.03$ | $0.424 \pm 0.01$ |
| w/o 3D conformers | $0.546 \pm 0.03$ | $1.197 \pm 0.09$ | $0.584 \pm 0.03$ | $0.543 \pm 0.02$ |

As shown in Table 8, we observe a consistent trend across datasets: increasing the number of conformers from 3 to 5 leads to improved regression performance (lower values indicate better results). However, beyond 5 conformers, the performance tends to converge or slightly fluctuate, confirming that our geometry-aware embedding approach using the FGW distance provides stable and reliable approximations. This aligns with the theoretical expectation that the approximation error scales with $O(1/K)$, where $K$ is the number of conformers used.

When 3D conformers generated by RDKit are not used, our FACET model simplifies significantly. In this configuration, the model only receives 2D molecular graphs along with fragment-level information, and key components such as the Graph Transformer are removed. Table 9 presents the performance comparison between the full FACET model and its 2D-only variant across four benchmark datasets:

These results clearly demonstrate that incorporating 3D conformers, even those generated by RDKit, is critical to the expressiveness and performance of FACET. The full model consistently outperforms its 2D-only counterpart, highlighting the importance of 3D geometry in learning accurate molecular representations.

# E   COMPARISON OF TRAINING TIME BETWEEN FACET AND CONAN-FGW

To provide a comprehensive comparison, we conducted additional experiments to compare the training time of FACET and ConAN-FGW, with the addition of SchNet, VisNet, and GemNet, a strong state-of-the-art 3D molecular model, on two benchmark datasets: BACE (1,059 molecules) and LIPO (2,940 molecules). All models were trained for 200 epochs under the same settings. Since both FACET and ConAN-FGW are originally built on the SchNet architecture, which is generally less expressive than GemNet, we also report the performance of FACET when upgraded to use GemNet as its backbone. From the results listed in Table 10, we have the following key observations:

- **FACET vs. ConAN-FGW**: FACET consistently shows reduced training time compared to ConAN-FGW, though the degree of reduction varies by dataset size.
  - **On BACE**: the time savings are marginal due to the additional cost introduced by the Graph Transformer component in FACET, which is trained using the pre-computed FGW distances from the optimal transport solver.
  - **On LIPO**: the training time reduction is more substantial. This is because ConAN-FGW incurs a high computational cost from directly computing FGW distances between sets of 3D conformers in every forward pass. In contrast, FACET leverages pre-learned geometry-aware embeddings, where the corresponding operation reduces to a lightweight matrix multiplication in the Graph Transformer.
- **FACET vs. GemNet**: FACET represents a balanced trade-off between ConAN-FGW and GemNet in terms of training time. Despite using the simpler SchNet backbone, FACET achieves competitive, sometimes better, performance compared to GemNet, thanks to its geometry-aware aggregation via FGW-based embeddings. This efficiency stems from re-

Table 10: Comparisons on performance in terms of MSE(↓) and corresponding training time(↓).

| Model | Metric | BACE | LIPO |
|---|---|---|---|
| GemNet | MSE | 0.51 ± 0.07 | 0.45 ± 0.01 |
| | Time | 2.04 hours | 4.8 hours |
| | Model Param | 1.95M | |
| FACET (GemNet) | MSE | **0.46 ± 0.03** | **0.39 ± 0.02** |
| | Time | 2.47 hours | 6.4 hours |
| | Model Param | 2.25M | |
| SchNet | MSE | 0.64 ± 0.05 | 0.56 ± 0.01 |
| | Time | 1.4 hours | 2.24 hours |
| | Model Param | 273K | |
| FACET (SchNet) | MSE | 0.50 ± 0.03 | 0.42 ± 0.01 |
| | Time | 2.3 hours | 3.16 hours |
| | Model Param | 584K | |
| VisNet | MSE | 0.61 ± 0.15 | 0.55 ± 0.45 |
| | Time | 1.89 hours | 4.27 hours |
| | Model Param | 1.8M | |

placing costly pairwise conformer comparisons with a latent-space transformer that captures 3D geometric information in a more scalable manner.

- **FACET (GemNet) vs. GemNet**: When both models share the same GemNet architecture, FACET outperforms GemNet in terms of predictive accuracy on both datasets. We observe that (i) the additional training time incurred by FACET is relatively modest: approximately +21% on BACE and +33% on LIPO, and (ii) given the performance gains, this extra time remains acceptable in practical scenarios and demonstrates FACET's scalability and effectiveness.

- **FACET** vs. other GNN baselines: Although FACET introduces additional fusion components, the overhead relative to each backbone remains small. The observed increases in end-to-end training time are moderate (e.g., GemNet: 2.04h → 2.47h on BACE; SchNet: 1.4h → 2.3h). Importantly, most of this extra time comes from the separate pre-training of the graph transformer in Step 2, which takes roughly 0.6 - 1 hour. The parameter growth is also limited mostly in the graph transformer module (e.g., GemNet: 1.95M → 2.25M; SchNet: 273K → 584K). In our experiments, these modest increases were consistently accompanied by improved predictive accuracy, suggesting a practical trade-off between accuracy and cost.

# F PERFORMANCE OF FACET AND CONAN-FGW ON MARCEL BENCHMARK

To provide a meaningful comparison, we benchmarked FACET against ConAN-FGW on 10% of the Drug-75k dataset and on the Kraken dataset, which serve as representative subsets. The results (provided below) show that FACET performs competitively or outperforms ConAN-FGW, even under these reduced-scale settings, reinforcing the efficiency and effectiveness of our approach. The results are shown in Tables 12 and 13.

# G COMPARISONS WITH SOTA METHODS IN 2D (OR 3D)

FACET is designed as a modular framework for enhancing molecular property prediction by integrating structure-aware aggregation over multiple conformers. A central strength of this design is that it can be plugged into a variety of existing backbone architectures, whether 2D or 3D, thus offering a complementary mechanism rather than an alternative to these models.

Table 11: Comparisons on performance with different standalone 3D architectures.

| Model | BACE($\downarrow$) | LIPO($\downarrow$) |
|---|---|---|
| SchNet | $0.64 \pm 0.05$ | $0.56 \pm 0.01$ |
| FACET (SchNet) | $0.50 \pm 0.03$ | $0.42 \pm 0.01$ |
| GemNet | $0.51 \pm 0.07$ | $0.45 \pm 0.01$ |
| FACET (GemNet) | $0.46 \pm 0.03$ | $0.39 \pm 0.02$ |
| VisNet | $0.61 \pm 0.15$ | $0.55 \pm 0.45$ |
| FACET (VisNet) | $0.47 \pm 0.01$ | $0.48 \pm 0.01$ |

Table 12: Comparisons of performance between FACET and ConAN-FGW on Kraken

| | L | BurL | B5 | BurB5 |
|---|---|---|---|---|
| ConAN-FGW (SchNet) | 0.397 | 0.117 | 0.272 | 0.195 |
| FACET (SchNet) | 0.398 | 0.125 | 0.251 | 0.180 |

**FACET improves standalone 3D architectures**   We integrated FACET with established 3D models such as SchNet, GemNet, and VisNet, and consistently observed performance improvements across datasets. Table 11 demonstrates that FACET's geometry-aware aggregation over multiple conformers complements even strong 3D baselines, validating its utility beyond what these models achieve on their own.

**FACET enhances simple 2D MPNNs**   We also applied FACET to a lightweight 2D message-passing neural network and found that incorporating FACET's fragment-level structure-aware aggregation significantly improved performance. This result underscores the compatibility of FACET with 2D backbones and its ability to enhance models that do not explicitly process 3D information.

## H   UNIFIED TRAINING PIPELINE

We investigated the performance of the proposed method when combining all training steps into an end-to-end pipeline. Below, we summarize our findings step by step:

- **Step 1 – Pretraining 2D and 3D MPNNs**: As suggested in prior work like ConAN-FGW, we begin by pretraining the 2D and 3D MPNNs independently. This initial phase is critical to ensure that the encoders, especially the 3D MPNN, converge to a stable and meaningful representation before introducing structure-aware aggregation. To test the necessity of this stage, we experimented with a variant where all three stages were co-trained from scratch. The results showed substantially lower performance, confirming that Stage 1 is crucial for learning rich, aligned, and stable representations.

- **Steps 2 and 3 – Co-training Graph Transformer and Downstream Fine-tuning**: While our default setup trains Step 2 (Graph Transformer with FGW supervision) and Step 3 (fine-tuning on molecular properties) sequentially, we explored an alternative setup where both steps are co-trained. To manage the computational cost of FGW supervision, we adopted an alternating strategy: after every five steps of property prediction optimization, we update the Graph Transformer to approximate FGW distances. This reduces the training burden compared to full FGW supervision at every iteration (as in ConAN-FGW).

Table 13: Comparisons of performance between FACET and ConAN-FGW on Drugs-7.5k

| | $\chi$ | IP | EA |
|---|---|---|---|
| ConAN-FGW (SchNet) | 0.374 | 0.541 | 0.587 |
| FACET (SchNet) | 0.365 | 0.535 | 0.552 |

As shown in Table 14, without separately training Step 1, the model got low performance, confirming that this stage helps the model ensure rich, aligned, and stable molecular representations before incorporating more advanced structure awareness. Secondly, on four MoleculeNet datasets, co-training Steps 2 and 3 produced slightly improved performance over the default FACET setup. For example, on ESOL, performance improved from 0.505 to 0.516, and on FreeSolv, from 0.867 to 0.967. This improvement can be attributed to the model's ability to jointly adapt the 2D/3D encoders and the Graph Transformer, leading to more aligned, task-relevant representations. However, there is a trade-off. This co-training strategy comes with an increased training cost, as FGW distances must still be computed periodically. As a result, while training is slower than the default FACET setup, it remains significantly faster than ConAN-FGW, and achieves a strong balance between efficiency and predictive performance, especially on large-scale datasets like Drug-75k.

Table 14: Comparisons of performance (MSE ↓)of different training strategies.

|  | ESOL($\downarrow$) | FreeSolv($\downarrow$) | BACE($\downarrow$) | Lipo($\downarrow$) |
|---|---|---|---|---|
| ConAN-FGW | $0.529 \pm 0.022$ | $1.068 \pm 0.083$ | $0.549 \pm 0.016$ | $\mathbf{0.422 \pm 0.016}$ |
| FACET | $0.516 \pm 0.044$ | $0.967 \pm 0.082$ | $\mathbf{0.495 \pm 0.115}$ | $0.424 \pm 0.009$ |
| FACET (Merge all steps) | $0.567 \pm 0.023$ | $1.264 \pm 0.094$ | $0.591 \pm 0.062$ | $0.530 \pm 0.013$ |
| FACET (Merge steps 2-3) | $\mathbf{0.505 \pm 0.014}$ | $\mathbf{0.867 \pm 0.102}$ | $0.497 \pm 0.035$ | $0.44\ 0\pm 0.014$ |

## I  LIMITATIONS OF FACET

### I.1  FACET OPERATES ON A PREDEFINED SET OF 3D CONFORMERS.

Our method enables efficient geometry-aware aggregation without requiring expensive alignment procedures at inference time. While FACET demonstrates improved performance even with a small subset of conformers, *the quality and representativeness of this subset can still influence downstream predictions*. In particular, if the selected conformers are heavily biased or fail to capture key structural variations, some aspects of molecular flexibility may be underrepresented. Addressing this challenge through better conformer sampling strategies or task-aware selection mechanisms could further enhance model robustness, especially for highly flexible molecules.

**Future direction**: A promising extension would be to develop end-to-end models that can learn to generate conformers dynamically during training, using gradient feedback from downstream prediction losses. Such a differentiable conformer generation module could enable task-aware structural modeling, ensuring that the generated conformers are optimized not just for physical plausibility, but also for relevance to the predictive task at hand.

### I.2  LIMITATIONS IN SCOPE: FOCUS ON SMALL MOLECULES

FACET has primarily been evaluated on standard molecular property prediction benchmarks such as those in MoleculeNet, which consist mostly of small, drug-like molecules. While this setup is well-suited for many pharmacological applications, it limits the assessment of FACET's generalizability to more complex molecular systems. For example, **biomacromolecules** (e.g., peptides, proteins, nucleic acids) exhibit high flexibility, long-range dependencies, and hierarchical organization that are not present in small molecules. **Polymers and materials** often involve much larger structures without well-defined conformers, challenging FACET's reliance on discrete 3D inputs. Additionally, FACET currently models only single-molecule properties and has not been extended to multi-molecular interactions, such as protein-ligand binding.

**Future direction**: To broaden FACET's applicability, several promising future directions can be explored. First, incorporating efficient attention to capture both local fragment-level information and long-range structural dependencies is essential for handling large biomolecules. Second, adapting FACET to support flexible input formats, such as voxel grids or material-specific graphs, would allow it to process polymers and crystalline materials that lack stable conformers. Third, extending FACET to jointly model molecular interactions through cross-graph attention or co-embedding mechanisms could open applications in drug docking and molecular complex prediction. Finally, applying and evaluating FACET on broader datasets, such as PDBbind (Liu et al., 2015), PolyInfo

(Otsuka et al., 2011), or CoRE-MOF 2019 (Chung et al., 2019), would provide a more comprehensive understanding of its strengths and limitations across molecular domains.

## J  PROOF OF THEOREM 1

Recall that we aim to establish the following novel theoretical bounds: Let $\boldsymbol{D}$ denote the pairwise FGW$_{p,\alpha}$ distance matrix, and let $\{\lambda_k, \boldsymbol{v}_k\}_{k=1}^K$ represent the eigendecomposition of the associated criterion matrix $\boldsymbol{F} = -\boldsymbol{C}\boldsymbol{D}\boldsymbol{C}$, where $\boldsymbol{C} = \boldsymbol{I}_K - \frac{1}{K}\boldsymbol{1}_K\boldsymbol{1}_K^\top$ is the centering matrix. The optimal stress value, denoted by $\mathcal{S}^*$, is bounded as follows: $\mathcal{L} \le \mathcal{S}^* \le \mathcal{U}$, where

$$\mathcal{L} := \sum_{k:\lambda_k<0} \lambda_k^2, \quad \mathcal{U} := \sum_{kl}(\Delta g_k + \Delta g_l)^2 + \mathcal{L} + \mathcal{C}, \quad \Delta g_k = \frac{1}{2}\sum_{l:\lambda_l<0} \lambda_l \cdot \boldsymbol{v}_{kl}^2, \ \forall k \in [K].$$

Here, $\boldsymbol{v}_{kl}$ denotes the $l$-th component of the $k$-th eigenvector $\boldsymbol{v}_k$ of $\boldsymbol{F}$, and $\mathcal{C}$ quantifies the approximation error between the empirical barycenter in the Euclidean embedding space and its counterpart in the original space of undirected attributed graphs. This is equivalent to that given $\boldsymbol{e} := \{\boldsymbol{e}_k\}_{k\in[K]} \in \mathbb{R}^{d\times K}$, our objective is to derive lower and upper bounds for the following cumulative stress:

$$\mathcal{S}^* = \min_{\boldsymbol{e}\in\mathbb{R}^{d\times K}} \mathcal{S}(\boldsymbol{e}), \quad \mathcal{S}(\boldsymbol{e}) = \mathcal{S}_1(\boldsymbol{e}) + \mathcal{S}_2(\boldsymbol{e}), \tag{13}$$

$$\mathcal{S}_1^* := \min_{\boldsymbol{e}\in\mathbb{R}^{d\times K}} \mathcal{S}_1(\boldsymbol{e}), \quad \mathcal{S}_1(\boldsymbol{e}) := \sum_{k,l\in[K]} \left( \|\boldsymbol{e}_k - \boldsymbol{e}_l\|_2^2 - D_{kl} \right)^2, \tag{14}$$

$$\mathcal{S}_2^* := \min_{\boldsymbol{e}\in\mathbb{R}^{d\times K}} \mathcal{S}_2(\boldsymbol{e}), \quad \mathcal{S}_2(\boldsymbol{e}) := \sum_{l\in[K]} \left( \|\overline{\boldsymbol{e}}_K - \boldsymbol{e}_l\|_2^2 - \overline{D}_{K,l} \right)^2. \tag{15}$$

To this end, we begin by specifying and formally defining the following important concepts in Appendix J.1.

### J.1  NON-EUCLIDEAN NATURE OF PAIRWISE FGW DISTANCE MATRIX

**Definition 1** (Euclidean Distance Matrix). *A $K \times K$ distance matrix $\boldsymbol{D}$ is said to be* Euclidean *if there exists a set of points $\boldsymbol{e} = \{\boldsymbol{e}_k\}_{k=1}^K$ in some Euclidean space $\mathbb{R}^d$ such that*
$$\forall k, l \in [K], \quad D_{kl} = \|\boldsymbol{e}_k - \boldsymbol{e}_l\|_2^2.$$
*The space of all Euclidean distance matrices (EDM) is denoted by $\mathcal{E}$.*

**Fact 1** (Conditions for Euclidean Distance Matrix, see, *e.g.,* Gower (1985)). *A matrix $\boldsymbol{D}$ is an EDM if and only if it satisfies the following three conditions:*

  (i) *Non-negativity: $D_{kl} \ge 0$ for all $k, l \in [K]$,*

  (ii) *Hollow diagonal: $D_{kk} = 0$ for all $k \in [K]$,*

  (iii) *Positive semidefiniteness: the associated double-centered matrix $\boldsymbol{F} := -\boldsymbol{C}\boldsymbol{D}\boldsymbol{C}$ is positive semidefinite (PSD), where $\boldsymbol{C} = \boldsymbol{I}_K - \frac{1}{K}\boldsymbol{1}_K\boldsymbol{1}_K^\top$ is the centering matrix, and $\boldsymbol{1}_K$ denotes the $K$-dimensional vector of ones.*

Recall that the pairwise FGW distance matrix $\boldsymbol{D}$ for a collection of $K$ distributions is defined entry-wise by $D_{kl} := \text{FGW}_{p,\alpha}(\mathcal{G}(\mathbb{S}_k), \mathcal{G}(S_l))$ for all $k, l \in [K]$, as introduced in Section 3. The following result establishes that this matrix does not correspond to a Euclidean distance matrix:

**Lemma 1** (Non-Euclidean Nature of Pairwise FGW Distance Matrix). *Consider the case where $d_f = \|\cdot\|_2$. Then the FGW distance matrix $\boldsymbol{D}$, whose entries are given by*
$$FGW_{p,\alpha}(\mathcal{G}_1, \mathcal{G}_2) := \min_{\boldsymbol{\pi}\in\Pi(\boldsymbol{\omega}_1,\boldsymbol{\omega}_2)} \langle (1-\alpha)\boldsymbol{M} + \alpha\, \mathsf{L}(\boldsymbol{A}_1, \boldsymbol{A}_2) \otimes \boldsymbol{\pi}, \boldsymbol{\pi} \rangle,$$
*with $\alpha \in [0, 1]$, does not define a Euclidean distance matrix.*

As established in Lemma 1, which is proved in Appendix J.4, the distance FGW$_{p,\alpha}$ is not a Euclidean distance. Therefore, we are interested in quantifying how accurately non-Euclidean distance matrices can be approximated by pairwise distances between learned embeddings. To this end, we analyze the lower and upper bound of the set $\mathcal{S}$ in Appendices J.2 and J.3, respectively.

### J.2   LOWER BOUNDS ON EMBEDDING NON-EUCLIDEAN FGW DISTANCES

We would like to find the lower bound of $\mathcal{S}$. We note that the original formulation is non-convex, making it analytically intractable. Nonetheless, by reparameterizing the objective as a function of the pairwise squared distances $\widehat{D}_{kl} := \|e_k - e_l\|_2^2$ and $\overline{\widehat{D}}_{Kl} := \|\overline{e}_K - e_l\|_2^2$ induced by the embedding, and by incorporating the necessary conditions to ensure that $\widehat{D}$ corresponds to a valid Euclidean distance matrix, the reformulated problem becomes convex for $\mathcal{S}_1$. Note that we can prove that $\mathcal{S}$ has a lower bound at $\widehat{L}^*$, where $\widehat{L}^*$ is a minimizer of $\mathcal{S}_1$, that is,

$$\mathcal{S}^* = \min_{\widehat{D} \in \mathcal{E}} \left[ \mathcal{S}_1(\widehat{D}) + \mathcal{S}_2(\widehat{D}) \right], \quad \mathcal{S}_2(\widehat{D}) := \sum_{l \in [K]} \left( \overline{\widehat{D}}_{Kl} - \overline{D}_{K,l} \right)^2, \tag{16}$$

$$\mathcal{S}_1(\widehat{L}^*) = \min_{\widehat{D} \in \mathcal{E}} \mathcal{S}_1(\widehat{D}), \quad \mathcal{S}_1(\widehat{D}) := \sum_{k,l \in [K]} \left( \widehat{D}_{kl} - D_{kl} \right)^2. \tag{17}$$

Indeed, given the previous reformulation of $\mathcal{S}$, we can establish the following lower bound via Proposition 1. Notably, to simplify the problem, in Proposition 1, we relax the EDM constraint by considering $\mathcal{E}_{\mathcal{L}}$, containing $\mathcal{E}$ by keeping only the PSD property from the EDM definition in Fact 1. We will reintroduce the missing constraints in $\mathcal{E}_{\mathcal{L}}$ and use the solution for the simplified problem to construct an upper bound in Appendix J.3.

**Proposition 1** (Error Lower Bound of $\mathcal{S}^*$). *The lower bound of $\mathcal{S}$ is provided as follows:*

$$\mathcal{S}^* = \min_{\widehat{D} \in \mathcal{E}} \left[ \mathcal{S}_1(\widehat{D}) + \mathcal{S}_2(\widehat{D}) \right] \geq \mathcal{S}_1(\widehat{L}^*) + \mathcal{S}_2(\widehat{L}^*) \geq \mathcal{L}_1 + \mathcal{L}_2 =: \mathcal{L}, \tag{18}$$

$$\mathcal{S}_1(\widehat{L}^*) = \min_{\widehat{D} \in \mathcal{E}_{\mathcal{L}}} \mathcal{S}_1(\widehat{D}) \geq \sum_{k:\lambda_k < 0} \lambda_k^2 =: \mathcal{L}_1, \tag{19}$$

$$\mathcal{S}_2(\widehat{L}^*) = \min_{\widehat{D} \in \mathcal{E}_{\mathcal{L}}} \mathcal{S}_2(\widehat{D}) = 0 =: \mathcal{L}_2. \tag{20}$$

*Here $\mathcal{E}_{\mathcal{L}}$ contains $\mathcal{E}$ by keeping only the PSD property from the EDM definition in Fact 1.*

*Proof of Proposition 1.* Note that if $\mathcal{S}_1$ is minimized at $\widehat{L}^*$, that is,

$$\mathcal{S}_1(\widehat{L}^*) = \min_{\widehat{D} \in \mathcal{E}} \mathcal{S}_1(\widehat{D}), \quad \mathcal{S}_1(\widehat{D}) := \sum_{k,l \in [K]} \left( \widehat{D}_{kl} - D_{kl} \right)^2. \tag{21}$$

We then can find the lower bound of $\mathcal{S}^* = \min_{\widehat{D} \in \mathcal{E}} \left[ \mathcal{S}_1(\widehat{D}) + \mathcal{S}_2(\widehat{D}) \right]$ via the minimizer $\widehat{L}^*$.

Using the definition of Frobenius norm and $\mathcal{E}_{\mathcal{L}}$, we can obtain:

$$\mathcal{S}_1(\widehat{L}^*) := \min_{\widehat{D} \in \mathcal{E}} \mathcal{S}_1(\widehat{D}) \geq \min_{\widehat{D} \in \mathcal{E}_{\mathcal{L}}} \mathcal{S}_1(\widehat{D}), \quad \mathcal{S}_1(\widehat{D}) = \|\widehat{D} - D\|_F^2,$$

We then obtain the following decomposition:

$$\|\widehat{D} - D\|_F^2 = \|A\|_F^2 + \|B\|_F^2, \quad A := C\widehat{D}C - CDC,$$

$$B := \frac{1}{K}O\widehat{D}C + \frac{1}{K}C\widehat{D}O + \frac{1}{K^2}O\widehat{D}O - \left( \frac{1}{K}ODC + \frac{1}{K}CDO + \frac{1}{K^2}ODO \right),$$

where $C = I_K - \frac{1}{K}O$ is the centering matrix and $O = \mathbf{1}_K \mathbf{1}_K^\top$ is the all-ones matrix. Indeed, using the definition of the centering matrix $C = I_K - \frac{1}{K}O$, we have $I_K = C + \frac{1}{K}O$.

$$\|\widehat{D} - D\|_F^2 = \|I_K \widehat{D} I_K - I_K D I_K\|_F^2 = \|A + B\|_F^2 = \|A\|_F^2 + \|B\|_F^2 + 2\operatorname{Tr}(AB) = \|A\|_F^2 + \|B\|_F^2,$$

Here we used the fact that the matrix product is invariant under cyclic permutation:

$$\operatorname{Tr}(AB) = \operatorname{Tr}\left( C(\widehat{D} - D)C(\widehat{D} - D)\frac{1}{K}O \right) = \operatorname{Tr}\left( \frac{1}{K}OC(\widehat{D} - D)C(\widehat{D} - D) \right) = 0,$$

and

$$\frac{1}{K}OC = \frac{1}{K}O \left( I_K - \frac{1}{K}O \right) = \frac{1}{K}O - \frac{1}{K^2}OO = 0.$$

Under only the PSD constraint, the optimal solution $\widehat{L}^*$ that minimizes $\mathcal{S}_1(\widehat{D})$ can be decomposed as:

$$\widehat{L}^* = \widehat{L}_A^* + \widehat{L}_B^*,$$

where $\widehat{L}_A^*$ and $\widehat{L}_B^*$ respectively minimize the terms $\|A\|_F^2$ and $\|B\|_F^2$ independently.

In particular, using the definition of the centering matrix $C = I_K - \frac{1}{K}O$, the entries of $\widehat{L}_B^*$ are given by:

$$\widehat{L}_{B,kl}^* := \left[\frac{1}{K}ODC + \frac{1}{K}CDO + \frac{1}{K^2}ODO\right]_{kl}$$

$$= \left[\frac{1}{K}OD + \frac{1}{K}(OD)^\top - \frac{1}{K^2}ODO\right]_{kl} = \overline{D}_k + \overline{D}_l - \overline{D},$$

where $\overline{D}_k$ denotes the mean of the $k$-th row (or column) of $D$, and $\overline{D}$ is the global mean of all elements in $D$. Therefore, the rows/columns mean of $\widehat{L}_B^*$ equal those of $D$ itself, and hence

$$\widehat{L}_B^* = \arg\min_{\widehat{D}\in\mathcal{E}_\mathcal{L}} \|B\|_F^2, \quad \min_{\widehat{D}\in\mathcal{E}_\mathcal{L}} \|B\|_F^2 = 0.$$

Therefore,

$$\min_{\widehat{D}\in\mathcal{E}_\mathcal{L}} \mathcal{S}_2(\widehat{D}) = \min_{\widehat{D}\in\mathcal{E}_\mathcal{L}} \sum_{l\in[K]} \left(\overline{\widehat{D}}_{Kl} - \overline{D}_{K,l}\right)^2 = 0.$$

Here we used the fact that the matrix $D$ is given by $D_{kl} := \text{FGW}_{p,\alpha}(\mathcal{G}(\mathbb{S}_k), \mathcal{G}(S_l))$ for all $k, l \in [K]$ and the empirical FGW barycenter is given by

$$\overline{\mathcal{G}}_K \in \arg\min_{\mathcal{G}\in\mathcal{P}_p(\mathbf{\Omega})} \frac{1}{K} \sum_{l=1}^K \text{FGW}_{p,\alpha}^p(\mathcal{G}, \mathcal{G}(S_l)) = \arg\min_{\mathcal{G}\in\mathcal{P}_p(\mathbf{\Omega})} \frac{1}{K} \sum_{l=1}^K \text{FGW}_{p,\alpha}(\mathcal{G}, \mathcal{G}(S_l)),$$

$$\overline{D}_{K,l} := \text{FGW}_{p,\alpha}(\overline{\mathcal{G}}_K, \mathcal{G}(S_l)) = \min_{\mathcal{G}\in\mathcal{P}_p(\mathbf{\Omega})} \frac{1}{K} \sum_{l=1}^K \text{FGW}_{p,\alpha}(\mathcal{G}, \mathcal{G}(S_l)) \ (=: \text{column } l\text{-th means of } D),$$

where $\mathcal{P}_p(\mathbf{\Omega})$ denotes the space of attributed graphs with finite $p$-th order FGW distance. To approximate this barycenter in embedding space, we require

$$\|\overline{e}_K - e_l\|_2^2 \approx \text{FGW}_{p,\alpha}(\overline{\mathcal{G}}_K, \mathcal{G}(S_l)) := \overline{D}_{K,l} \text{ for all } l \in [K],$$

where $\overline{e}_K = \frac{1}{K} \sum_{k=1}^K e_k$ is the mean embedding and $e_k := \mathcal{T}_\theta(\mathbf{H}_k)$ is the learned representation.

Now we would like to find a local analytic solution $\widehat{L}_A^*$ minimizing $\|A\|_F^2$ such that the global solution $\widehat{L}^* = \widehat{L}_A^* + \widehat{L}_B^*$ minimizes both terms $\|A\|_F^2$ and $\|B\|_F^2$ simultaneously. That is,

$$\min_{\widehat{D}\in\mathcal{E}_\mathcal{L}} \|A\|_F^2 = \min_{\widehat{D}\in\mathcal{E}_\mathcal{L}} \|C(\widehat{L}_A + \widehat{L}_B)C - CDC\|_F^2$$

$$= \|C(\widehat{L}_A^* + \widehat{L}_B^*)C - CDC\|_F^2 = \|C\widehat{L}_A^*C - CDC\|_F^2.$$

Here we used the fact that by definition of $\widehat{L}_B^*$, it holds that $C\widehat{L}_B^*C = 0$. Hence, the optimization becomes:

$$\min_{\widehat{D}\in\mathcal{E}_\mathcal{L}} \|C\widehat{L}_A C - CDC\|_F^2.$$

This is in fact the problem of computing the nearest PSD approximation $C\widehat{L}_A C$ to a symmetric matrix $CDC$. Using the result from Higham (1988), we find the analytic solution as follows:

$$\widehat{L}_A^* = -\sum_{k:\lambda_k>0} \lambda_k v_k v_k^\top. \tag{22}$$

Here $\{\lambda_k, v_k\}_{k\in[K]}$ are the eigenvalues and eigenvectors of $F = -CDC$. Because $CDC$ has rows/columns means 0, the ones vector $\mathbf{1}_K$ is an eigenvector of $CDC$ with eigenvalue 0. This leads to $\mathbf{1}_K$ is also in the null space $\widehat{L}_A^*$ and:

$$\widehat{L}_A^* = C\widehat{L}_A^*C, \quad \frac{1}{K}O\widehat{L}_A^* = \frac{1}{K}\left(O\widehat{L}_A^*\right)^\top = 0.$$

Therefore,

$$\|\widehat{L}^* - D\|_F^2 = \|\widehat{L}_A^* + \widehat{L}_B^* - D\|_F^2 = \sum_{k:\lambda_k<0} \lambda_k^2.$$

Combining all together, Proposition 1 is derived as follows:

$$\mathcal{S}^* \geq \min_{\widehat{D}\in\mathcal{E}_\mathcal{L}} \|A\|_F^2 + \min_{\widehat{D}\in\mathcal{E}_\mathcal{L}} \|B\|_F^2 + \min_{\widehat{D}\in\mathcal{E}_\mathcal{L}} \mathcal{S}_2(\widehat{D}) = \sum_{k:\lambda_k<0} \lambda_k^2 + 0 + 0 = \sum_{k:\lambda_k<0} \lambda_k^2 =: \mathcal{L}.$$

$\square$

### J.3    Upper Bounds on Embedding of Pairwise Empirical FGW Barycenter Distances

As discussed in Appendix J.2, the lower bound stated in Proposition 1 is derived by simplifying the problem and relaxing the EDM constraint. Specifically, this relaxation involves considering the set $\mathcal{E}_{\mathcal{L}}$, which contains $\mathcal{E}$ but retains only the PSD requirement from the EDM characterization given in Fact 1. In Proposition 2, we reintroduce the missing constraints excluded in $\mathcal{E}_{\mathcal{L}}$ and leverage the closed-form solution obtained from the relaxed problem to construct an upper bound under the original EDM constraint set $\mathcal{E}$.

**Proposition 2** (Error Upper Bound of $\mathcal{S}^*$). *There exists a matrix $\widehat{\boldsymbol{U}}^* \in \mathcal{E}$ such that the following upper bounds hold:*

$$\mathcal{S}^* = \min_{\widehat{\boldsymbol{D}} \in \mathcal{E}} \left[ \mathcal{S}_1(\widehat{\boldsymbol{D}}) + \mathcal{S}_2(\widehat{\boldsymbol{D}}) \right] \leq \mathcal{S}_1(\widehat{\boldsymbol{U}}^*) + \mathcal{S}_2(\widehat{\boldsymbol{U}}^*) \leq \mathcal{U}_1 + \mathcal{U}_2 =: \mathcal{U}, \tag{23}$$

$$\mathcal{S}_1(\widehat{\boldsymbol{U}}^*) = \min_{\widehat{\boldsymbol{D}} \in \mathcal{E}} \mathcal{S}_1(\widehat{\boldsymbol{D}}) \leq \mathcal{U}_1 := \sum_{k:\lambda_k < 0} \lambda_k^2 + \sum_{kl} (\Delta p_k + \Delta p_l)^2,$$

$$\Delta p_k = \frac{1}{2} \sum_{l:\lambda_l < 0} \lambda_l \cdot \boldsymbol{v}_{kl}^2, \; \forall k \in [K] \tag{24}$$

$$\mathcal{S}_2(\widehat{\boldsymbol{U}}^*) = \min_{\widehat{\boldsymbol{D}} \in \mathcal{E}} \mathcal{S}_2(\widehat{\boldsymbol{D}}) \leq \sum_l \left( \Delta \overline{\boldsymbol{p}}_l \right)^2 =: \mathcal{U}_2, \tag{25}$$

*where the aggregated error term is defined as:*

$$\Delta \overline{\boldsymbol{p}}_l := \frac{1}{2K} \sum_{k=1}^{K} \sum_{l:\lambda_l < 0} \lambda_l \cdot \boldsymbol{v}_{kl}^2.$$

We aim to exploit the information derived from the truncation of the negative eigenspace of the matrix $\boldsymbol{CDC}$, specifically the matrix introduced in Eq.(22), defined as:

$$\widehat{\boldsymbol{L}}_{\boldsymbol{A}}^* = - \sum_{k:\lambda_k > 0} \lambda_k \boldsymbol{v}_k \boldsymbol{v}_k^\top,$$

where $\{\lambda_k, \boldsymbol{v}_k\}_{k \in [K]}$ denote the eigenvalues and corresponding eigenvectors of the matrix $\boldsymbol{F} = -\boldsymbol{CDC}$.

Recall that the entries of $\widehat{\boldsymbol{L}}_{\boldsymbol{B}}^*$ are given by:

$$\widehat{\boldsymbol{L}}_{\boldsymbol{B},kl}^* = \left[ \frac{1}{K} \boldsymbol{OD} + \frac{1}{K} (\boldsymbol{OD})^\top - \frac{1}{K^2} \boldsymbol{ODO} \right]_{kl} = \overline{\boldsymbol{D}}_k + \overline{\boldsymbol{D}}_l - \overline{\boldsymbol{D}}.$$

As a consequence, the sum $\widehat{\boldsymbol{L}}_{\boldsymbol{A}}^* + \widehat{\boldsymbol{L}}_{\boldsymbol{B}}^*$ may not be strictly hollow or PSD when $\boldsymbol{D}$ is not an EDM. To address this, we seek to construct a symmetric matrix $\boldsymbol{P}$ to be added to $\widehat{\boldsymbol{L}}_{\boldsymbol{A}}^*$, resulting in the matrix $\widehat{\boldsymbol{U}}^* := \widehat{\boldsymbol{L}}_{\boldsymbol{A}}^* + \boldsymbol{P}$, which is both hollow and PSD. This adjustment is designed to avoid any additional penalty on the term $\|\boldsymbol{A}\|_F^2$, though it may introduce some approximation errors in $\|\boldsymbol{B}\|_F^2$ and in the quantity $\mathcal{S}_2$. These induced errors contribute to the upper bound $\mathcal{U}$ for the optimal score $\mathcal{S}^*$.

We begin with the requirement that the matrix $\boldsymbol{P}$ does not contribute any additional penalty to the term $\|\boldsymbol{A}\|_F^2$. This can be ensured by imposing the constraint $\boldsymbol{CPC} = 0$. Under this condition, the matrix $\widehat{\boldsymbol{U}}^*$ remains a minimizer of $\|\boldsymbol{A}\|_F^2$, as demonstrated below:

$$\min_{\widehat{\boldsymbol{D}} \in \mathcal{E}_{\mathcal{L}}} \|\boldsymbol{A}\|_F^2 = \min_{\widehat{\boldsymbol{D}} \in \mathcal{E}_{\mathcal{L}}} \|\boldsymbol{C}(\widehat{\boldsymbol{L}}_{\boldsymbol{A}} + \widehat{\boldsymbol{L}}_{\boldsymbol{B}})\boldsymbol{C} - \boldsymbol{CDC}\|_F^2$$

$$= \|\boldsymbol{C}(\widehat{\boldsymbol{L}}_{\boldsymbol{A}}^* + \boldsymbol{P} + \widehat{\boldsymbol{L}}_{\boldsymbol{B}}^*)\boldsymbol{C} - \boldsymbol{CDC}\|_F^2$$

$$= \|\boldsymbol{C}\widehat{\boldsymbol{L}}_{\boldsymbol{A}}^*\boldsymbol{C} - \boldsymbol{CDC}\|_F^2,$$

where the final equality holds due to the constraint $\boldsymbol{CPC} = 0$.

This leads to the condition $(\boldsymbol{CP})\boldsymbol{C} = \boldsymbol{C}(\boldsymbol{PC}) = 0$, implying that $\boldsymbol{CP}$ lies in the left null space of $\boldsymbol{C}$, and $\boldsymbol{PC}$ lies in its right null space. As a result, all rows of $\boldsymbol{PC}$ must be constant, and this expression can be written as:

$$\boldsymbol{1}_K \boldsymbol{c}^\top = \boldsymbol{PC} = \boldsymbol{P} \left( \boldsymbol{I}_K - \frac{1}{K} \boldsymbol{O} \right) \text{ or } \boldsymbol{P} = \boldsymbol{1}_K \boldsymbol{c}^\top + \boldsymbol{P} \frac{1}{K} \boldsymbol{O},$$

where $c$ is a column vector to be defined subsequently. Here, we have used the fact that $C$ is the centering matrix defined by $C = I_K - \frac{1}{K}O$.

Multiplying both sides on the left by $\frac{1}{K}O$ yields:

$$\frac{1}{K}OP = \frac{1}{K}O1_Kc^\top + \frac{1}{K}O\left(\frac{1}{K}PO\right) = 1_Kc^\top + \frac{1}{K^2}OPO.$$

This leads to

$$c^\top = \frac{1}{K}1_K^\top P - \frac{1}{K^2}1_K^\top OPO.$$

Indeed, via the definition of $O = 1_K1_K^\top$, we can verify this as follows:

$$1_Kc^\top + \frac{1}{K^2}OPO = 1_K\left(\frac{1}{K}1_K^\top P - \frac{1}{K^2}1_K^\top OPO\right) + \frac{1}{K^2}OPO$$

$$= \frac{1}{K}1_K1_K^\top P - \frac{1}{K^2}1_K1_K^\top OPO + \frac{1}{K^2}OPO$$

$$= \frac{1}{K}1_K1_K^\top P - \frac{1}{K^2}OOPO + \frac{1}{K^2}OPO$$

$$= \frac{1}{K}OP.$$

Hence,

$$P = 1_K\left(\frac{1}{K}1_K^\top P - \frac{1}{K^2}1_K^\top OPO\right) + P\frac{1}{K}O$$

$$= \frac{1}{K}1_K(1_K^\top P) + \frac{1}{K}(P1_K)1_K^\top - \frac{1}{K^2}1_K1_K^\top OPO$$

Since $P1_K$ is a column vector, to satisfy this constraint, $P$ must be of the form:

$$P = 1_K\frac{p^\top}{K} + \frac{p}{K}1_K^\top - \widehat{p}\frac{1_K1_K^\top}{K},$$

where $p \in \mathbb{R}^K$ is a vector of free parameters, and $\widehat{p}$ denotes its average. This construction implies that $P$ has only $K$ degrees of freedom. However, to ensure that $\widehat{L}_A^* + P$ has zero diagonal (i.e., the resulting matrix is hollow), the diagonal entries of $P$ must satisfy the following $K$ linear constraints:

$$p_k - \frac{1}{2}\widehat{p} = -\frac{1}{2}[\widehat{L}_A^*]_{kk}, \quad \forall k \in [K].$$

Solving this linear system yields:

$$p_k = \frac{1}{2}\left(\sum_{l:\lambda_l>0}\lambda_l \cdot v_{kl}^2 + \frac{1}{K}\widehat{p}\right),$$

$$\widehat{p} = \frac{1}{K}\sum_{k=1}^K p_k = \frac{1}{K}\sum_{k=1}^K\sum_{l:\lambda_l>0}\lambda_l \cdot v_{kl}^2,$$

where we have used the fact that $\widehat{L}_A^* = -\sum_{l:\lambda_l>0}\lambda_l v_l v_l^\top$, and hence its diagonal entries are given by $[\widehat{L}_A^*]_{kk} = -\sum_{l:\lambda_l>0}\lambda_l \cdot v_{kl}^2$.

Consequently, the resulting matrix $P$ can be expressed element-wise as:

$$P_{k,l} = -\frac{[\widehat{L}_A^*]_{kk} + [\widehat{L}_A^*]_{ll}}{2} \geq 0,$$

where the inequality follows from the fact that $\widehat{L}_A^*$ is negative semi-definite.

In summary, the matrix $\widehat{U}^* := \widehat{L}_A^* + P$ satisfies all three constraints specified in Definition 1.

Although $\widehat{U}^*$ preserves the value of $\|A\|_F^2$, it differs from $\widehat{L}_A^*$ and introduces approximation errors in the $\|B\|_F^2$ term and the $\mathcal{S}_2$ term. Note that the sum of the untruncated version of $CDC$ and the matrix

$$\frac{1}{K}ODC + \frac{1}{K}CDO + \frac{1}{K^2}ODO$$

is equal to $D$ and remains hollow. Recall the decomposition:

$$\|\widehat{D} - D\|_F^2 = \|A\|_F^2 + \|B\|_F^2, \quad A := C\widehat{D}C - CDC,$$

$$\boldsymbol{B} := \frac{1}{K}\boldsymbol{O}\widehat{\boldsymbol{D}}\boldsymbol{C} + \frac{1}{K}\boldsymbol{C}\widehat{\boldsymbol{D}}\boldsymbol{O} + \frac{1}{K^2}\boldsymbol{O}\widehat{\boldsymbol{D}}\boldsymbol{O}$$
$$- \left(\frac{1}{K}\boldsymbol{O}\boldsymbol{D}\boldsymbol{C} + \frac{1}{K}\boldsymbol{C}\boldsymbol{D}\boldsymbol{O} + \frac{1}{K^2}\boldsymbol{O}\boldsymbol{D}\boldsymbol{O}\right),$$

where $\boldsymbol{C} = \boldsymbol{I}_K - \frac{1}{K}\boldsymbol{O}$ is the centering matrix and $\boldsymbol{O} = \mathbf{1}_K\mathbf{1}_K^\top$ is the all-ones matrix.

The matrix

$$\frac{1}{K}\boldsymbol{O}\boldsymbol{D}\boldsymbol{C} + \frac{1}{K}\boldsymbol{C}\boldsymbol{D}\boldsymbol{O} + \frac{1}{K^2}\boldsymbol{O}\boldsymbol{D}\boldsymbol{O}$$

can be written similarly to $\boldsymbol{P}$ by including the contributions from the negative eigenvalues, resulting in the matrix $\widetilde{\boldsymbol{P}}$, parameterized by:

$$\widetilde{\boldsymbol{p}}_k = \frac{1}{2}\left(\sum_l \lambda_l \cdot \boldsymbol{v}_{kl}^2 + \frac{1}{K}\widetilde{\overline{\boldsymbol{p}}}\right),$$

$$\widetilde{\overline{\boldsymbol{p}}} = \frac{1}{K}\sum_{k=1}^{K}\widetilde{\boldsymbol{p}}_k = \frac{1}{K}\sum_{k=1}^{K}\sum_l \lambda_l \cdot \boldsymbol{v}_{kl}^2.$$

Define the correction due to negative eigenvalues as:

$$\Delta\boldsymbol{p}_k := \frac{1}{2}\sum_{l:\lambda_l<0}\lambda_l \cdot \boldsymbol{v}_{kl}^2, \quad \forall k \in [K].$$

The resulting error in the $\|\boldsymbol{B}\|_F^2$ term is given by:

$$\|\boldsymbol{B}\|_F^2 = \|\widetilde{\boldsymbol{P}} - \boldsymbol{P}\|_F^2 = \sum_{k,l}\left(\Delta\boldsymbol{p}_k + \Delta\boldsymbol{p}_l\right)^2.$$

Furthermore, the contribution to $\mathcal{S}_2$ is bounded as:

$$\mathcal{S}_2 = \min_{\widehat{\boldsymbol{D}}\in\mathcal{E}}\mathcal{S}_2(\widehat{\boldsymbol{D}}) = \sum_{l\in[K]}\left(\widehat{\overline{\boldsymbol{D}}}_{K,l} - \overline{\boldsymbol{D}}_{K,l}\right)^2 \leq \sum_l\left(\Delta\overline{\boldsymbol{p}}_l\right)^2 =: \mathcal{U}_2,$$

where the aggregated error term is defined as:

$$\Delta\overline{\boldsymbol{p}}_l := \frac{1}{2K}\sum_{k=1}^{K}\sum_{l:\lambda_l<0}\lambda_l \cdot \boldsymbol{v}_{kl}^2.$$

### J.4 PROOF OF LEMMA 1

The proof is proved via leveraging Proposition 8.2 from Peyré et al. (2019), applied to the specific case $\alpha = 0$, and relies on the relationships among FGW, Wasserstein (W), and Gromov-Wasserstein (GW) distances.

The FGW cost $\text{FGW}_{p,\alpha}(\mathcal{G}_1, \mathcal{G}_2)$ is defined via two components: the node feature cost matrix $\boldsymbol{M}[i,j] = d_f(\boldsymbol{H}_1[i], \boldsymbol{H}_2[j])^p$, and the structural discrepancy tensor $\mathbf{L}(\boldsymbol{A}_1, \boldsymbol{A}_2)[i,j,l,m] = |\boldsymbol{A}_1[i,j] - \boldsymbol{A}_2[l,m]|^p$.

Let $\mathcal{G}_1 = (\boldsymbol{H}_1, \boldsymbol{A}_1, \boldsymbol{\omega}_1)$ and $\mathcal{G}_2 = (\boldsymbol{H}_2, \boldsymbol{A}_2, \boldsymbol{\omega}_2)$ be two attributed graphs with $N_1$ and $N_2$ nodes, respectively. Their associated probability measures are

$$\mu_1 = \sum_k \omega_{1k}\delta_{(\boldsymbol{x}_{1k}, \boldsymbol{a}_{1k})}, \quad \mu_2 = \sum_l \omega_{2l}\delta_{(\boldsymbol{x}_{2l}, \boldsymbol{a}_{2l})}.$$

We define the marginals $\mu_{\boldsymbol{H}_1} = \sum_k \omega_k\delta_{\boldsymbol{x}_k}$ and $\mu_{\boldsymbol{A}_1} = \sum_k \omega_k\delta_{\boldsymbol{a}_k}$ (and analogously for $\mu_{\boldsymbol{H}_2}$ and $\mu_{\boldsymbol{A}_2}$) as projections of $\mu_1$ and $\mu_2$ onto the feature and structural spaces, respectively.

Using these definitions, we introduce the following notation:

$$J_p(\boldsymbol{A}_1, \boldsymbol{A}_2, \boldsymbol{\pi}) = \sum_{ijkl} L_{ijkl}(\boldsymbol{A}_1, \boldsymbol{A}_2)^p \boldsymbol{\pi}_{ij}\boldsymbol{\pi}_{kl}, \tag{26}$$

$$\text{GW}_p(\mu_{\boldsymbol{H}_1}, \mu_{\boldsymbol{H}_2})^p = \min_{\boldsymbol{\pi}\in\boldsymbol{\Pi}(\boldsymbol{\omega}_1, \boldsymbol{\omega}_2)} J_p(\boldsymbol{A}_1, \boldsymbol{A}_2, \boldsymbol{\pi}), \tag{27}$$

$$H_p(\boldsymbol{M}, \boldsymbol{\pi}) = \sum_{kl} d_f(\boldsymbol{x}_{1k}, \boldsymbol{x}_{2l})^p \boldsymbol{\pi}_{kl}, \tag{28}$$

$$\text{W}_p(\mu_{\boldsymbol{A}_1}, \mu_{\boldsymbol{A}_2})^p = \min_{\boldsymbol{\pi}\in\boldsymbol{\Pi}(\boldsymbol{\omega}_1, \boldsymbol{\omega}_2)} H_p(\boldsymbol{M}, \boldsymbol{\pi}). \tag{29}$$

Let $\boldsymbol{\pi} \in \boldsymbol{\Pi}(\boldsymbol{\omega}_1, \boldsymbol{\omega}_2)$ be any admissible coupling. If both $\mu_1$ and $\mu_2$ are defined over a common metric space $(\boldsymbol{\Omega}, \boldsymbol{A}, \mu)$, then the FGW distance is given by:

$$\text{FGW}_{p,\alpha}(\mathcal{G}_1, \mathcal{G}_2) := \min_{\boldsymbol{\pi} \in \boldsymbol{\Pi}(\boldsymbol{\omega}_1, \boldsymbol{\omega}_2)} \langle (1-\alpha)\boldsymbol{M} + \alpha\,\mathsf{L}(\boldsymbol{A}_1, \boldsymbol{A}_2) \otimes \boldsymbol{\pi}, \boldsymbol{\pi} \rangle. \tag{30}$$

We now derive the following fundamental identity:

$$\mathbb{E}_{p,\alpha}\left(\boldsymbol{M}, \boldsymbol{A}_1, \boldsymbol{A}_2, \boldsymbol{\pi}\right) := \sum_{ijkl} \left[ (1-\alpha)d_f(\boldsymbol{x}_{1k}, \boldsymbol{x}_{2l})^p + \alpha\,|\boldsymbol{A}_1(i,k) - \boldsymbol{A}_2(j,l)|^p \right] \boldsymbol{\pi}_{ij}\boldsymbol{\pi}_{kl}$$

$$= (1-\alpha)H_p(\boldsymbol{M}, \boldsymbol{\pi}) + \alpha J_p(\boldsymbol{A}_1, \boldsymbol{A}_2, \boldsymbol{\pi}). \tag{31}$$

Moreover, let $\boldsymbol{\pi}_\alpha$ denote the optimal coupling that minimizes the FGW objective $\mathbb{E}_{p,\alpha}\left(\boldsymbol{M}, \boldsymbol{A}_1, \boldsymbol{A}_2, \cdot\right)$. Then the FGW distance admits the following decomposition:

$$\text{FGW}_{p,\alpha}^p(\mu_1, \mu_2) = \min_{\boldsymbol{\pi} \in \boldsymbol{\Pi}(\boldsymbol{\omega}_1, \boldsymbol{\omega}_2)} \mathbb{E}_{p,\alpha}\left(\boldsymbol{M}, \boldsymbol{A}_1, \boldsymbol{A}_2, \boldsymbol{\pi}\right) = \mathbb{E}_{p,\alpha}\left(\boldsymbol{M}, \boldsymbol{A}_1, \boldsymbol{A}_2, \boldsymbol{\pi}_\alpha\right)$$

$$= (1-\alpha)H_p(\boldsymbol{M}, \boldsymbol{\pi}_\alpha) + \alpha J_p(\boldsymbol{A}_1, \boldsymbol{A}_2, \boldsymbol{\pi}_\alpha)$$

$$\geq (1-\alpha)\text{W}_p^p(\mu_{\boldsymbol{A}_1}, \mu_{\boldsymbol{A}_2}) + \alpha\,\text{GW}_p^p(\mu_{\boldsymbol{H}_1}, \mu_{\boldsymbol{H}_2}). \tag{32}$$

This inequality follows from the optimality of the W and GW distances with respect to the cost functions $H_p$ and $J_p$, respectively, and highlights the interpolation nature of the FGW distance between these two metrics as governed by the parameter $\alpha$.

The generalized FGW cost $\mathbb{E}_{p,\alpha}\left(\boldsymbol{M}, \boldsymbol{A}_1, \boldsymbol{A}_2, \boldsymbol{\pi}\right)$ admits the following explicit formulation:

$$\mathbb{E}_{p,\alpha}\left(\boldsymbol{M}, \boldsymbol{A}_1, \boldsymbol{A}_2, \boldsymbol{\pi}\right) = \langle (1-\alpha)\,\boldsymbol{M}^p + \alpha\,\mathsf{L}(\boldsymbol{A}_1, \boldsymbol{A}_2)^p \otimes \boldsymbol{\pi}, \boldsymbol{\pi} \rangle$$

$$= \sum_{i,j,k,l} \left[ (1-\alpha)d_f(\boldsymbol{x}_{1k}, \boldsymbol{x}_{2l})^p + \alpha\,|\boldsymbol{A}_1(i,k) - \boldsymbol{A}_2(j,l)|^p \right] \boldsymbol{\pi}_{ij}\boldsymbol{\pi}_{kl}.$$

Based on the formulation above, we obtain the following upper bound on the FGW distance:

$$\text{FGW}_{p,\alpha}(G_1, G_2) \leq \langle (1-\alpha)\,\boldsymbol{M} + \alpha\,\mathsf{L}(\boldsymbol{A}_1, \boldsymbol{A}_2) \otimes \boldsymbol{\pi}, \boldsymbol{\pi} \rangle$$

$$\leq \sum_{k,l} \left[ (1-\alpha)\,d_f(\boldsymbol{x}_{1k}, \boldsymbol{x}_{2l}) + 2^{p-1}\alpha\,\boldsymbol{A}[k,l] \right]^p \boldsymbol{\pi}_{kl}, \tag{33}$$

where the second inequality follows from the convexity of the function $x \mapsto x^p$ for $p \geq 1$ and an application of Minkowski-type bounds on the structural term. Importantly, inequality in Eq.(33) holds for any admissible coupling $\boldsymbol{\pi} \in \boldsymbol{\Pi}(\boldsymbol{\omega}_1, \boldsymbol{\omega}_2)$, and in particular, it remains valid when $\boldsymbol{\pi} = \overline{\boldsymbol{\pi}}$, the optimal coupling associated with the Wasserstein distance $\text{W}_p(\mu_1, \mu_2)$ over the product metric space $(\boldsymbol{\Omega}, \overline{d})$. Here, the effective distance $\overline{d}$ between structured nodes $(\boldsymbol{x}_1, \boldsymbol{a}_1)$ and $(\boldsymbol{x}_2, \boldsymbol{a}_2)$ is defined as:

$$\overline{d}((\boldsymbol{x}_1, \boldsymbol{a}_1), (\boldsymbol{x}_2, \boldsymbol{a}_2)) = (1-\alpha)\,d_f(\boldsymbol{x}_1, \boldsymbol{x}_2) + 2^{p-1}\alpha\,\boldsymbol{A}(\boldsymbol{a}_1, \boldsymbol{a}_2).$$

Combining this with the Wasserstein formulation in Eq.(29), we observe the following inequality:

$$\text{FGW}_{p,\alpha}(\mathcal{G}_1, \mathcal{G}_2) \leq \text{W}_p(\mu_{\boldsymbol{A}_1}, \mu_{\boldsymbol{A}_2}), \quad \text{and} \quad \text{FGW}_{p,\alpha}(\mathcal{G}_1, \mathcal{G}_2) = \text{W}_p(\mu_{\boldsymbol{A}_1}, \mu_{\boldsymbol{A}_2}) \text{ when } \alpha = 0. \tag{34}$$

## K  E(3) INVARIANT PROPERTY

We utilize a 2D-MPNN, where node embeddings are iteratively refined across layers as follows:

$$\mathbf{h}_v^\ell = \text{UPD}^\ell\left(\mathbf{h}_v^{\ell-1}, \text{AGG}^\ell\left(\{\mathbf{M}^\ell\left(\mathbf{h}_v^{\ell-1}, \mathbf{h}_u^{\ell-1}, \boldsymbol{e}_{v,u}\right) : u \in N(v)\}\right)\right), \tag{35}$$

with $\mathbf{M}^\ell$ denoting the message function, $\text{AGG}^\ell$ representing aggregation by summation, and $\text{UPD}^\ell$ implemented as either the identity function or a multilayer perceptron. The final atom-level representation is obtained by integrating three modalities: the 2D molecular graph embeddings $\mathbf{H}_{\text{2D}}$, the 3D conformational features $\mathbf{H}_{\text{3D}}$, and the geometry-based structural descriptors $\mathbf{H}_{\text{GT}}$. This fusion is performed using trainable linear projections:

$$\mathbf{H}_{\text{comb}} = \widetilde{\mathbf{W}}_{\text{2D}}\,\mathbf{H}_{\text{2D}} + \widetilde{\mathbf{W}}_{\text{3D}}\,\mathbf{H}_{\text{3D}} + \widetilde{\mathbf{W}}_{\text{GT}}\,\mathbf{H}_{\text{GT}}, \tag{36}$$

where $\widetilde{\mathbf{W}}_{\text{2D}}, \widetilde{\mathbf{W}}_{\text{3D}}$, and $\widetilde{\mathbf{W}}_{\text{GT}}$ are trainable parameter matrices. Assuming that $\mathbf{H}_{\text{2D}}$ and $\mathbf{H}_{\text{GT}}$ are composed of $K$ repeated copies of their respective feature vectors, we compute the fused representation as:

$$\mathbf{H}_{\text{comb}} = \widetilde{\mathbf{W}}_{\text{2D}}\mathbf{H}_{\text{2D}} + \widetilde{\mathbf{W}}_{\text{3D}}\mathbf{H}_{\text{3D}} + \gamma\,\widetilde{\mathbf{W}}_{\text{GT}}\mathbf{H}_{\text{GT}}, \tag{37}$$

where $\gamma$ is a hyperparameter controlling the influence of the barycenter features. This fusion scheme allows balanced contributions from all modalities, which is empirically beneficial.

To predict the molecular property, we perform a mean pooling over the $K$ conformations and apply a linear transformation:

$$\widehat{\boldsymbol{y}} = \mathbf{W}^{\mathcal{G}} \left( \frac{1}{K} \sum_{k=1}^{K} \mathbf{H}_{\text{comb}}[k] \right) + \mathbf{b}^{\mathcal{G}}, \tag{38}$$

where $\mathbf{W}^{\mathcal{G}}$ and $\mathbf{b}^{\mathcal{G}}$ are the weight matrix and bias vector used for the final prediction.

We demonstrate that the function specified in Eq.(35) through Eq.(38)remains **invariant under both the action of the $E(3)$ and permutations of the input conformers.**

**Theorem 2** (E(3) Invariant Property). *Let $\mathcal{G}$ denote the 2D molecular graph, and let $(\mathbb{S}_1, \ldots, \mathbb{S}_K)$ be a collection of $K$ conformers, where each $\mathbb{S}_k = \{\mathbf{r}_{k,n}, Z_{k,n}\}_{n=1}^{N}$ for $k \in [K]$. Consider the function $\widehat{\boldsymbol{y}} = f_{\boldsymbol{\theta}}(\mathcal{G}, (\mathbb{S}_1, \ldots, \mathbb{S}_K))$ as defined by Eq.(35) to Eq.(38). Then, for any transformations $g_1, \ldots, g_K \in E(3)$, the following holds:*
$$f_{\boldsymbol{\theta}}(\mathcal{G}, (g_1\mathbb{S}_1, \ldots, g_K\mathbb{S}_K)) = f_{\boldsymbol{\theta}}(\mathcal{G}, (\mathbb{S}_1, \ldots, \mathbb{S}_K)).$$
*Furthermore, for any permutation $\pi \in \text{Sym}([K])$, we have:*
$$f_{\boldsymbol{\theta}}(\mathcal{G}, (S_{\pi(1)}, \ldots, S_{\pi(K)})) = f_{\boldsymbol{\theta}}(\mathcal{G}, (\mathbb{S}_1, \ldots, \mathbb{S}_K)).$$

*Proof of Theorem 2.* We establish the result in several steps. First, we consider the invariance properties of the geometric representation $\mathbf{H}_{\text{GT}}$. By construction, the geometry-aware embedding aggregation used to obtain $\overline{\mathbf{H}} = \mathbb{E}\left( \{\mathcal{T}_{\theta}(\mathbf{H}_i)\}_{i=1}^{K} \right)$, is invariant under permutation of conformers. Additionally, because $E(3)$ transformations preserve Euclidean distances and given that the upstream 3D MPNN is assumed to be $E(3)$-invariant, the generated features $\mathbf{H}_i$ are likewise invariant under such transformations.

Next, consider the aggregated representation defined in Eq.(37):
$$\mathbf{H}_{\text{comb}} = \widetilde{\mathbf{W}}_{2\text{D}}\mathbf{H}_{2\text{D}} + \widetilde{\mathbf{W}}_{3\text{D}}\mathbf{H}_{3\text{D}} + \widetilde{\mathbf{W}}_{\text{GT}}\mathbf{H}_{\text{GT}}.$$
From the prior step, we know that $\mathbf{H}_{\text{GT}}$ is invariant under both $E(3)$ actions and conformer permutations. Additionally, $\mathbf{H}_{3\text{D}}$ inherits $E(3)$ invariance from the 3D MPNN and is permutation equivariant, i.e., permuting the conformer inputs permutes the columns of $\mathbf{H}_{3\text{D}}$ accordingly. However, because the final prediction in Eq.(38) is based on an average over the conformer-wise features:

$$\widehat{\boldsymbol{y}} = \mathbf{W}^{\mathcal{G}} \left( \frac{1}{K} \sum_{k=1}^{K} \mathbf{H}_{\text{comb}}[k] \right) + \mathbf{b}^{\mathcal{G}}.$$

which is invariant to column permutations of the matrix $\mathbf{H}_{3\text{D}}$, leading to the final $\widehat{\boldsymbol{y}}$ is invariant to $E(3)$ group and permutation of 3D conformers. $\qquad\square$

