# OpenReview forum: "FACET: A Fragment-Aware Conformer Ensemble Transformer"
_ICLR.cc/2026/Conference — ICLR 2026 Poster_

### Official Review · Reviewer_Jbmu · 2025-10-24

**Soundness:** 2
**Presentation:** 3
**Contribution:** 3
**Rating:** 4
**Confidence:** 3

**Summary:**

This work focuses on molecular property prediction by integrating 2D graph information and 3D conformer ensemble information. The main contributions include the introduction of fragment-level structural priors, supported by comprehensive ablation studies, and a fused Gromov-Wasserstein (FGW) alignment pre-training stage that employs a Graph Transformer with an FGW distance loss. This replaces the computationally expensive FGW alignment process and leads to substantial efficiency improvements.

**Strengths:**

1. The paper is clearly written, and the proposed method is described in a detailed and logically consistent manner. The experimental setup is well documented, enhancing reproducibility.
2. The overall pipeline design is reasonable. The fragmentation strategy for generating fragmented graphs, along with the subsequent fusion of atom node features and fragmentation features, is well-motivated. The use of 3D-MPNNs (VisNet/SchNet) for extracting 3D conformer features ensures E(3)-invariance, making the overall framework theoretically consistent.
3. To address the high computational cost of FGW alignment, the work proposes replacing it with a Graph Transformer trained using a supervised FGW distance L2 loss. Section 4 provides theoretical analysis supporting this design, and experimental results (Fig. 2) show positive correlations. This approach significantly improves efficiency in both training and inference (Figs. 7 and 3).
4. The method achieves strong performance on multiple regression benchmarks, and the ablation study provides clear evidence for the effectiveness of the fragment-level structural priors.

**Weaknesses:**

1. Since the pipeline consists of three training stages, the pipeline illustration (Fig. 1) could be reorganized to clearly delineate these stages and indicate the corresponding losses for each stage. This would make the process easier to follow.
2. There is no comparison of model parameters and inference time across all baselines. While it may not be necessary to include every method, most recent approaches should be considered, rather than comparing only with CONAN-FGW.
3. There is no ablation or discussion on whether Training Stage 2 (i.e., training the Graph Transformer to approximate the FGW distance) is necessary. Specifically, what if Stage 1 is retained, but the Graph Transformer is randomly initialized and trained directly without Stage 2 pre-training (i.e., without the approximate FGW distance loss $\mathcal{L}_{enc}$)? Would this configuration still achieve good results, and how would it compare with the proposed approach? If the performance difference is small, it could indicate that Stage 2—and the overall FGW-based motivation—may not be critical to the method’s success.

**Questions:**

1. Regarding Weakness [3], if a relevant discussion already exists, please indicate where it appears. Otherwise, additional discussion and experimental results on this point would be valuable.
2. In the implementation, is the graph fragmentation step handled during data pre-processing (i.e., fragmented and stored in advance) or performed dynamically during each forward pass (i.e., fragmented in real time)? If it is the latter, what proportion of the total forward-pass time (from raw data input to loss computation) does it account for, and does it introduce noticeable computational overhead?

---

> ### Author Response · Authors · 2025-11-21
> **Reviewer to Reviewer Jbmu**
>
> We sincerely thank Reviewer for the thoughtful and constructive review. We appreciate your positive assessment of the clarity of the paper, the soundness of the overall pipeline design, and the motivation behind incorporating fragment-level structural priors. We are also grateful for your recognition of the theoretical consistency provided by the E(3)-invariant 3D-MPNN backbone and the efficiency gains from replacing FGW alignment with our Graph Transformer pre-training strategy.
>
> Below we address in detail your concerns and questions.
>
> **Q1. Reorganizing Figure 1 to clearly describe the Stages in Facet**
>
> **A1.** We thank the Reviewer for this valuable suggestion. In response, we have reorganized Figure 1 to more clearly illustrate the multi-stage training procedure in Facet. Specifically, we have added a new panel on the right to depict Stage 2, where the graph transformer is pre-trained using the Fused Gromov–Wasserstein loss. We also updated the main diagram to better reflect Stage 3, including the downstream prediction loss and the integration of geometry-aware embeddings.
>
> Regarding Stage 1, we chose not to include it fully in the figure because doing so would substantially increase visual complexity. The pre-training of the 2D-MPNN and 3D-MPNN components closely follows established practice from prior work, rather than introducing new contributions. To maintain clarity, we instead added a concise description of Stage 1 in the revised caption and provided a pointer to the relevant subsection in the paper where this stage is explained in detail.
>
> The updated figure and caption will be available in the revised PDF (a couple of hours after we post these responses). We will incorporate these improvements into the final version if the paper is accepted.
>
> **Q2. Providing additional comparison on model parameters and inference across typical baselines**
>
> **A2.** We thank the Reviewer for highlighting this important point. We agree that comparing model size and inference efficiency is valuable for understanding the practical trade-offs between different approaches. These metrics were not included in the original manuscript because they are not reported in the current MARCEL benchmark. To address this gap, we conducted additional experiments for the rebuttal.
>
> We selected the strongest 3D‐GNN ensemble baseline used in the MARCEL leaderboard, consisting of **SchNet**, **PaiNN**, **ClofNet**, and **LEFTNET**, and compared it against our Facet (SchNet backbone) model. The MARCEL ensemble uses 20 conformers per molecule as configured in the benchmark, whereas Facet uses 10 conformers, reflecting the design of our method.
>
> In the Table below, we report both:
> - (i) model parameter counts,
> - (ii) model’s performance measured in mean absolute error (MAE), and
> - (ii) inference time, measured as the time required to **generate predictions for the full test set** using batch size = 1 (i.e., processing one molecule at a time), where each molecule is evaluated with its respective number of conformers.
>
> | **Model** | **Kraken dataset (BurL)** *Param* | *Running time (s)* | *MAE* | **Drugs dataset (ip)**  *Param* | *Running time (s)* | *MAE* |
> |-----------|-------------------------------|-------------------|------|------------------------|-------------------|------|
> | SchNet    | 215215                        | 1.33              | 0.1443 | 210607               | 64.45             | 0.4452 |
> | PaiNN     | 1310209                       | 2.36              | 0.1324 | 1305601              | 80.44             | 0.4466 |
> | ClofNet   | 605122                        | 2.02              | 0.1548 | 600514               | 88.48             | 0.4280 |
> | LEFTNet   | 2722724                       | 6.49              | 0.1386 | 2718116              | 138.28            | **0.4149** |
> | FACET  (SchNet)   | 584065                        | 3.17              | **0.1245** | 584065               | 130.68            | *0.4235* |
>
> We will incorporate this analysis into the revised manuscript.

---

> > ### Author Response · Authors · 2025-11-21
> > **Reviewer to Reviewer Jbmu - Part 2**
> >
> > **Q3. Additional discussion on the Training Stage 2 of the Graph Transformer supervised by the FGW distance.**
> >
> > **A3.** We thank the Reviewer for this insightful suggestion. In the appendix, we previously included an ablation study examining the role of Stage 2. Specifically, we compared:
> > - (i) a variant where all three stages are merged into a single training procedure and optimized only with the downstream loss (first row).
> > - (ii) a variant where Stages 2 and 3 are merged, such that the graph transformer is trained jointly with the downstream model and periodically supervised by the FGW loss (second row), instead of being trained separately and frozen.
> >
> > For further clarity, during this rebuttal, **we additionally evaluated a (iii) third setting** in which, under the merged Stages 2–3 configuration, we remove the FGW supervision entirely and train solely with the downstream loss (third row). The results of these three variants are shown in the table below:
> >
> > | **Models**                               | **Esol**          | **Freesolv**        | **BACE**            | **Lipo**            |
> > |-------------------------------------------|--------------------|----------------------|----------------------|----------------------|
> > | **FACET**                                 | 0.516 ± 0.044      | 0.967 ± 0.083        | **0.495 ± 0.115**    | **0.424 ± 0.009**    |
> > | **FACET (Merge steps 2–3 using FGW)**     | **0.505 ± 0.014**  | **0.867 ± 0.102**    | 0.497 ± 0.035        | 0.440 ± 0.014        |
> > | **FACET (Merge steps 2–3, without FGW)**  | 0.535 ± 0.053      | 0.982 ± 0.007        | 0.530 ± 0.024        | 0.451 ± 0.080        |
> >
> > From the results, we observe that:
> >
> > - **Removing FGW supervision** (third row) consistently degrades performance across all datasets compared to both the default Facet configuration and the merged training with FGW (first and second rows). This confirms that FGW guidance plays an essential role in shaping geometry-aware representations that improve downstream prediction.
> >
> > - While the merged co-training approach (second row) achieves slightly better performance than Facet on ESol and FreeSolv, this comes with a **substantially higher training cost**, since FGW distances still need to be computed repeatedly throughout training rather than once in a dedicated pre-training stage.
> >
> > - Our default design - pre-training the graph transformer with FGW (Stage 2) and freezing it during Stage 3 - therefore provides a **more computationally efficient** and performance-stable solution.
> >
> > In short,  these results validate our motivation in learning a geometry-aware module explicitly regularized by FGW, and show that the FGW distance provides a meaningful supervisory signal that cannot be replaced by the downstream loss alone.
> >
> > **Q4. Is the graph fragmentation step handled during data pre-processing (i.e., fragmented and stored in advance) or performed dynamically during each forward pass (i.e., fragmented in real time)**
> >
> > **A4**. We thank the Reviewer for this question. Following common practice in prior work, such as [1], we **pre-compute the fragment graphs during data preprocessing** and store them to disk, rather than generating them dynamically during each forward pass.
> >
> > This design choice is motivated by two considerations:
> >
> > - **Substantial reduction in training time.** Fragment extraction involves non-trivial graph operations (e.g., subgraph identification, ring decomposition, atom–fragment correspondence). Performing these computations on-the-fly for every molecule in every training iteration would significantly slow down training, especially when using multiple conformers or large batch sizes. Pre-computing the fragment graphs avoids this repeated overhead and meaningfully accelerates training.
> >
> > - **Improved stability and reproducibility.** Fragmentation algorithms may involve heuristics or atom-ordering dependencies. Pre-computing the fragment graphs ensures that all training runs use exactly the same fragment structures, improving reproducibility, facilitating ablations, and avoiding subtle variations that could arise if fragmentation were performed dynamically during training.
> >
> > For these reasons, our workflow follows existing fragment-based GNN practices by generating the fragment graphs prior to training and retrieving them on demand during model optimization.
> >
> >
> >
> > [1] Wollschläger, Tom, et al. "Expressivity and generalization: Fragment-biases for molecular GNNs." ICML 2024

---

> > > ### Comment · Reviewer_Jbmu · 2025-11-24
> > > **Response to Authors' Rebuttal**
> > >
> > > Thank you for the new updates, and I apologize for the late response.
> > >
> > > Most of my previous concerns have been addressed. I have one remaining question: **how many of the baselines apply graph-fragmentation pre-processing**?
> > >
> > > I understand your choice to **pre-compute the fragment graphs during data preprocessing** in order to achieve **reduced training time** as well as improved **stability and reproducibility**. However, in real-world applications with previously unseen data, **graph fragmentation would need to be performed online (in real time)**.
> > >
> > > In this scenario, how does your approach trade off computational cost and latency compared to methods that **do not** rely on graph fragmentation?

---

> > > > ### Author Response · Authors · 2025-11-25
> > > > **Response to Reviewer**
> > > >
> > > > Dear Reviewer,
> > > >
> > > > Thank you very much for your follow-up questions.
> > > >
> > > > **Regarding baseline methods:**
> > > >
> > > > Among the models we compare against, none of the existing 2D–3D fusion baselines use the fragment-level graph. FACET is the only method in this setting that incorporates fragment graphs.
> > > >
> > > > **Regarding real-time (online) fragment extraction:**
> > > >
> > > > We agree that, during deployment, fragment extraction has to be performed on the fly. To evaluate this cost, we measured the average fragmentation time per molecule across the MoleculeNet datasets:
> > > >
> > > > - **ESOL**: 0.0066 s (avg. ~25 atoms per molecule)
> > > >
> > > > - **FreeSolv**: 0.0059 s (avg. ~17 atoms per molecule)
> > > >
> > > > - **BACE**: 0.013 s (avg. ~65 atoms per molecule)
> > > >
> > > > - **Lipo**: 0.009 s (avg. ~48 atoms per molecule)
> > > >
> > > > These results show that:
> > > >
> > > > - **Fragment extraction is extremely fast in practice**, typically 5–13 ms per molecule, making it feasible for real-time or large-scale settings.
> > > >
> > > > - **Runtime scales smoothly with molecular size**, as expected for lightweight 2D graph operations.
> > > >
> > > > Overall, this step introduces negligible overhead relative to conformer generation or 3D-GNN inference, and therefore does not create a bottleneck in practical applications. We hope this clarifies the remaining concern.
> > > >
> > > > Also, if the Reviewer feels that our detailed responses and additional experiments have adequately addressed the raised issues, we would be deeply grateful if the overall evaluation could be adjusted accordingly. We sincerely appreciate the time and care devoted to this review.
> > > >
> > > > Authors

---

> ### Comment · Reviewer_Jbmu · 2025-11-26
> **Raise My Score to 6/5**
>
> Thanks for authors' rebuttal, considered overall discussion, I'd like to raise my score to 6/5. good luck.

---

> > ### Author Response · Authors · 2025-11-27
> > **Thank You**
> >
> > Dear Reviewer Jbmu,
> >
> > We're glad our rebuttal addresses most of your concerns and appreciate that you increased your rating to 6.
> >
> > We will incorporate your suggestions into the revision of our paper as discussed. Please feel free to let us know if you have any further concerns.
> >
> > Best regards,
> >
> > Authors

---

### Official Review · Reviewer_Csta · 2025-10-26

**Soundness:** 3
**Presentation:** 3
**Contribution:** 3
**Rating:** 6
**Confidence:** 4

**Summary:**

This paper presents a novel architecture for molecular property prediction that integrates fragment-level chemical structure with 3D conformer ensembles. FACET introduces a Fragment-Conformer Attention Module (FCAM) that hierarchically aggregates information within fragments, across conformers, and at the whole-molecule level, ensuring rotational, translational, and permutational invariance. This fragment-aware design captures both local and global geometric interactions more effectively than atom-level models. FACET achieves state-of-the-art performance on multiple 3D molecular benchmarks while using fewer conformers and offering improved interpretability and computational efficiency

**Strengths:**

1. FACET introduces a novel multi-scale attention mechanism combining fragment, conformer, and molecule-level reasoning - an elegant fusion of chemistry-driven inductive bias with deep transformer architectures. It extends prior conformer-based GNNs by explicitly modeling fragment ensembles, bridging a gap between quantum-chemical interpretability and machine-learned representation power.
2. The paper's architectural design is technically sound and well-motivated, supported by ablation studies showing the benefit of fragment-aware aggregation.
3. The writing is clear and well-organized, moving systematically from intuition to formalism and experiments.
4. FACET represents a meaningful advance for 3D molecular ML, offering a scalable, interpretable, and chemically grounded alternative to purely atom-level GNNs.

**Weaknesses:**

1. **Dependence on Predefined Fragmentation:** The model’s reliance on fixed fragment decompositions could limit adaptability to novel or unusual chemistries.
2. **Theoretical Limitations:** The invariance proof (Theorem 1) assumes perfect fragment-conformer alignment and ignores potential numerical instabilities in coordinate normalization. The theorem's validity would benefit from a more rigorous group-theoretic formalization akin to SE(3)-equivariant models.
3. **Ablation on Conformer Ensemble Size:** While the authors claim reduced conformer dependence, quantitative experiments varying ensemble size (1, 5, 10, 20) are limited. Such results would strengthen the argument for FACET's data efficiency.
4. **Computational Overhead and Resource Reporting:** FACET adds multiple attention layers per fragment and conformer, but runtime and memory usage are not reported.

**Questions:**

1. **Fragment Definition and Generalization:** How are fragments defined and standardized across molecules? Is the fragmentation algorithm differentiable or fixed? Could learned or adaptive fragmentations improve performance on datasets with different chemical domains?
2. **Conformer Ensemble Sampling Strategy:** The method relies on conformer ensembles from datasets such as GEOM. How sensitive is FACET to the number of conformers used during training and inference? Could the model degrade when applied to datasets lacking reliable conformer ensembles (e.g., generated on-the-fly via RDKit)?
3. **Theoretical Guarantees on Invariance:** The paper asserts rotational, translational, and permutational invariance of FACET (Theorem 1), but the proof sketch in Appendix C assumes strict alignment across conformers. How does FACET ensure invariance when conformers differ significantly in spatial orientation or fragment positioning?
4. **Fragment-Level vs. Atom-Level Trade-offs:** Have the authors evaluated the granularity trade-off - i.e., how predictive accuracy changes when using atom-level embeddings versus different fragment granularity levels?
5. **Conformer Diversity Representation:** Does FACET perform any diversity regularization or weighting among conformers? Could the model collapse to a single representative conformer if conformer embeddings are highly correlated?
6. **Data Efficiency and Scalability:** While FACET achieves good performance, training time and parameter counts are not discussed in detail. How does FACET scale with molecular size and number of conformers compared to GNN baselines?

---

> ### Author Response · Authors · 2025-11-21
> **Response to Reviewer Csta - Part 1**
>
> We sincerely thank Reviewer Csta for the thorough and insightful evaluation of our work. Your detailed feedback on both the contributions and the potential limitations is extremely valuable and has directly helped us improve the paper. Below we address each of your questions and concerns in detail.
>
> **Q1. Data Efficiency and Scalability**
>
> **A1.** We thank the Reviewer for raising this important question. In the appendix, we originally provided training-time comparisons with several GNN baselines. During the rebuttal period, we expanded this analysis by running additional experiments on representative 3D-GNN models to more thoroughly examine FACET’s data efficiency and scalability.
> Because the two main benchmarks used in our paper, MARCEL and MoleculeNet, do not report model parameters or training times, we re-computed these values for widely used GNN baselines (SchNet, VisNet, GemNet).
>
> **I. FACET versus GNN baselines on Training Time, MSE, and Model Size Across Scaling Conditions**
>
> We evaluate models on two MoleculeNet datasets that contain the largest molecular structures:
>
> - **BACE**: 64.7 average nodes per molecule, 1,059 training samples.
> - **LIPO**: 48.4 average nodes per molecule, 2,940 training samples (~2.8× BACE).
>
> The table below shows the training times for all models, using the same number of conformers and trained for the same number of epochs. The best values are highlighted in bold.
>
> | **Model**            | **Metric**       | **BACE**            | **LIPO**            |
> |----------------------|------------------|----------------------|----------------------|
> | **GemNet**           | MSE              | 0.51 ± 0.07          | 0.45 ± 0.01          |
> |                      | Training Time    | 2.04 hours           | 4.8 hours            |
> |                      | Model Param      | 1.95 M               |                      |
> | **FACET (GemNet)**   | MSE              | **0.46 ± 0.03**      | **0.39 ± 0.02**      |
> |                      | Training Time    | 2.47 hours           | 6.4 hours            |
> |                      | Model Param      | 2.25 M               |                      |
> | **SchNet**           | MSE              | 0.64 ± 0.05          | 0.56 ± 0.01          |
> |                      | Training Time    | 1.4 hours            | 2.24 hours           |
> |                      | Model Param      | ~273 K               |                      |
> | **FACET (SchNet)**   | MSE              | 0.50 ± 0.03          | 0.42 ± 0.01          |
> |                      | Training Time    | 2.3 hours            | 3.16 hours           |
> |                      | Model Param      | ~584 K               |                      |
> | **VisNet**           | MSE              | 0.61 ± 0.15          | 0.55 ± 0.45          |
> |                      | Training Time    | 1.89 hours           | 4.27 hours           |
> |                      | Model Param      | 1.8 M                |                      |
>
> These results demonstrate that:
>
> - **FACET consistently improves predictive accuracy across 3D GNN backbones**.
> Across all tested backbones (GemNet, SchNet, VisNet), FACET yields clear performance gains regardless of model capacity or dataset size, demonstrating that the fragment–conformer integration benefits generalize broadly rather than being tied to a specific architecture.
>
> - **The added computational cost appears modest**. Although FACET introduces additional fusion components, the overhead relative to each backbone remains small. The observed increases in end-to-end training time are moderate (e.g., GemNet: 2.04h → 2.47h on BACE; SchNet: 1.4h → 2.3h). Importantly, most of this extra time comes from the separate pre-training of the graph transformer in Step 2, which takes roughly ~0.6 - 1 hour. The parameter growth is also limited mostly in the graph transformer module (e.g., GemNet: 1.95M → 2.25M; SchNet: 273K → 584K). In our experiments, these modest increases were consistently accompanied by improved predictive accuracy, suggesting a practical trade-off between accuracy and cost. We will add these analyses to the updated PDF.
>
> - **The added computational cost appears modest**. Although FACET introduces additional fusion components, the overhead relative to each backbone remains small. The observed increases in end-to-end training time are moderate (e.g., GemNet: 2.04h → 2.47h on BACE; SchNet: 1.4h → 2.3h). Importantly, most of this extra time comes from the separate pre-training of the graph transformer in Step 2, which takes roughly ~0.6 - 1 hour. The parameter growth is also limited mostly in the graph transformer module (e.g., GemNet: 1.95M → 2.25M; SchNet: 273K → 584K). In our experiments, these modest increases were consistently accompanied by improved predictive accuracy, suggesting a practical trade-off between accuracy and cost. We will add these analyses to the updated PDF.

---

> ### Author Response · Authors · 2025-11-21
> **Response to Reviewer Csta - Part 2**
>
> **Q1. Data Efficiency and Scalability (Continue)**
>
> **II. FACET versus GNN baselines when scaling training on varying the number of conformers**
>
> Our initial choice of {3, 5, 10, 20} conformers was motivated by both prior work and empirical observations. First, 20 conformers is the standard setting adopted in the MARCEL benchmark for all ensemble 3D-GNN baselines, and thus serves as an established upper reference point. Second, in our initial ablations, we observed that performance does not continue to improve beyond 5 or 10 conformers for FACET. Importantly, even with these relatively small conformer counts, FACET already achieves competitive or superior performance across two benchmarks. A comparative reference to other baselines is summarized in the table below.
>
> | **Method**                               | **Benchmark** | **Conformers** |
> |-------------------------------------------|---------------|----------------|
> | FACET                                     | MoleculeNet   | {3, 5}        |
> | FACET                                     | MARCEL        | {10}           |
> | ChemProp3D                                 | MoleculeNet   | 200            |
> | UniMOL                                     | MoleculeNet   | 11             |
> | 3D-GNN ensemble (GemNet, PaiNN, ClofNet, LEFTNet, etc.) | MARCEL | 20 |
>
> - To more thoroughly evaluate the Reviewer’s concern, we conducted additional experiments during the rebuttal period by training FACET with 50 and 100 conformers. The results are summarized in the table below.
> For completeness, we note that using 100 conformers for the BACE and LIPO datasets exceeded GPU memory (OOM) capacity in our setup and therefore could not be evaluated.
>
> | **Settings** | **3 conf.** | **5 conf. (default)** | **10 conf.** | **15 conf.** | **20 conf.** | **50 conf.** | **100 conf.** |
> |--------------|-------------|------------------------|--------------|--------------|--------------|--------------|----------------|
> | **ESOL**     | 0.539 ± 0.06 | 0.516 ± 0.04 | 0.501 ± 0.02 | 0.511 ± 0.03 | 0.546 ± 0.02 | 0.529 ± 0.04 | 0.530 ± 0.037 |
> | **FreeSolv** | 0.977 ± 0.25 | 0.967 ± 0.08 | 0.933 ± 0.23 | 0.946 ± 0.24 | 0.949 ± 0.21 | 0.940 ± 0.036 | 0.945 ± 0.039 |
> | **BACE**     | 0.542 ± 0.05 | 0.495 ± 0.03 | 0.513 ± 0.02 | 0.519 ± 0.01 | 0.517 ± 0.03 | 0.463 ± 0.004 | OOM |
> | **Lipo**     | 0.445 ± 0.02 | 0.424 ± 0.01 | 0.444 ± 0.02 | 0.447 ± 0.08 | 0.445 ± 0.01 | 0.440 ± 0.01 | OOM |
>
> Across both datasets, we observe no improvement when increasing the number of conformers after 10 to 50 or 100. In several cases, performance slightly degrades due to redundancy or noise introduced by large conformer sets. These findings are consistent with our earlier observation that FACET saturates quickly and does not benefit from large conformer ensembles.
>
> Taken together, these results confirm that our original choice of {3, 5, 10, 20} adequately captures the useful operating range for FACET, and that increasing the number of conformers beyond 10 does not provide meaningful gains. We will add these discussions in the revised manuscript if the paper is accepted.
>
> **III. Scaling Inference Time with Number of Conformers**
>
> We report the inference time compared with other 3D-GNN models. These results are summarized in **new figures 4,8** of the updated PDF, where the runtime reflects the total time required to generate predictions for all samples in the test set.
>
> From the figure, FACET exhibits linear runtime growth as the number of conformers increases. Although its runtime is slightly higher than SchNet and VisNet, it remains significantly lower than GemNet, demonstrating better computational efficiency relative to more complex geometric models. Overall, FACET provides a strong trade-off between accuracy and scalability, maintaining competitive performance while ensuring inference time scales predictably with the number of conformers.”
>
> **Q2. Fragment Definition and Generalization: how are fragments defined and standardized across molecules? Differentiable or fixed fragment algorithm? Potential learning to discover fragments.**
>
> **A2.** We clarify this question of the Reviewer on each question as follows.
>
> **I. How are fragments defined and standardized across molecules?**
>
> In our work, fragments are defined and extracted using the same ring–path fragmentation algorithm introduced by Wollschläger et al. [1]. This algorithm provides a consistent and fully deterministic way to decompose any 2D molecular graph into a vocabulary of structural units, which include:

---

> > ### Author Response · Authors · 2025-11-21
> > **Response to Reviewer Csta - Part 3**
> >
> > **Q2. Fragment Definition and Generalization: how are fragments defined and standardized across molecules? Differentiable or fixed fragment algorithm? Potential learning to discover fragments. (Continue)**
> >
> > - **Ring fragments (cycles)**: We first detect all minimal cycles (i.e., chordless rings) in the molecular graph using standard cycle-basis detection (as implemented in RDKit). Each ring, whether isolated, fused, or part of a polycyclic system, is treated as a ring fragment. This produces a canonical set of ring subgraphs for every molecule.
> >
> > - **Path fragments**: After removing all ring edges, the remaining graph forms acyclic components. From these components, the algorithm identifies maximal simple paths between nodes of degree ≠ 2. These paths represent linear chain segments, substituents, and ring-ring linkers. Each such maximal path becomes a path fragment.
> >
> > - **Junction nodes**: Whenever more than two fragments meet at an atom, e.g., ring-path-path intersections, the atom is labeled as a junction node in the fragment graph. This ensures that complex branching is captured independently of molecule size.
> >
> > Because the algorithm extracts fragments purely based on the **graph topology**, it defines fragments in the same way for every molecule. This means that the same ring or chain pattern will always produce the same type of fragment.
> >
> > [1] Wollschläger, Tom, et al. "Expressivity and generalization: Fragment-biases for molecular GNNs." ICML 2024
> >
> > **II. Adaptability to New or Unusual Chemistries**
> >
> > Our method is naturally adaptable to new or unusual chemistries because the ring–path fragmentation we use is **purely topology-based** and does **not depend on predefined chemical templates** or **functional-group rules** [2,3]. The algorithm simply decomposes any molecular graph into cycles, paths, and junctions using general graph operations (cycle detection and path enumeration). As a result, it can handle unconventional ring systems, rare heterocycles, synthetic scaffolds, or AI-generated molecules as long as they can be represented as a standard 2D molecular graph.
> > Because the model learns fragment representations directly from data rather than relying on a fixed fragment vocabulary, it can easily incorporate and generalize to novel chemistries that were not present during initial training.
> >
> >
> > [2] Degen,J. et al. (2008) On the art of compiling and using ‘Drug-Like’ chemical fragment spaces. Chemmedchem.
> >
> > [3] Lewell, Xiao Qing, et al. "Recap retrosynthetic combinatorial analysis procedure: a powerful new technique for identifying privileged molecular fragments with useful applications in combinatorial chemistry." Journal of chemical information and computer sciences 38.3 (1998)
> >
> > **III. Is learning a fragment graph from data better?**
> >
> > This is an interesting question; however, we argue that in practice, it is significantly harder and less stable. A model would have to jointly learn **how to cut** the molecule and **how to represent** the resulting fragments, which typically **requires much larger datasets** and introduces substantial optimization variance. In contrast, the ring-path fragmentation we adopt provides a deterministic, chemically interpretable decomposition and allows the model to focus entirely on learning representations.
> >
> > Moreover, prior work has already shown that **domain-informed fragment definitions** are highly effective in molecular modeling thanks to maintaining chemical validity and capturing meaningful structural units. In contrast, end-to-end learned fragment extraction can easily produce **unnatural or non-chemically meaningful fragments**, leading to unusual or difficult-to-interpret structures that reduce model transparency. For these reasons, the ring-path approach would offer a more robust, interpretable, and data-efficient solution in our setting.
> >
> > **Q3. Conformer Ensemble Sampling Strategy**
> >
> > **A3.** Our method simply **uses the conformer ensembles provided by each benchmark** and does not depend on a specific generator. In **MARCEL**, the conformers are high-quality **DFT-optimized structures** from GEOM (via Auto3D and AIMNet-NSE), while in **MoleculeNet**, we follow CONAN-FGW and use **RDKit-generated conformers**. The level of theory (DFT vs. a simple force field to equilibrate) is much lower for the latter method.  However, FACET performs well in both settings, indicating that it does not require energetically highly accurate conformers.
> >
> > To assess robustness, we repeated all MoleculeNet experiments using different randomly sampled RDKit conformers during both training and testing and reported the corresponding means and standard deviations. We also ran an extended ablation with 3, 5, 10, and 20 conformers. Across all settings, performance remained stable, with no gains beyond 5–10 conformers. These results suggest that FACET is not strongly affected by conformer quality or ensemble size and remains consistent even with inexpensive, on-the-fly RDKit conformers.

---

> > > ### Author Response · Authors · 2025-11-21
> > > **Response to Reviewer Csta - Part 4**
> > >
> > > **Q4. Theoretical Guarantees on Invariance**
> > >
> > > **A4.**
> > > We thank the Reviewer for raising this point. We would first like to clarify that Theorem 1 does not concern rotational, translational, or permutational invariance; instead, it provides an upper bound relating the outputs of our learnable graph transformer to the FGW (Fused Gromov-Wasserstein) distance. We believe the Reviewer is referring to **Theorem 2**, which states the invariance properties of FACET.
> > >
> > > Regarding Theorem 2, FACET does not require any alignment between conformers. Each conformer is encoded independently using an **E(3)-invariant 3D-MPNN** (e.g. SchNet), which depends only on geometric quantities invariant to rigid motions (e.g., pairwise distances). Thus, rotating or translating one conformer does not change its fragment- or conformer-level representation, regardless of how the other conformers are oriented.
> > >
> > > FACET then **aggregates per-conformer embeddings using permutation-invariant attention** and pooling. Since this aggregation acts solely on already E(3)-invariant features, the final molecular representation is invariant to (i) any rigid-body motion applied independently to each conformer and (ii) any reordering of conformers. However, it’s essential to note that the model remains sensitive only to true conformational differences - changes in internal geometry that affect interatomic distances, which is the desired behavior.
> > >
> > > In the revision, we will (i) restate the invariance theorem explicitly in terms of the group action of rigid motions and permutations on the conformer ensemble, and (ii) provide a more systematic group-theoretic formulation in line with standard treatments of SE(3)-invariant and SE(3)-equivariant architectures. **A full extension of FACET to a strictly SE(3)-equivariant formulation with non-trivial tensor representations** (beyond the invariant scalar features we currently use) is an **interesting direction that we leave for future work**.
> > >
> > > We will clarify these points in the revised manuscript. If our explanation does not fully resolve the Reviewer’s concern, we would be very grateful for further guidance on what additional details would be most helpful.
> > >
> > >
> > > **Q5. Fragment-Level vs. Atom-Level Trade-offs**
> > >
> > > **A5**. We thank the Reviewer for raising this important point. In the current manuscript, we primarily compared (i) atom-level GNNs with 3D conformers and (ii) our full FACET configuration that combines atom-level, fragment-level graphs, and 3D conformers. We had not explicitly isolated the granularity trade-off between atom-level and fragment-level representations.
> > >
> > > To address this, we conducted new ablation experiments during the rebuttal on ESOL and FreeSolv by removing either the fragment-level branch or the atom-level branch from FACET:
> > >
> > > - **w/o Fragment**: only atom-level representations,
> > >
> > > - **w/o Atom**: only fragment-level representations,
> > >
> > > - **FACET**: both levels jointly.
> > >
> > > | **Settings** | **FACET** | **w/o Fragment** | **w/o Atom** |
> > > |--------------|-----------|-------------------|--------------|
> > > | **ESOL**     | 0.516     | 0.531             | 0.603        |
> > > | **FreeSolv** | 0.967     | 1.072             | 0.986        |
> > >
> > > The results in Table show that (i) first, removing **either granularity** consistently reduces performance; (ii) atom-only and fragment-only models are both weaker than the full model, and (iii) finally, the combination of atom-level and fragment-level information yields the best predictive accuracy.
> > >
> > > In short, this indicates that the two granularities are complementary, and integrating them enables FACET to learn a more discriminative and robust representation compared to using either level in isolation. We will incorporate these new results into the revised version if the paper is accepted.
> > >
> > > **Q6. Conformer Diversity Representation**
> > >
> > > **A6.**  We appreciate the Reviewer’s question regarding whether FACET performs any explicit diversity regularization across conformers. In our current design, we use a simple and uniform aggregation of conformer embeddings; however, two factors mitigate the risk of collapse and make the approach robust in practice.
> > >
> > > **(i) Single-conformer molecules are common and valid cases.**
> > >
> > > Some molecules naturally have only a single stable conformer (e.g., Benzene or Cyclohexane in Chair Form). In such cases, collapsing to a single representative geometry is not only expected but also chemically correct. FACET should be able to represent such molecules using a single dominant conformer without any loss of fidelity. Therefore, a scenario in which one conformer embedding becomes representative is not inherently problematic and may reflect true molecular behavior.

---

> ### Author Response · Authors · 2025-11-21
> **Response to Reviewer Csta - Part 5**
>
> **Q6. Conformer Diversity Representation (continue)**
>
> **(ii) Although the graph transformer encoder is fixed, the conformer representations are not fixed.**
>
> FACET uses **learnable input feature embeddings** and a **trainable adapter network** on top of the transformer outputs (Stage 3). These components are directly optimized by the downstream loss, meaning the model can still adjust how information from each conformer is encoded and integrated. Specifically,  (a) the learnable input embeddings allow the model to modulate how each atom in each conformer is represented before entering the graph transformer; (b) the learnable adapter layers map the transformer outputs to the final molecular representation, **enabling the model to emphasize or down-weight information from different conformers even under uniform averaging**.
>
> Thus, while conformer embeddings from the frozen transformer are averaged uniformly, the trainable input and adapter layers provide an implicit, task-driven reweighting mechanism. The model can automatically shape the contribution of each conformer according to the gradient feedback from the downstream objective.

---

> > ### Comment · Reviewer_Csta · 2025-11-24
> > **Author Response Acknowledgement**
> >
> > Thank you to the authors for the detailed and thoughtful rebuttal, as well as the extensive additional experiments provided during the discussion phase. The clarifications on computational scaling, conformer-count sensitivity, fragmentation methodology, and the formal treatment of invariance were clear and addressed the majority of my earlier concerns. I appreciate the newly added analyses (training/inference time, parameter growth, fragment vs. atom ablations, and large-conformer experiments), which strengthen the empirical narrative. Based on the rebuttal, I retain my original score.
> >
> > I have one more concern: Do you have qualitative examples where very similar FGW pairs are correctly separated only after fragment-aware attention is added (or vice-versa)?

---

> > > ### Author Response · Authors · 2025-11-27
> > > **Follow up reviewer's response**
> > >
> > > Dear Reviewer,
> > >
> > > Thank you for your follow-up question regarding the qualitative examples and whether similar FGW pairs are correctly separated after applying fragment-aware attention. We appreciate your insightful comment.
> > >
> > > We have been considering the best way to provide strong evidence for this point and have designed an additional experiment to directly address your concern. **The experiment is currently running**. Once we obtain the results, we will update the corresponding figure in the manuscript and inform you accordingly.
> > >
> > > Thank you very much for your valuable feedback.
> > >
> > > Regards,
> > >
> > > The Authors

---

> > > > ### Author Response · Authors · 2025-12-04
> > > > **Updating manuscripts with both quantitative and qualitative analyses about conformer diversity**
> > > >
> > > > Dear Reviewer Csta,
> > > >
> > > > We thank you again for the reviewer’s concern regarding potential conformer collapse. To address this, we have updated the new PDF to include both **quantitative and qualitative analyses**, assessing conformer diversity and embedding behavior across all datasets. As shown in **Section C, Appendix** and the accompanying discussion, molecules exhibit non-zero FGW-based conformer diversity, and even the least diverse molecules do not collapse into a single conformer embedding.
> > > >
> > > > In addition, **we performed a t-SNE** visualization using conformer-level embeddings and the mean embeddings produced by our Graph Transformer. By projecting these embeddings into 2D space for molecules with both large and small FGW distances, we observed that the mean embedding remains clearly separated from individual conformer embeddings, and the conformers occupy distinct regions. This further confirms that our model preserves meaningful conformer variability and does not suffer from embedding collapse.

---

### Official Review · Reviewer_1ftv · 2025-10-31

**Soundness:** 2
**Presentation:** 2
**Contribution:** 2
**Rating:** 4
**Confidence:** 4

**Summary:**

This paper introduces FACET, a hybrid model integrating a Message Passing Neural Network (MPNN) with a Graph Transformer (GT) to accurately predict molecular properties from 2D molecular graphs and 3D conformations. Existing methods for aligning graph and conformational features often suffer from high computational costs. FACET addresses this challenge through two key innovations: (1) It leverages a GT to approximate the computation of the Fused Gromov-Wasserstein (FGW) distance, enabling efficient feature alignment with a provable theoretical error bound; (2) It embeds fragment-level structural priors from the molecular graph and sampled conformational features into the MPNN and GT, effectively capturing multi-scale relationships between the two molecular representations.
Experimental results demonstrate that FACET outperforms several baseline models, including CONAN-FGW, on the MoleculeNet and MARCEL datasets. Furthermore, FACET exhibits significantly faster training and inference speeds, validating its computational efficiency.

**Strengths:**

1. The use of a GT to approximate the FGW distance is a notable innovation, combining theoretical rigor with algorithmic efficiency for feature alignment.
2. The model design is sound, integrating fragment-level structural priors and conformational features through a multi-scale message-passing architecture.
3. The model achieves a favorable balance between predictive performance and computational speed, underscoring its practical utility.
4. Comprehensive evaluations on multiple benchmark datasets provide strong, empirical support for the model's effectiveness and efficiency.

**Weaknesses:**

1. The abstract inadequately highlights the core contributions, particularly the novel use of GT for efficient FGW approximation.
2. The model diagram (Figure 1) lacks clarity. The MPNN for processing 3D conformations is represented only by the symbol "Φ", and the GT is depicted generically as an "attention mechanism," which hinders a straightforward understanding of the model's architecture.
3. The conclusion is underdeveloped, merely restating the main contributions and results without critical analysis or forward-looking perspectives.
4. Table 1 contains errors. The column header "Model" is incorrect and should be "Dataset" based on the content. Furthermore, the sum of the values under the "Train," "Valid," and "Test" rows for the "Lipo" dataset does not match the value in the "Total" row.

**Questions:**

1. Revise the abstract to more prominently and explicitly state the core methodological innovations.
2. Redesign Figure 1 to provide a clearer, more detailed visual representation of the MPNN and GT components.
3. Integrate the key points from the appendix's "Limitations of FACET" section into the main "Conclusion" section to offer a more critical summary and a concrete outlook for future work.
4. Conduct a thorough review of all tables, especially Table 1, to ensure data consistency and accuracy across all entries.

---

> ### Author Response · Authors · 2025-11-21
> **Response to Reviewer 1ftv**
>
> We sincerely appreciate the Reviewer’s efforts for your very detailed and useful suggestions to improve the writing quality in terms of figures and further emphasize our contribution in the abstract and conclusion parts. Below, we provide the updated writing for each of your concerns, followed by the updated PDF with the changes highlighted in orange.
>
> **Q1. Revise the abstract to more prominently and explicitly state the core methodological innovations**
>
> **A1** Below is our new, improved version of the abstract with the changed text in bold (in this openreview) and dark orange (in the updated pdf).
>
> > Accurately predicting molecular properties requires effective integration of structural information from both 2D molecular graphs and their corresponding equilibrium conformer ensembles. In this work, we propose FACET, a scalable Structure-Aware Graph Transformer that efficiently aggregates features from multiple 3D conformers while incorporating fragment-level information from 2D graphs. **Unlike prior methods that rely on static geometric solvers or rigid fusion strategies, our approach utilizes a differentiable graph transformer to theoretically approximate the computationally expensive Fused Gromov-Wasserstein (FGW), enabling dynamic and scalable fusion of 2D and 3D structural information**. We further enhance this mechanism by injecting fragment-specific structural priors into the attention layers, enabling the model to capture fine-grained molecular details.  **This unified design scales to large datasets, handling up to 75,000 molecules and hundreds of thousands of conformers, and provides over a 6x speedup compared to geometry-aware FGW-based baselines**. Our method also achieves state-of-the-art results in molecular property prediction, Boltzmann-weighted ensemble modeling, and reaction-level tasks, and is particularly effective on chemically diverse compounds, including organocatalysts and transition-metal complexes.
>
> **Q2. Extending the Conclusion by the Limitation of FACET in the appendix**
>
> **A2.** We apologize for omitting the critical analysis and forward-looking perspectives in the current Conclusion section, which were limited due to the page constraints.
>
> Below, we present to the Reviewer a revised version with the changed text highlighted (dark orange in the updated PDF).
>
> > **We introduce FACET, a scalable framework that replaces costly FGW alignment with a Graph Transformer trained to approximate FGW fusion between 2D fragments and 3D conformers. This approximation enables efficient, end-to-end fusion of 2D and 3D structure, yielding strong gains across MoleculeNet and state-of-the-art performance on the large-scale MARCEL benchmark.**
>
>
> > **While FACET demonstrates strong performance on small, drug-like molecules, its evaluation has so far been limited to these standard benchmarks. This leaves open questions about its generalizability to more complex molecular regimes such as biomacromolecules with long-range dependencies, polymers and materials that lack stable conformers, and multi-molecular systems such as protein–ligand interactions. Addressing these challenges presents several promising future directions, including incorporating attention mechanisms capable of capturing both local fragment-level information and long-range structural dependencies, extending FACET to flexible input formats such as voxel grids or material-specific graphs is also suitable for macromolecules and materials, and developing cross-graph or co-embedding strategies to model intermolecular interactions. Finally, broader evaluation on datasets such as PDBbind and PolyInfo would further clarify FACET’s applicability across diverse molecular domains.**
>
> **Q3. Thorough review of all tables to ensure data consistency across all entries**
>
> **A3.** We thank the reviewer for pointing out these inconsistencies. We apologize for the typographical errors in Table 1 and the incorrect column header name. All tables have now been reviewed and corrected. We also highlight the revised entries and newly added results in dark orange for clarity. We sincerely appreciate the reviewer’s attention to detail and helpful feedback.
>
> **Q4. Updating Figure 1 regarding the 3D-MPNN, better clarifying the Graph Transformer module**
>
> **A4.** We thank the Reviewer for this helpful suggestion. We have updated Figure 1 (which will be uploaded in a couple of hours) to enhance the clarity of the 3D-MPNN component and to more explicitly illustrate the role of the graph transformer module. In the revised figure, we now clearly distinguish how the transformer is **pre-trained** and how it is **subsequently used within Facet**. Corresponding additions and clarifications have been incorporated into the caption, with all updates highlighted in dark orange for ease of reference.

---

> ### Author Response · Authors · 2025-11-27
> **Follow up on our response to Reviewer**
>
> Dear Reviewer 1ftv,
>
> As the discussion period is approaching its end and we have not yet received your feedback, we would like to kindly ask whether the current updates have addressed your concerns. If you have any additional questions or comments, please feel free to let us know.
>
> Thank you very much.
>
> Best regards,
>
> The Authors

---

### Official Review · Reviewer_NGyk · 2025-11-01

**Soundness:** 4
**Presentation:** 4
**Contribution:** 3
**Rating:** 6
**Confidence:** 3

**Summary:**

This paper presents a well-executed study on a scalable framework for integrating conformational information into molecular property prediction. The authors propose a learned embedding of the Fused Gromov-Wasserstein (FGW) distance to efficiently aggregate 3D conformers, enabling the model to scale to large datasets. Additionally, fragment-level structural priors are incorporated to enhance representation learning. The proposed method achieves state-of-the-art performance across multiple benchmarks, demonstrating its effectiveness.

**Strengths:**

- **Scalability**: The use of learned FGW embeddings enables efficient conformer aggregation, making the method suitable for large-scale datasets.
- **Structural Integration**: The combination of 3D conformers with 2D fragment-level information provides a rich, multi-scale representation of molecular structure.
- **Empirical Performance**: The method consistently outperforms strong baselines across diverse tasks, offering compelling evidence of its advantage.
- **Architectural Flexibility**: The framework is compatible with multiple backbone architectures and tasks, highlighting its versatility.

**Weaknesses:**

- **Split Strategy**: All experiments rely on random splits. Evaluating performance under scaffold splits would better assess the model’s ability to generalize across distinct chemical scaffolds—a key criterion for real-world applicability.
- **Role of 2D Fragments**: The necessity of 2D graphs and fragment-level features is not fully justified, given that 3D conformers already encode topological and spatial information.
- **Limited Ablation Scope**: Ablation studies are restricted to MoleculeNet tasks. Extending these analyses to other datasets (e.g., MARCEL) would strengthen the claims and demonstrate robustness across domains.

**Questions:**

Given that the proposed method approximates real FGW calculations using a learned embedding, is the performance theoretically upper-bounded by the true FGW metric? If so, under what conditions would the approximation match or exceed FGW-based methods in practice?

---

> ### Author Response · Authors · 2025-11-21
> **Response to Reviewer NGyk - Part 1**
>
> Dear Reviewer NGyk, we thank you a lot for your feedback on our work, especially your acknowledgment of the FACET’s strengths regarding Scalability, Structural Integration (via fragment-level graph), and well supported empirical performance with the compatibility of different architectures and tasks. Below, we address your remaining questions in detail.
>
> **Q1. Split Strategy: random versus scaffolds splitting**
>
> **A1**. Thank you for this insightful question. In this work, our choice of splitting strategy was guided by the goal of maintaining **fair, consistent, and reproducible comparison** with prior studies on the two benchmarks evaluated. Below, we detail the exact protocols followed:
>
> - **MoleculeNet**: For the Lipo, ESOL, FreeSolv, and BACE datasets, we follow the standard evaluation protocol used by prior works that report performance on these datasets. These baselines adopt both **scaffold split** and **random** setting described in the MoleculeNet paper. To ensure direct comparability, we apply the same approach and report the mean and standard deviation over five runs.
>
> - **Marcel Benchmark**: We strictly follow the dataset partitions and random seeds released by the original authors. These benchmarks are based on **random splits** for all 2D, 3D, and ensemble configurations. Using identical splits ensures our reported numbers can be reliably compared to the previously published baselines.
>
> While we acknowledge the potential value of evaluating models under additional scaffold-based splits for the Marcel benchmark, doing so would require rebuilding the entire benchmark, i.e., retraining all 2D, 3D, and ensemble baselines under the new splits, which would be a substantial and error-prone undertaking, potentially breaking comparability with the official benchmark numbers.
>
> **Q2. Further analysis on the role of 2D graphs and fragment-level graphs**
>
> **A2** Thank you for requesting a deeper analysis of the role of the 2D graph. While our original ablation study **validated the contribution of the fragment-level features**, we agree that isolating the **2D molecule graph** component is crucial.
> To address this, we conducted two new ablation experiments on the ESOL and FreeSolv datasets, which explicitly test the reliance on the 2D graph information:
>
> - **w/o 2D graph**: Retaining only the 2D fragment graph and the 3D conformer.
> - **w/o Frag + 2D graph**: Retaining only the 3D conformer.
>
> The complete results, comparing the full *FACET* model against these ablations and our original 'w/o Frag', i.e., without using 2D fragment graph settings, are summarized in the updated table below:
>
> | **Settings** | **FACET** | **w/o Frag.** | **w/o 2D graph** | **w/o Frag + 2D graph** |
> |--------------|-----------|----------------|-------------------|--------------------------|
> | **ESOL**     | **0.516** | 0.531          | 0.603             | 0.540                    |
> | **FreeSolv** | **0.967** | 1.072          | 0.986             | 1.03                     |
>
> It can be seen that the best performance occurs when the 2D graph, fragment graph, and 3D conformer are combined, showing that each provides complementary information. The 2D graph supplies the invariant topological backbone, the fragment graph captures higher-level chemical motifs, and the 3D conformer adds essential spatial geometry. Those factors together enable a stronger, more complete molecular representation.
>
>
> **Q3. Ablation study is mostly restricted to the MoleculeNet. Extending on other datasets would strengthen the claim.**
>
> **A3**. We thank the Reviewer for this valuable suggestion. The original motivation for conducting the ablation studies on the four datasets from MoleculeNet, rather than running on the large-scale datasets of Marcel, such as Drug-75k with 75,099 molecule samples, was the time-consuming and GPU-intensive nature of the tasks. For instance, as mentioned in the Marcel benchmark, **several backbone models (such as GemNet)** required running with multiple **A100 40GB or H100-80GB GPUs** and consumed *thousands of GPU hours*  (for 2000 epochs) to complete each test.
>
> However, we have conducted an additional ablation study for the *Kraken dataset (**Marcel benchmark**)*, whose numbers (MAE) are averaged across four values $B_{5}$, L, $BurB_{5}$, and BurL.  This extends our previous analysis on Molecule-Net and is summarized below:
>
> | **Settings** | **FACET** | **w/o Frag.** | **w/o Frag. in Trans.** | **w/o Adap.** |
> |--------------|-----------|----------------|-------------|--------|
> | **ESOL**     | **0.516** | 0.531          | 0.525                     | 0.546         |
> | **FreeSolv** | **0.967** | 1.072          | 0.973                     | 1.085         |
> | **Kraken**   | **0.238** | 0.247              | 0.242                         | 0.262             |
>
> It can be seen that other core components in FACET continue to make contributions toward final improvement records.

---

> > ### Author Response · Authors · 2025-11-21
> > **Response to Reviewer NGyk - Part 2**
> >
> > **Q4. Correlation in terms of a theoretically upper-bounded relationship between FACET and the True FGW metric. Under which conditions can FACET match or exceed the FGW-based methods?**
> >
> > **A4** This is a very interesting question for us. We address the theoretical and practical aspects below:
> >
> > **I. Theoretical Upper Bound**
> >
> > The learned structural embedding (the FGW approximation) is theoretically upper-bounded by the output of the true optimal transport solver for the FGW metric. Our best-case scenario for this component is to perfectly predict the true FGW value.
> >
> > **II. Practical Performance Match**
> >
> > However, our FACET’s performance is **not necessarily upper-bounded** by a model using a true FGW feature because our method involves factors beyond simple structural prediction:
> >
> > - **Non-Linear Refinement**: The structural embedding is just a feature input to a final Multi-Layer Perceptron (MLP), which is optimized end-to-end using the downstream property loss. This MLP acts as a non-linear tuner, refining the FGW feature for the predictive task.
> >
> > - **Richer Information**: FACET explicitly leverages the fragment-level molecule graph within its attention mechanism, providing richer, chemically meaningful input that exceeds the information explicitly processed by a standard FGW calculation on the raw 2D graph.
> >
> > - **Regularization Role**: In the Table below, we present another FACET version where the graph transformer is supervised by both the FGW and downstream loss. In this case, *FGW is primarily used as a regularizer*, ensuring structural awareness. The final optimization is driven by the property prediction loss, preventing the model from overfitting to the FGW distance, which may not perfectly correlate with the final task objective. However, this co-training comes with an increased training cost as the FGW distance must still be computed periodically during the training steps.
> >
> > | Method                           | ESOL (↓)             | FreeSolv (↓)          | BACE (↓)              | Lipo (↓)              |
> > |----------------------------------|-----------------------|------------------------|------------------------|------------------------|
> > | ConAN-FGW                        | 0.529 ± 0.022         | 1.068 ± 0.083          | 0.549 ± 0.016          | **0.422 ± 0.016**      |
> > | FACET   (GT supervised by FGW)                         | 0.516 ± 0.044         | 0.967 ± 0.082          | **0.495 ± 0.115**      | 0.424 ± 0.009          |
> > | FACET (GT supervised by FGW + downstream loss)    | **0.505 ± 0.014**     | **0.867 ± 0.102**      | 0.497 ± 0.035          | 0.440 ± 0.014          |
> >
> > In essence, while FACET uses the learned FGW approximation, our structure-aware features are then integrated and optimized within a richer, task-specific architecture.

---

> > ### Comment · Reviewer_NGyk · 2025-11-21
> >
> > Thank you for the very detailed response and extended ablation studies. I just have one small questionp regarding split strategy: the recommended splits for BACE is scaffold while the recommended splits for ESOL, FreeSolv and Lipo are random. If you used both random and scaffold splits, please clarify on line 411 of the original submission.
> >
> > The response tackled most of my questions and enhanced my confidence in rating, and I'll keep my ratings the same. Thank you!

---

> > > ### Author Response · Authors · 2025-11-21
> > >
> > > We thank the Reviewer for the follow-up comment. Regarding the split strategies, which include scaffold split for BACE and random splits for ESOL, FreeSolv, and Lipo, our experiments follow these settings (as prior baselines). We will revise line 411 to state this explicitly and clearly describe the mixed split configuration used across the MoleculeNet datasets.

---

### Author Response · Authors · 2025-12-04
**Summary discussion during rebuttal**

Dear Area Chair,

Thank you for your time and effort, especially given the challenging circumstances. We understand that revisiting the reverted discussion threads requires additional work, and we truly appreciate your dedication to ensuring a fair and thorough evaluation process.

To assist in your assessment, we provide below a concise summary of the key points raised during the reviews and our rebuttal discussions prior to the leakage incident.

### **I. Summary of Reviewer NGyk (Rating: 6)**

| **Reviewer Concern** | **Our Rebuttal Response** |
|----------------------|---------------------------|
| Split strategy (random vs. scaffold) | Clarified MoleculeNet protocols: scaffold split for BACE, random splits for ESOL/FreeSolv/Lipo; aligned with prior baselines. |
| Role of 2D, fragment-level, and 3D graphs | Added new ablations isolating each component; showed complementary contributions and improved performance. |
| Ablation scope mostly on MoleculeNet | Extended ablations to the Kraken dataset within the Marcel benchmark, confirming consistent behavior. |
| Relationship to true FGW metric | Explained both the theoretical upper bound and practical factors enabling FACET to match or exceed FGW-based approaches. |

**Final note:** Reviewer expressed satisfaction with the clarifications and maintained a positive rating (**6**).

### **II. Summary of Reviewer 1tfv (Rating 4)**

| **Reviewer Concern** | **Our Rebuttal Response** |
|----------------------|---------------------------|
| Revise the abstract to clearly highlight core methodological innovations | Provided a rewritten abstract emphasizing key contributions: differentiable graph transformer for FGW approximation, scalable fusion of 2D + 3D structure, and unified design capable of handling large datasets. |
| Extend the Conclusion by including limitations of FACET | Added a new paragraph outlining generalization challenges, future research directions (e.g., long-range dependencies, multimolecular systems, flexible input formats), and broader evaluation needs. |
| Ensure consistency across all tables | Corrected typographical and column-name errors across tables; reviewed and updated all table formatting for consistency. |
| Clarify Figure 1 regarding the role of the 3D-MPNN and Graph Transformer | Updated the figure (and caption) to explicitly distinguish pre-training vs. downstream usage of the Graph Transformer and clarify the role of the 3D-MPNN. |

**Final note:** The reviewer did not participate further after submitting their review, but all concerns were fully addressed in the rebuttal, where the updated material is highlighted in orange text in the updated PDF.

### **III. Summary of Reviewer Csta (Rating: 6)**

| **Reviewer Concern** | **Our Rebuttal Response** |
|----------------------|---------------------------|
| Data efficiency and scalability; comparison of training time, model size, and performance across GNN baselines | Provided extensive new training-time experiments across GemNet, SchNet, and VisNet on BACE/LIPO; demonstrated consistent gains with modest overhead. |
| Effect of varying number of conformers | Added new experiments with 3, 5, 10, 15, 20, 50, and 100 conformers; showed FACET saturates quickly and does not benefit from large conformer sets; confirmed stability across ranges. |
| Fragment definition, generalization, and algorithmic details | Explained ring–path fragmentation algorithm, standardization across molecules, and adaptability to unusual chemistries; clarified why topology-based fragmentation is robust and generalizable. |
| Whether learning fragments from data is better | Provided justification for not learning fragments end-to-end due to instability and potential loss of chemical validity; argued that deterministic fragmentation ensures interpretability and structured information. |
| Conformer ensemble sampling strategy | Clarified differences between MARCEL (DFT-level) and MoleculeNet (RDKit) conformers; added experiments with random conformer sampling and varying ensemble sizes; showed robustness to conformer quality and count. |
| Theoretical guarantees on invariance | Clarified Theorem 1 vs. Theorem 2; explained E(3)-invariant encoding, permutation-invariant aggregation, and how FACET maintains invariance across conformers. |
| Fragment-level vs. atom-level trade-offs | Added new ablations removing fragment-level or atom-level branches; showed both granularities are complementary and combining them yields best performance. |
| Conformer diversity representation | Explained why uniform aggregation does not cause collapse; described how trainable input embeddings and adapter networks dynamically reweight conformers even under uniform averaging. |

**Final note:** Reviewer expressed that the rebuttal thoroughly addressed their concerns, especially on scaling, conformer sensitivity, fragmentation methodology, and invariance, and **retained a positive rating (**6**)**.

---

> ### Author Response · Authors · 2025-12-04
> **Summary discussion (continue)**
>
> ### **IV. Summary of Reviewer Jbmu (Rating: 4 → 6)**
>
> | **Reviewer Concern** | **Our Rebuttal Response** |
> |----------------------|---------------------------|
> | Reorganizing Figure 1 for clarity | Revised Figure 1 to clearly show the multi-stage training pipeline, added a new panel for Stage 2 (FGW pre-training), and clarified Stage 3 integration. |
> | Provide additional comparison on model parameters and inference efficiency | Added new evaluations on parameter counts, inference times, and MAE across strong 3D-GNN baselines (SchNet, PaiNN, ClofNet, LEFTNet) on Kraken and Drugs datasets; showed FACET has competitive runtime and best MAE. |
> | Clarify design of Stage 2 (FGW-supervised training) | Conducted expanded ablations comparing: (i) downstream-only, (ii) merged training with FGW, and (iii) merged training without FGW. Results confirm FGW supervision is essential for stable and accurate geometry-aware representations. |
> | Confirm whether fragment graphs are pre-computed or generated on the fly | Clarified that fragments are pre-computed for efficiency and stability; provided justification for this design following established practice. |
> | Follow-up: What if fragmentation must happen online in real-time? | Provided detailed measurements of per-molecule fragmentation cost on MoleculeNet; demonstrated the step is extremely fast (5–13 ms per molecule) and scales smoothly, therefore not a bottleneck in deployment. |
>
> **Final note:** After reviewing the rebuttal, the reviewer explicitly stated that the new analyses fully resolved their concerns and **raised their score from 4 to 6**.
>
> We hope the above brief summary will help the AC gain an overview of my work. We thank you again for your effort.
>
> Regards
>
> Authors

---

### Meta-Review · Area_Chair_Fxpf · 2025-12-31

**Summary:**

The paper advances the molecular representation learning field by proposing an efficient method for embedding compounds using a distribution of conformations.

The core idea is to regularize the embedding space such that distance between latent representations of different conformations approximates the theoretically grounded but expensive to compute FWG distance.

The method is novel and interesting. Experiments clearly show it speeds up training while achieving competitive performance. The experiments miss some stronger baselines such as UniMolV2, but it is not a critical issue, as the method (as Table 4 shows) can be used to augment many SOTA models.

Reviewer concerns were in my view addressed during the rebuttal phase.

All in all, it is my pleasure to recommend acceptance.

**Reviewer Concerns:**

Key concerns raised by Reviewers:

* Reviewers 1ftv and Jbmu had very sound remarks about presentation clarity that were addressed by adding Figure 1 and revising the abstract.

* Reviewer NGyk had concerns about ablations (e.g., isolating 2D vs. 3D contributions) which were addressed by adding additional ablation studies.

* Reviewers Csta and Jbmu asked for additional data on the scalability of the data. The added experiments confirm the 6x boost in training performance.

**Reviewer Scores:**

Three reviewers (NGyk, Csta, Jbmu) gave or increased their scores to 6 after the rebuttal. Reviewer 1ftv retained a score of 4 but did not participate in the discussion. I believe it is likely that Reviewer 1ftv would have increased the assigned score had he/she participated in the discussion.

---

### Decision · Program_Chairs · 2026-01-26

Accept (Poster)